# Seasonal vertical migration of large polar copepods reinterpreted as a dispersal mechanism throughout the water column
Katrin Schmidt [1] ✉, Barbara Niehoff[2,13], Astrid Cornils [2,13], Wilhelm Hagen[3], Hauke Flores [2], Céline Heuzé [4], Nahid Welteke [2], Nadine Knüppel[2], Sabrina Dorschner[3], Matthias Woll [2], Katie Jones [1], Giuliano Laudone[1], Robert G. Campbell [5], Carin J. Ashjian[6], Cecilia E. Gelfman [5], Katyanne M. Shoemaker[5], Rebecca Jenkins[5], Kristina Øie Kvile[7], Benoit Lebreton[8], Gaël Guillou[8], Clara J. M. Hoppe [2], Serdar Sakinan[9], Fokje L. Schaafsma[9], Nicole Hildebrandt [2,10], Giulia Castellani[2,11], Simon T. Belt[1], Allison A. Fong[2], Angus Atkinson [12] & Martin Graeve [2]

Seasonal vertical migration of large lipid-rich copepods is often described as a mass descent of animals when primary production ceases, with important implications for mesopelagic food webs and global carbon sequestration. This view ignores the existence of surface-resident individuals, but here we show that non-migrants can form a substantial part of the populations of polar migrant species. In the Central Arctic Ocean, the biomass-dominant *Calanus hyperboreus* was evenly distributed throughout the water column from November 2019 to March 2020, with ~20% of subadults and adult females remaining in the upper 200 m and ~41% migrating to 1000–2000 m. These vertical positions aligned with differences in the copepods' cholesterol content, which can enhance the tissue density at higher temperatures. Gonad development and the vertical distribution of their offspring indicate that both non-migrant and migrant females contribute to the population recruitment. We reinterpret copepod seasonal migration as a bet-hedging strategy that balances nutritional benefits near the surface with survival benefits at depth, and thereby contributes to the species' resilience under climatic change.

Across the globe, mammals, birds, reptiles, fish, insects and crustaceans travel between their summer and winter habitats in response to the seasonality in weather, food availability or risk of predation and disease. These long-distance migrations can be highly demanding but often lead to higher fitness in migrant populations than in non-migrants[1,2]. The enhanced fitness can come from multiple factors, e.g. access to resources such as water or food, longer daylight for foraging, suitable temperatures, shelter from predators or parasites, reduced competition and cannibalism, encounter with mates, behavioural interactions, separation from infected animals or contaminated habitats[3-5]. At the same time, these migrants provide major ecosystem services by dispersing vital nutrients, energy, pollen, seeds and other organisms across large distances, enhancing biodiversity and the connectivity of global regions[4,6].

One of the most extensive seasonal migrations on the planet is carried out by small pelagic crustaceans (class: Copepoda, genera: *Calanus, Calanoides, Neocalanus, Rhincalanus, Eucalanus*) which can descend from the ocean surface to depths of up to two to three thousand metres[7-9]. On a migration distance-to-body size scale, these copepods match African herd animals (i.e. >1 million times their own body size[10]). Like herd animals, the seasonally migrating copepods are primarily herbivorous during the short, intensive algal blooms and dominate the 'grazer' biomass of polar and upwelling regions[11-13]. The migration to deeper and often colder waters is

[1]School of Geography, Earth and Environmental Sciences, University of Plymouth, Plymouth, UK. [2]Alfred Wegener Institut Helmholtz-Zentrum für Polar- und Meeresforschung, Bremerhaven, Germany. [3]Marine Zoology, BreMarE — Bremen Marine Ecology, University of Bremen, Bremen, Germany. [4]Department of Earth Sciences, University of Gothenburg, Gothenburg, Sweden. [5]Graduate School of Oceanography, University of Rhode Island, Narragansett, RI, USA. [6]Biology Department, Woods Hole Oceanographic Institution, Woods Hole, MA, USA. [7]Norwegian Institute for Water Research (NIVA), Grimstad, Norway. [8]Joint Research Unit 7266 Littoral, Environnement et Sociétés (CNRS — University of La Rochelle), Institut du littoral et de l'environnement, La Rochelle, France. [9]Wageningen Marine Research, Den Helder, The Netherlands. [10]AquaEcology GmbH & Co. KG, Oldenburg, Germany. [11]Norwegian Polar Institute (NP), Framsenteret, Tromsø, Norway. [12]Plymouth Marine Laboratory, Plymouth, UK. [13]These authors contributed equally: Barbara Niehoff, Astrid Cornils. ✉e-mail: katrin.schmidt@plymouth.ac.uk

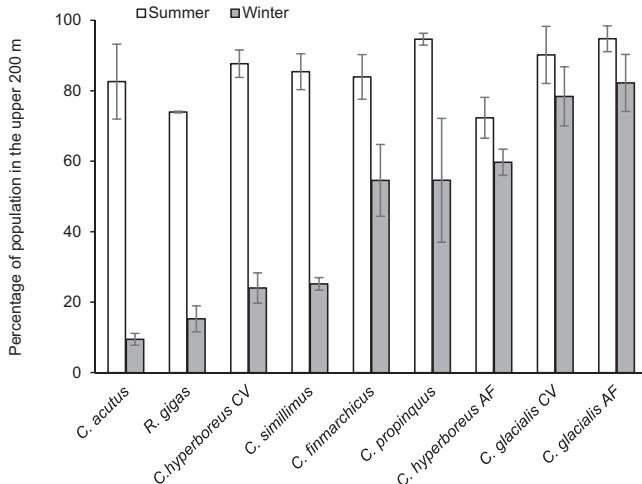

**Fig. 1 | Summer-winter residency of biomass-dominant seasonally migrating copepods in the upper water column of polar waters.** Mean percentages of *Calanus hyperboreus* and *C. glacialis*, Central Arctic Ocean (summer $n = 11$, winter $n = 26$)[36,37], *C. finmarchicus*, Svalbard Islands (summer $n = 3$; winter $n = 9$)[28,113], *Calanoides acutus, Calanus propinquus, Calanus simillimus, Rhincalanus gigas*, South Georgia, Scotia and Weddell Sea (summer $n = 3$; winter $n = 3$)[29,30,35] in the upper 200 or 250 m water column, with a total sampling depth of 1500 m[37] or 1000 m[28,35,36,110]. Error bars are standard errors. Only late developmental stages and/or adults are considered. The data from Rudyakov[36] and Ashjian et al.[37] were extracted from the 'historical data (1935–2016)'[9].

regarded as an adaptation to endure long periods of food shortage—either during winter, summer or the non-upwelling season - while reducing predation risk, metabolic costs, and exposure to advective processes in the upper ocean[14]. In preparation for these seasonal migrations, the copepods produce unique energy-rich fatty acids (FA) and fatty alcohols (FAlc) (i.e. long-chain monounsaturated 20:1 and 22:1 FA and FAlc) that can be traced through marine food webs from zooplankton to commercial fish stocks, seals, whales, and polar bears[12,15,16].

The actual mechanisms controlling these seasonal vertical migrations have been debated for decades[17–19]. However, an upsprung of interest has come from the realisation that large amounts of carbon are sequestered while the copepods rest at the ocean interior, relying on their lipid reserves and respiring $CO_2$ in waters that can remain isolated from the atmosphere for hundreds of years. This has been termed the 'seasonal lipid pump'[20,21], which on a local scale can sequester similar amounts of carbon to the gravitational flux of organic matter[20,22,23]. On a global scale, seasonally migrating copepods may contribute ~2.5% of the marine carbon sequestration (0.03–0.25 Pg C year$^{-1}$)[24]. However, these estimates often rely on rigid assumptions that are not always met in the field. In the North Atlantic and European Arctic, substantial parts of the seasonally migrating *Calanus finmarchicus* and *Calanus glacialis* populations stay near the surface in mid-winter[7,19,25–28] (Fig. 1). Likewise in the Southern Ocean, even the most renowned seasonal migrant, *Calanoides acutus*, maintains some of the population near the surface year-round, and this resident fraction is even greater for the other migrating copepods[29,30] (Fig. 1). Complexity is added by the fact that species might change their winter depth distribution depending on the location. In the Central Arctic Ocean (CAO), for instance, the overwintering population of the high-Arctic *Calanus hyperboreus* is described as centred in the upper 900 m[31], whereas in the Greenland Sea further South, they aggregated near the seafloor at 2500 m[32].

Several hypotheses have been proposed to explain the shallow overwintering depths of some of the seasonal migrants. These include: (1) insufficient lipid stores that prevent the initiation of descent[33], (2) attraction to food in surface waters, released for instance from ridging sea ice during

winter storms[28], (3) reduced predation risk from visual predators under sea ice[9], (4) avoidance of mesopelagic predators[28], (5) impaired downward migration due to positive buoyancy[34]. However, these hypotheses have rarely been tested in combination, and we therefore still lack a full mechanistic understanding of copepod diapause and seasonal migration[13]. Moreover, most winter net sampling campaigns in remote polar regions did not extend below 1000 m[28–31,35] and might therefore have missed parts of the copepod populations[32,36].

The Multidisciplinary Drifting Observatory for the Study of Arctic Climate (MOSAiC) expedition provided an ideal opportunity to address these research gaps. From November 2019 to March 2020 and in August-September 2020, zooplankton was sampled near weekly across five depth strata from 2000 m to the surface, while the vessel drifted across the CAO (≥85°N, Fig. 2a). Our aim was to re-investigate the seasonal vertical migration of the biomass-dominant copepod *C. hyperboreus* through multifaceted data streams. These include, firstly, the vertical distribution of subadults and adult females (AF) in comparison to winter data from the previous ice-drift expeditions in the CAO, 'Severny Polyus 1950–1956'[36] and 'SHEBA 1997/1998'[37]. Secondly, the role of internal (e.g. lipid reserves, tissue density) and external (e.g. physical forcing, food availability, predator abundance) factors for the initiation of seasonal descent and overwintering depth. Thirdly, the potential benefits of partial migration for population recruitment. Where possible, the congener species *C. glacialis* was examined for comparison. Due to the rapidly changing sea ice conditions in the CAO, this study may serve as a baseline for future field work and encourage new laboratory and modelling approaches.

## Results

### Seasonal differences in the vertical distribution of *C. hyperboreus* and *C. glacialis* in the CAO

The compilation of *C. hyperboreus* vertical profiles from MOSAiC and two previous ice-drift expeditions (Severny Polyus 1950–1956, SHEBA 1997/1998) shows a uniform pattern when based on a maximum sampling depth of 1000 m (MOSAiC and Severny Polyus) or 1500 m (SHEBA) (Fig. 2b). From June to August, >80% of the subadults (Copepodite stage V, CV) and AF resided in the upper 200 m water column. Lowest surface proportions occurred during the equinox in spring and autumn, especially during the SHEBA expedition, with ~10% AF and almost no CV in surface waters. In winter, on average, 40–60% of AF and 10–30% of CV remained between 200 and 0 m.

However, while the net sampling resolution was limited or variable for Severny Polyus and SHEBA, five depth strata down to 2000 m were consistently sampled during MOSAiC in winter (Nov. 2019–Mar. 2020, $n = 14$), summer (July 2020, $n = 3$) and late summer (Aug.–Sep. 2020, $n = 9$). The MOSAiC profiles allow a detailed examination of the data in the form of anomaly plots. The plots show that CV and AF strongly aggregate in surface waters (50–0 m) in July but keep a nearly even vertical distribution throughout the water column in winter (Figs. 2c and S1, Table S1). Across the fourteen winter sampling events, ~20 ± 11% of the AF and CV population remained in the upper 200 m water column, ~11 ± 7% resided at 500–200 m, 27 ± 9% at 1000–500 m and 41 ± 16% at 2000–1000 m (Fig. S2). In comparison, the co-occurring *C. glacialis* also aggregated between 50 and 0 m in summer and spread out across the upper 200 m in winter but had year-round strong negative anomalies in deeper waters (Fig. 2c).

Mature *C. hyperboreus* AF with ripe oocytes were sampled between Nov. 2019 and May 2020, with a peak occurrence in Jan.–Mar. 2020, whereas mature *C. glacialis* were found in summer. This suggests that the large copepod eggs and *Calanus* nauplii, collected during the winter months, were derived at least partly from *C. hyperboreus*, and those in late summer from *C. glacialis*. In winter, copepod eggs and *Calanus* nauplii occurred rather evenly throughout the water column, with some aggregations in the upper ocean (eggs) or the 1000–500 m depth stratum (nauplii) and negative

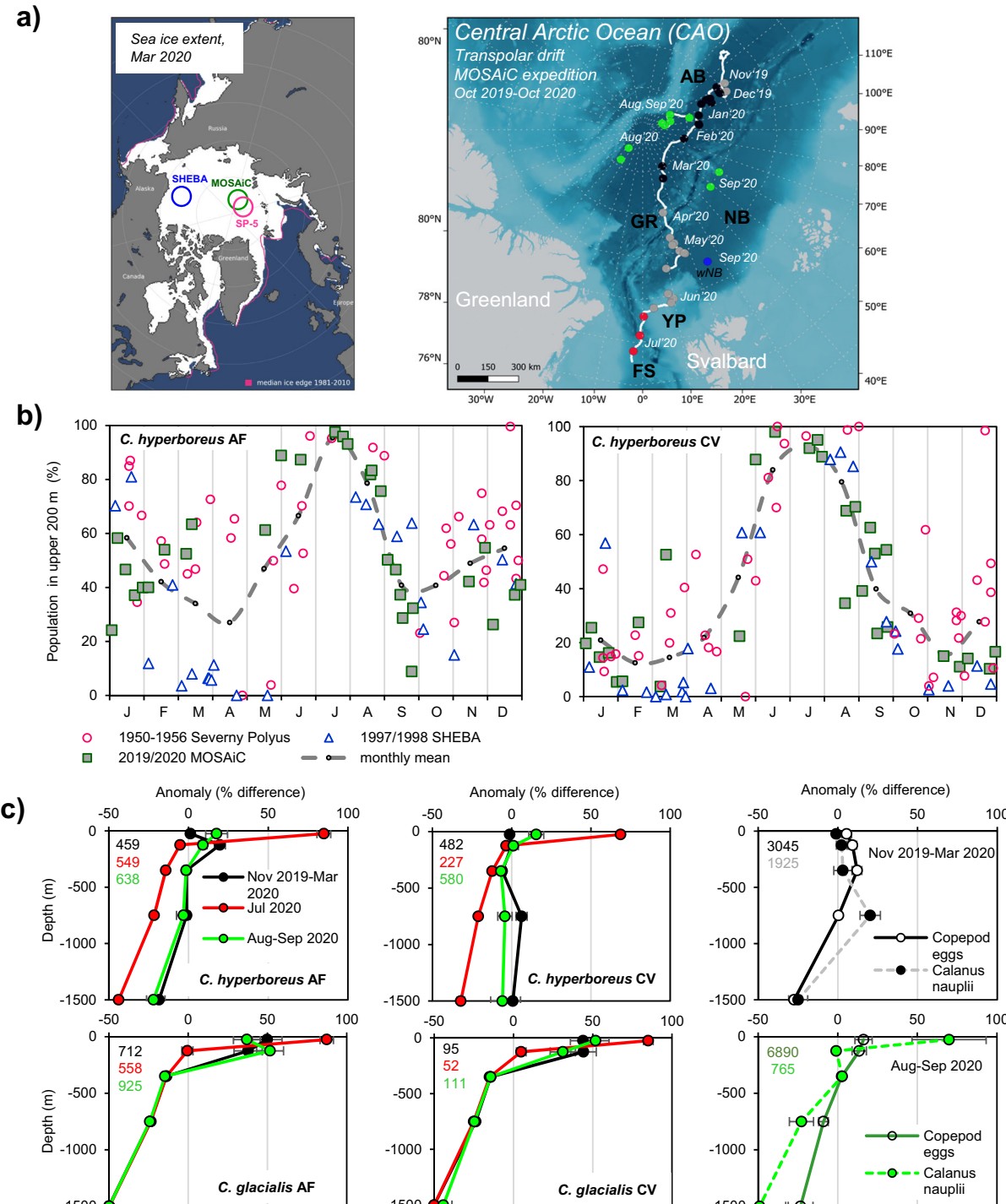

**Fig. 2 | Vertical distribution of *C. hyperboreus* and *C. glacialis* during MOSAiC (Nov. 2019–Sep. 2020) and the previous ice-drift expeditions in the Central Arctic Ocean (Severny Polyus 1950–1956, SHEBA 1997/1998). a** (left) Winter locations of the MOSAiC expedition and previous ice-drift expeditions; SP-5 (Severny Polyus-5 1955/56), SHEBA (Surface heat budget of the Arctic Ocean 1997/1998), with winter sea ice extent for March 2020 provided by NOAA. (right) Trajectory of the MOSAiC expedition (white line). Grey dots represent locations of shallow MultiNet sampling. Deep nets were deployed in Nov. 2019–Mar. 2020 (black), July 2020 (red) and Aug.–Sep. 2020 (green dots). Separate stations during Aug'20 and Sep'20 were sampled while the vessel was in transit. AB Amundsen Basin, GR Gakkel Ridge, NB Nansen Basin, YP Yermak Plateau, FS Fram Strait, wNB western NB. **b** Seasonal cycle of the residency of *C. hyperboreus* copepodite stage V (CV) and adult females (AF) in the upper water column. The cut-off depth

for the upper ocean was 200 m for MOSAiC and SHEBA (1997/1998), and ~250 m for Severny Polyus (1950–1956), and the total depth was 1000 m for MOSAiC and Severny Polyus, and ~1500 m for SHEBA. The data were extracted from the 'historical data (1935–2016)'[9]. **c** MOSAiC expedition: Average vertical distributions of *C. hyperboreus* (top) and *C. glacialis* (below) AF, CV, copepod eggs and *Calanus* nauplii in Nov. 2019–Mar. 2020 (CAO, ≥85°N, *n* = 14), July 2020 (Fram Strait, *n* = 3) and Aug.–Sep. 2020 (CAO, ≥85°N, *n* = 9) are presented as the percentage anomaly from an even distribution of animals throughout the water column. Total abundances within the upper 2000 m water column are given as ind m⁻² (left upper corner of each plot). Error bars are standard errors. The seasonal cycles of *C. hyperboreus* (CV, AF) and *C. glacialis* (CV, AF) abundances (ind m⁻³) above 200 m and below 500 m are compiled for MOSAiC and historical data (1935–2016) in Fig. S3.

anomalies below 1000 m (Fig. 2c). In late summer, nauplii showed aggregations in surface waters (50–0 m), and strong negative anomalies below 500 m, as seen for *C. glacialis* AF and CV.

The abundances (ind m$^{-3}$) of *C. hyperboreus* and *C. glacialis* encountered during MOSAiC overlap with the compiled historical data (1935–2016) from the CAO (0.1–10 ind m$^{-3}$) (Fig. S3).

### Changes in abundance and body condition of *Calanus hyperboreus* over the winter months

While the total abundances of AF in the upper (200–0 m) and lower (2000–200 m) water column did not show any significant trends over the four winter months (Nov. 2019–Mar. 2020), the abundances of mature AF followed a bell-shaped curve in the upper water column with a peak in Jan./Feb. 2020 (Fig. 3a, b). The abundances of copepod eggs significantly increased in the upper 200 m over the winter, whereas the abundances of *Calanus* nauplii increased in both the upper and especially the deeper ocean (Fig. 3c, d). At the same time, the dry mass, lipid content and proportion of polyunsaturated FA significantly dropped for AF in the upper ocean, while a similar negative trend was insignificant for AF in deeper waters (Fig. 3e–g). Expressed as per individual, the average loss over the four winter months was 25% in dry mass (5.30–3.95 mg ind$^{-1}$), 35% in lipid content (2.52–1.62 mg ind$^{-1}$) and 45% in PUFA content (0.23–0.12 mg ind$^{-1}$). The concentration of the sea ice diatom-derived highly branched isoprenoids (HBIs) IP$_{25}$ and IPSO$_{25}$ significantly increased in AF from the upper ocean towards spring and indicates potential food uptake, whereas at depth those trophic markers showed no significant changes (Fig. 3h).

### Feeding history and lipid stores of resident and migrating *C. hyperboreus* in late summer

*C. hyperboreus* sampled from different depth strata in Aug.–Sep. 2020 in the CAO were separated by Principal Component Analysis (PCA) based on their biochemical composition (Fig. 4a). Both AF and CV sampled at depth (2000–1000 m and 1000–500 m) had high δ$^{13}$C values, high diatom-to-flagellate FA ratios and high proportions of the *Calanus*-produced 20:1 and 22:1 FA and FAlc. In contrast, copepods from the upper water column (200–50 m and 50–0 m) contained more polyunsaturated fatty acids (PUFA), more FA from flagellates and heterotrophic food, and had higher δ$^{15}$N values (Fig. 4a). These differences in the copepods' trophic marker signature indicate a different feeding history, with specimens from an intermediate depth (500–200 m) overlapping with both groups.

Despite the different feeding history, the copepods' key body characteristics (prosome length, dry mass) and overall lipid stores (total lipid content, proportion of wax esters, potential ATP gain from beta-oxidation) were either identical or of minimal (<3%) difference between individuals from surface vs. depth (Figs. 4b–d and S4, Table S2).

Separate analyses of PUFA in polar (deriving from structural lipids) and neutral fractions (deriving from storage lipids) confirm a PUFA predominance in biomembranes, with 75–77% of total FA (TFA) in the polar fraction, in both AF and CV, regardless of their sampling depth (Fig. 4e, Table S2). In neutral lipids, the PUFA content was overall lower and more variable (Fig. 4f). AF at the surface had an average PUFA content of 19.9 ± 4.3% in their neutral fraction, those at depth of only 15.6 ± 1.8% (Mann–Whitney *U*-test, *p* = 0.016). In CV, the difference was even more pronounced with 34.3 ± 6.0% PUFA at the surface and 24.5 ± 6.6% at depth (Mann–Whitney *U*-test, *p* = 0.005; Fig. 4f). The lower PUFA proportions in the neutral lipid fraction were balanced by higher proportions of the *Calanus*-produced 20:1 and 22:1 isomers (Fig. 4g).

Compound-specific stable isotope analysis (CSIA) showed little surface vs. depth differences in the δ$^{13}$C values of the copepods' key FA and FAlc, except for the three PUFA, 18:4(*n*−3), 20:5(*n*−3) and 22:6(*n*−3) (Figs. 4h–j and S5, Table S2). PUFA from copepods sampled at depth were $^{13}$C-enriched by 2–5‰ compared to those from surface copepods (e.g. CV PUFA, depth: −30.0 ± 1.9‰, surface: −33.5 ± 2.5‰, *p* = 0.001), which also points to a different feeding history.

### Physical constraints on the copepods' seasonal vertical migration

During the MOSAiC expedition, there were a few incidences of vertical instability in the upper water column during Aug.–Sept. 2020 (blue patches on Fig. 5a), which resulted in vertical mixing and potentially in the downward advection of copepods. Apart from those, the CAO was well stratified for most of the year with increasing density throughout the water column (red colour on Fig. 5a) and strongest barriers within the halocline, at 30–40 m depth (dark red regions on Fig. 5a). These density barriers were much less pronounced in April and May 2020, which would have made upward or downward migration easier. Moreover, there were strong day-to-day changes in water column stratification that could have supported copepod downward (blue areas in Fig. 5b) or upward movement (red areas in Fig. 5b), even if copepods remained immobile. Such 'downward corridors' occurred repeatedly at the end of summer (blue vertical regions in July–Sept. 2020, Fig. 5b).

Temperature is another physical constraint on the copepods' buoyancy and therefore vertical migration, as lipids have a higher thermal expansion and compressibility than seawater. In late summer, *C. hyperboreus* encounter lowest water temperatures near the surface (~-1.6 °C) and highest temperatures (>1 °C) in the Atlantic Water layer between 200 and 500 m depth (Fig. 6a). Density measurements can reveal whether their lipid-rich tissue expands under increasing temperatures, potentially leading to uplift when the copepods enter the warmer Atlantic Water layer. From the samples prepared for biochemical analyses ("Results" sub-section "Feeding history and lipid stores of resident and migrating *C. hyperboreus* in late summer"), we compiled small subsamples for eleven density measurements at six temperatures that cover the above-mentioned range (−2, −1, 0, 1, 2, 3 °C). Ten of these samples derived from the CAO (≥85°N) and included *C. hyperboreus* AF from six depth strata, *C. hyperboreus* CV and *C. glacialis* AF from two depth strata, and one sample of *C. hyperboreus* AF was collected on the return journey from the western Nansen Basin (wNB). Overall, the density of the dried tissue samples increased, rather than decreased, with rising temperature (Fig. 6b). A detailed examination revealed that it was primarily copepods collected from warmer parts of the water column that showed the highest densities (Fig. 6c). For instance, *C. hyperboreus* AF sampled in the upper 100 m water column had a significantly lower tissue density than those from 2000–1000 m (Mann–Whitney *U*-test, *p* = 0.031, AF 100–0 m: 0.90 ± 0.06 g cm$^{-3}$; AF 2000–1000 m: 1.19 ± 0.13 g cm$^{-3}$). An exception were the *C. hyperboreus* AF from the wNB, which had the lowest densities despite the highest in situ temperatures (Mann–Whitney *U*-test, *p* = 0.013, AF wNB: 0.687 ± 0.09; AF 100–0 m: 0.90 ± 0.06 g cm$^{-3}$) (Fig. 6c). However, copepods from this location had the lowest cholesterol content within their lipid fraction, which seems to be a predictor of the maximum density that was reached across the tested temperature range ($R^2$ = 0.5741, *p* = 0.007, *n* = 11, Fig. 6d).

### Vertical distribution of *Calanus* predators

A two-step approach was used to assess whether *C. hyperboreus* developmental stages (eggs to AF) experience a higher predation risk when living in surface waters of the high Arctic than at depth. First, potential *Calanus* predators were identified among 18 zooplankton taxa and polar cod (*Boreogadus saida*), and second, the vertical distribution of these predators was investigated. The 18 taxa were selected based on their relatively large size, high abundance, or high biomass, which provided sufficient material for biochemical analysis. We used the *Calanus*-produced 20:1 and 22:1 FA isomers as a tracer of the taxon's predation on *Calanus*. These FA occur in all *Calanus* developmental stages (including eggs) but are usually most abundant in the lipid-rich older stages (Hirche and Kattner[38]). *Calanus hyperboreus* had the highest proportions of the 20:1, 22:1 FA among the three *Calanus* species encountered in the CAO (Fig. 7a). The 20:1 and 22:1 FA accumulate in lipid-rich tissue, which can lead to bias, as seen in polar cod (B.s.: 20:1, 22:1 in liver: >40%, in muscle: 10%). We assume that all taxa with >10% of 20:1, 22:1 FA in their TFA are at least occasionally preying on *Calanus* life stages. In Aug.–Sep. 2020, these taxa were polar cod, cnidarians,

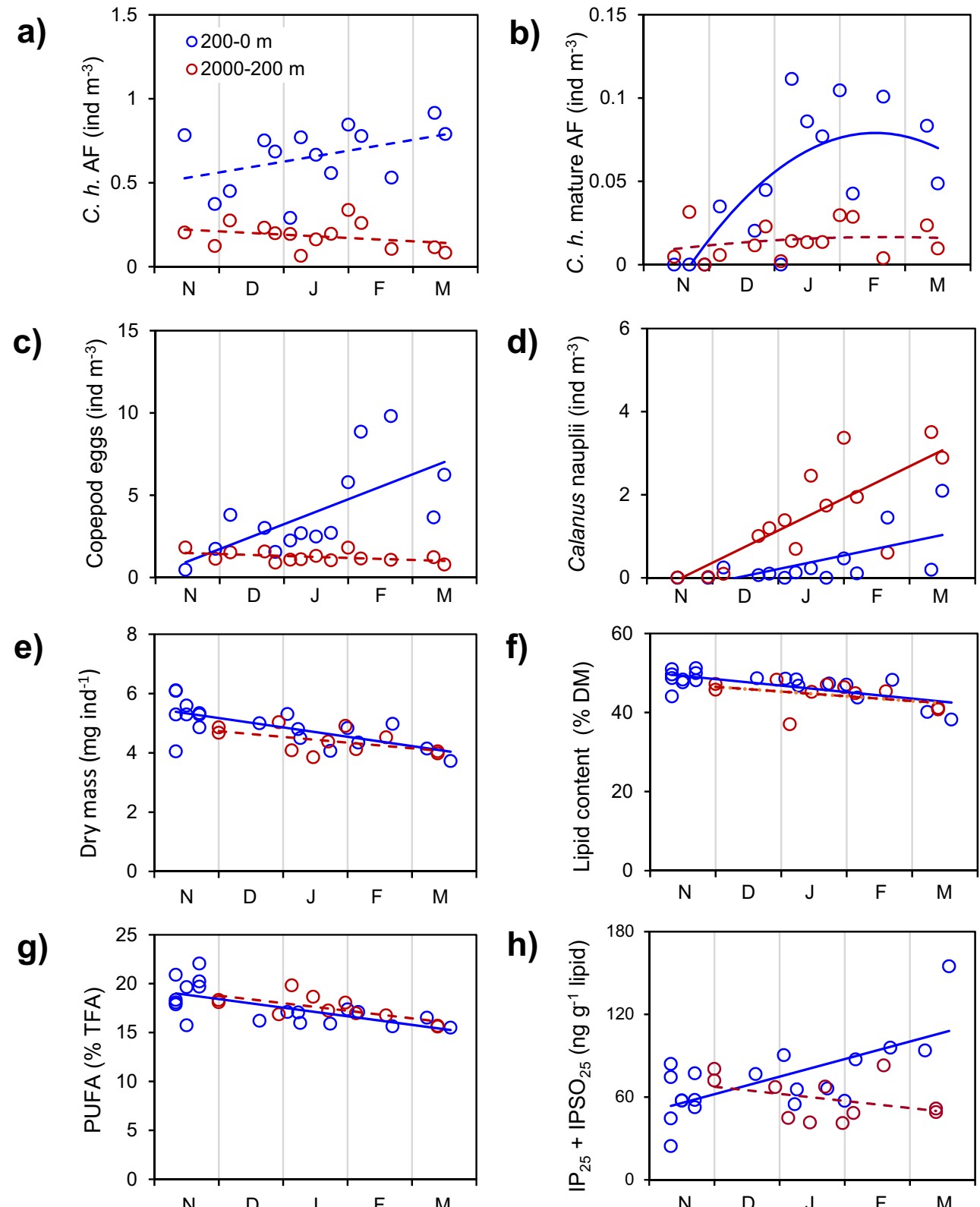

**Fig. 3 | Timelines of *Calanus hyperboreus* abundances, body reserves and trophic markers in the upper and deeper ocean over the winter months (MOSAiC, Nov. 2019–Mar. 2020).** Abundances (ind m$^{-3}$) of **a** adult females (AF), **b** mature AF, **c** copepod eggs and **d** *Calanus* nauplii. Body reserves expressed as **e** dry mass (mg ind$^{-1}$), **f** lipids (% dry mass) and **g** polyunsaturated fatty acids (% total fatty acids), and **h** the diatom-specific trophic markers IP$_{25}$ and IPSO$_{25}$ (ng g$^{-1}$ lipid). Lines show trends of the data.

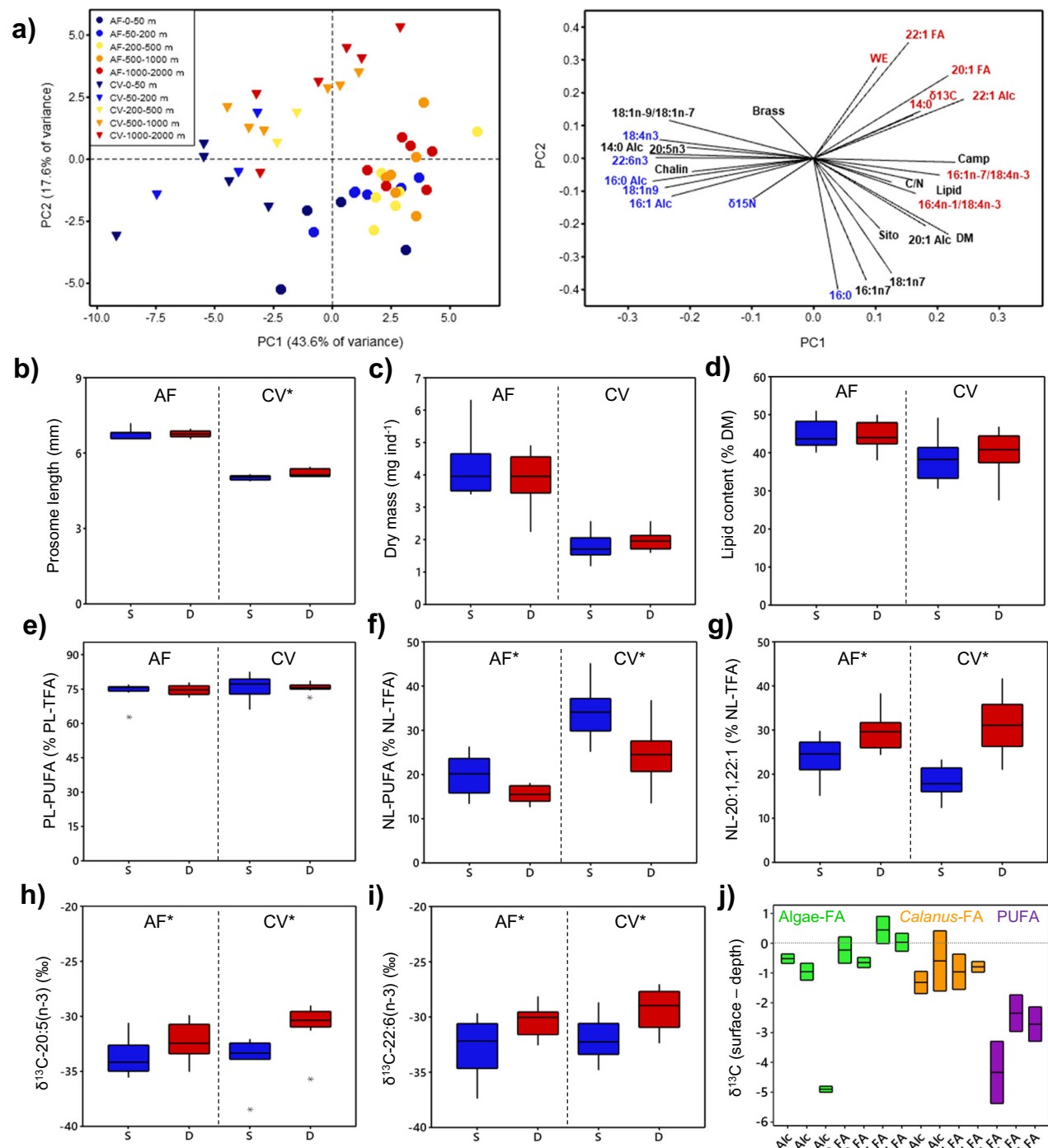

**Fig. 4 | Feeding history and lipid composition of *C. hyperboreus* sampled in shallow *versus* deep waters of the CAO in late summer (MOSAiC, Aug.–Sep. 2020). a** Principal component analysis of *C. hyperboreus* biochemical composition. Score—(left) and loading plot (right) for adult females (AF) and copepodite stage V (CV) from six stations with five sampling strata: 50–0 m (dark blue); 200–50 m (light blue); 500–200 m (yellow); 1000–500 m (orange); 2000–1000 m (red). In the score plot, parameters are presented in colour if the median values were significantly higher at depth (red) or surface (blue) (details in Table S2). Each symbol represents a pooled sample of ~10 AF or ~20 CV. Body characteristics of AF and CV from the surface (S, blue, 50–0 m and 200–50 m, $n = 10$) vs. depth (D, red, 1000–500 m and 2000–1000 m, $n = 12$): **b** prosome length, **c** dry mass, **d** lipid content, **e** PUFA content in the polar lipids (PL, membranes, <20% of total lipids) and **f** PUFA content in neutral lipids (NL, wax esters, >80% of total lipids), **g** *Calanus*-produced 20:1 and 22:1 FA, **h** the δ¹³C

values of the essential PUFA 20:5($n$−3) and **i** δ¹³C values of the essential PUFA 22:6($n$−3). Adult females (AF) and copepodite stage V (CV) were labelled with an asterisk ('AF*' or 'CV*') if they showed significant differences between specimens sampled at surface vs. depth (Mann–Whitney *U*-test, details in Table S2). **j** Differences in the δ¹³C values of key FA and FAlc in *C. hyperboreus* AF and CV from depth and the upper water column. Trophic marker fatty acids: *flagellates*: 18:4($n$−3); 22:6($n$−3); *diatoms*: 16:4($n$−1); 20:5($n$−3); *PUFA (polyunsaturated fatty acids)*: 16:3($n$−4), 16:4($n$−1), 18:3($n$−3), 18:4($n$−3), 20:4($n$−3), 20:5($n$−3), 22:5($n$−3), 22:6($n$−3); *20:1 isomer*: 20:1($n$−11), 20:1($n$−9), 20:1($n$−7), 20:1($n$−5); *22:1 isomer*: 22:1($n$−11), 22:1($n$−9), 22:1($n$−7). Phytosterols: epi-brassicasterol (Brass), ß-sitosterol (Sito), chalinasterol (Chalin), campesterol (Camp). The box plots show the median (central line), the upper and lower quartiles (box), the minimum and maximum 25% of scores (lower and upper whiskers) and outliers (asterisks).

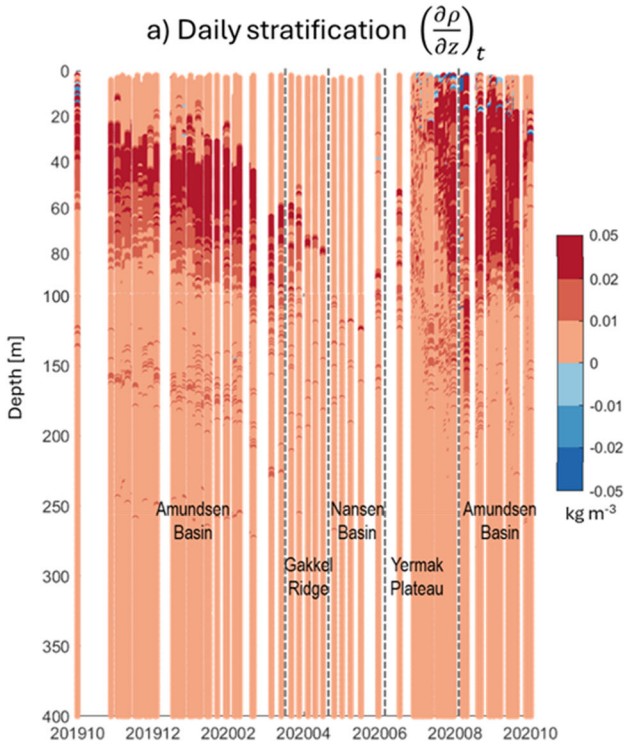

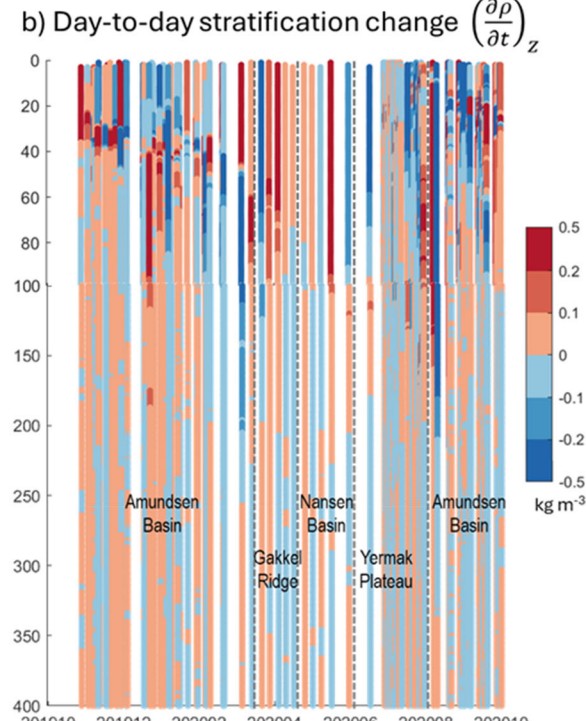

**Fig. 5 | Physical constraints on seasonal vertical migration (MOSAiC, Oct. 2019–Oct. 2020).** For each hydrographic profile, difference in density **a** for that profile, between one depth level and 1 m below, i.e. similar to stratification, and **b** for each depth level, between that profile and the subsequent one, i.e. similar to changes in stratification. Red indicates that density increases with depth (**a**) or time (**b**). Conditions favourable for downward migrations are in blue: the surrounding waters' density decreases, and the copepods are less buoyant.

pelagic amphipods (*Cyclocaris guilelmi, Themisto libellula, T. abyssorum*), sympagic amphipods (*Onisimus* spp., *Eusirus holmii*), decapods (*Hymenodora glacialis*), chaetognaths, copepods (*Paraeuchaeta* spp., *Scaphocalanus* spp.), pteropods (*Limacina helicina*) and ostracods. The lipid-poor copepod *Metridia longa* had low proportions of 20:1, 22:1 FA in Aug.-Sep. ($1.0 \pm 0.1\%$, $n = 12$) but higher values in Nov.–Mar. ($2.5 \pm 0.4\%$, $n = 9$), likely indicating their winter feeding on *Calanus* eggs. All 'potential predator' taxa, except *L. helicina*, occupied a higher trophic level (TL) than *C. hyperboreus* based on their $\delta^{15}$N values (Fig. 7a).

For those potential predators that frequently occurred in the MultiNet samples, the anomaly of their vertical distribution was calculated for Nov. 2019–Mar. 2020 and Aug.–Sep. 2020 (Fig. 7b). Most taxa showed little seasonal differences in their vertical distribution, but some shifted towards shallower waters in winter (*Themisto abyssorum, Paraeuchaeta* spp., ostracods, *Scaphocalanus magnus, Hymenodora glacialis*). The positive anomalies indicate that potential *Calanus* predators are widely spread throughout the water column of the CAO, with some aggregating in surface or subsurface waters, and others in the Atlantic Water layer or below. Only the deepest sampling stratum (1000–2000 m), where most taxa showed a negative anomaly, is potentially a zone of lower predation risk for *Calanus*. Based on their total estimated biomass, cnidarians, chaetognaths and other copepod species are likely the main *Calanus* predators in the CAO (Fig. 7b), while ctenophores and Arctic cod (*Arctogadus glacialis*) were not considered here.

## Discussion

The traditional view that seasonally migrating *Calanus* collectively translocate to great depth[13,14] likely derived from pioneering studies, for example by Østvedt[39] in the Norwegian Sea, who found 87% of *C. hyperboreus* AF and 93% of CV below 1000 m and the entire population below 600 m during winter[39]. However, for the cold-water species *C. hyperboreus*, the Norwegian Sea is at the fringe of its habitat range, and even in summer, 50–70% of the subadult/ adult population avoided the warm Atlantic surface waters

(~9 °C) and instead remained in polar deep waters (−1 °C) below 600 m[39]. The lack of young copepodites in the population suggests that older life stages are advected rather than successfully recruiting here[22,36]. In contrast, Dawson[31] suggested that in the CAO, the *C. hyperboreus* population is centred in the upper 900 m year-round, but he rarely sampled below 400 m in winter and might have missed part of the population[31].

Our study shows both a surface-resident part of the subadult/adult population (~20% above 200 m) and a deep overwintering part (~41% below 1000 m) (Fig. 8). The only previous study that deployed deep nets in the CAO throughout the winter found similar *C. hyperboreus* proportions below 1000 m (CV: 35%, AF: 18%)[36], and the surface proportions also match our findings (Fig. 2b). We therefore suggest that within a water column with suitable low temperatures as found in the CAO, the winter distribution of *C. hyperboreus* resembles a deep dispersal rather than a strict translocation to depth. Other seasonal migrants such as *C. acutus, C. propinquus, C. simillimus, Rhincalanus gigas* and *C. glacialis* fit a similar picture, where individuals disperse to various depths of the water column, but a substantial part remains in the upper ocean year-round (Fig. 1). Insufficient lipid reserves can be ruled out as an explanation for the lack of descent as *C. hyperboreus* AF and CV from surface *vs.* depth were equally lipid-rich (Fig. 3f). Moreover, faster gonad development in AF, sampled in the upper 200 m than at depth[7] (Fig. 3b this study) despite equal or even lower temperatures, is likely supported by the uptake of food. Both stomach content analysis (Shoemaker, personal communications) and trophic marker concentrations (Fig. 3h) indicate winter feeding. We therefore conclude that the non-migrants are not failures or outliers, but a healthy part of the recruiting population. The occurrence of all developmental stages from eggs and nauplii to young copepodites and subadults/ adults suggests that the CAO can be a suitable habitat for *C. hyperboreus* to complete their life cycle (Fig. S6)[40,41].

Our observations suggest that, firstly, the behavioural plasticity in seasonal migrating copepods is greater than previously acknowledged, and secondly, that biogeochemical budgets assuming full-scale migration of

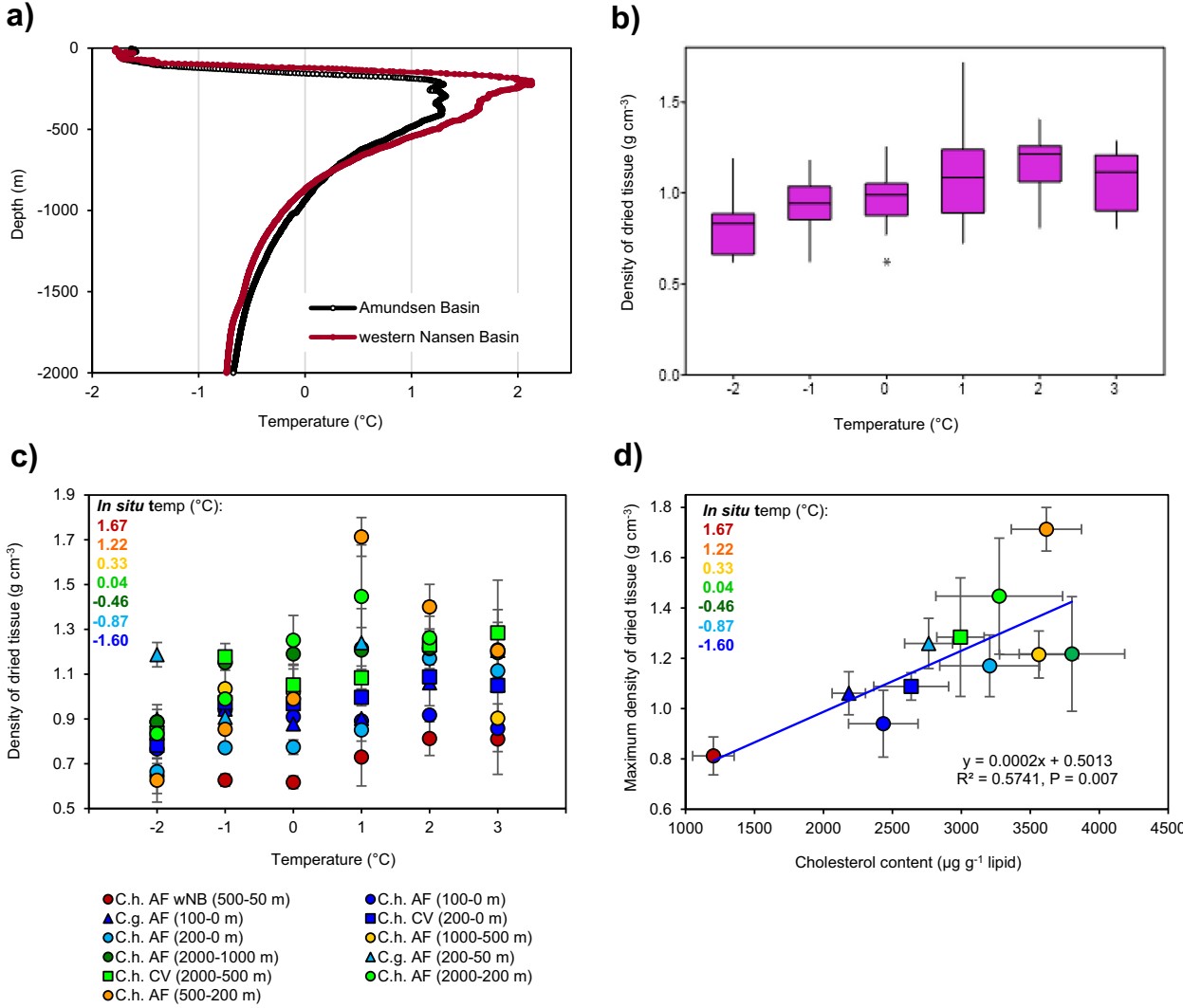

**Fig. 6 | Tissue density of *Calanus hyperboreus* from different sampling depths of the CAO (MOSAiC, Aug.–Sep. 2020). a** Vertical profiles of temperature across the upper 2000 m water column from six stations in the CAO (≥85°N, average) and one station at the western Nansen Basin (wNB) with a stronger influence of warmer Atlantic water (locations in Fig. 2a). **b** Box plot of tissue densities from eleven samples of *C. hyperboreus* and *C. glacialis* over a temperature range from −2 °C to 3 °C. **c** Tissue density of individual *Calanus* samples that contributed to the box plot in (**b**). Samples are colour-coded by the average in situ temperature of their depth strata. **d** Relationship between maximum tissue density and cholesterol content of the samples. The box plots show the median (central line), the upper and lower quartiles (box), the minimum and maximum 25% of scores (lower and upper whisker) and outliers (asterisk). The error bars represent the standard error.

copepod species likely overestimate the vertical transport of carbon or nutrients. Dispersal rather than translocation to depth is a very different concept of understanding seasonal migration. Below, we discuss key aspects related to winter dispersal, including the benefits for the population, the physical mechanisms, the evolutionary context and the implications for global carbon export estimates.

The results from the CAO suggest that for seasonally migrating *Calanus* species, the observed 'winter dispersal' is more beneficial than the proposed 'translocation to depth' strategy. Firstly, a surface-resident and a migrating part of the population at multiple depths gives more flexibility under highly variable polar conditions. The active *C. hyperboreus* near the surface prioritises food uptake over predator avoidance, whereas the opposite is true for the dormant *C. hyperboreus* at depth, in line with the concept of 'bet-hedging'[42]. In late summer 2020, surface-dwelling *C. hyperboreus* took advantage of an autumn phytoplankton bloom that contained more PUFA than the earlier ice algae bloom[43] and changed their FA composition accordingly, while copepods that had already migrated to depth missed out on this bloom (Fig. 4). *Calanus* eggs

contain high proportions of PUFA (32-38% TFA)[38,44], and any PUFA surplus in the AF's neutral lipids can support their subsequent egg production and nauplii hatching success[45]. This is especially relevant for *C. hyperboreus* that can start egg production soon after the late summer bloom[8]. Other advantages of remaining near the surface throughout the winter come from occasional winter food, e.g. microbes and algae released from the abrasion of adjacent ice floes[46], and their ability to directly monitor the onset of the spring bloom[47] that can occur as early as mid-March in the central Arctic[48]. In contrast, copepods at depth rely on indirect cues (e.g. photoperiod, exhaustion of lipid reserves or an internal biological clock) and require some time to respond. This can lead to a phenological mismatch as the onset of the short Arctic bloom season may vary by 4–6 weeks even at similar latitude[49]. On the other hand, high *Calanus* biomass in the stomachs of visual predators, e.g. little auk or polar cod[26,50], underlines the potential penalty for remaining near the surface.

Another advantage of winter dispersal comes from reducing predator-prey encounter probabilities. In the Southern Ocean, even the strongly

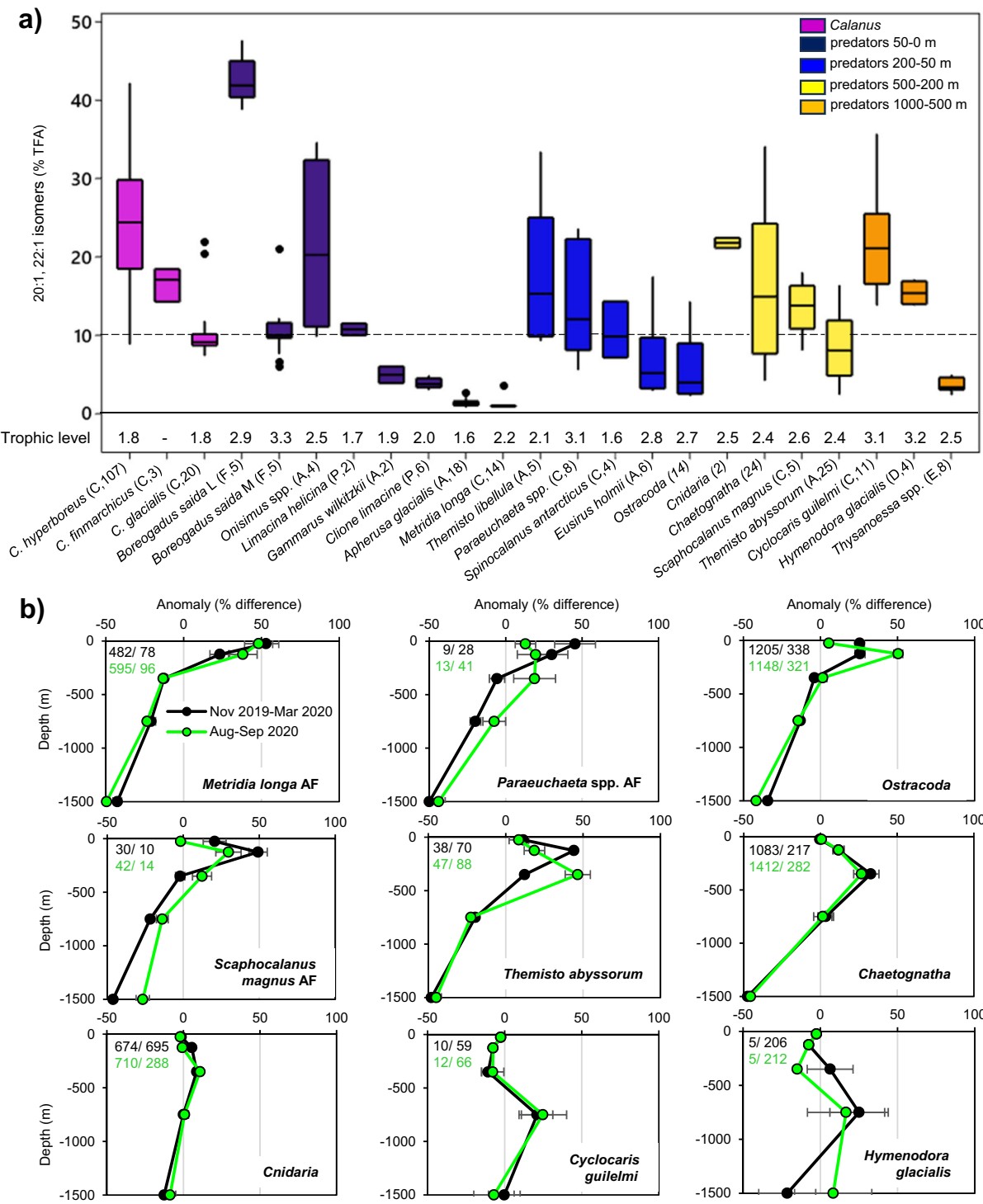

**Fig. 7 | Potential predators of *Calanus* life stages and their vertical distribution in the Central Arctic Ocean (MOSAiC, Nov. 2019–Mar. 2020, Aug.–Sep. 2020).**
**a** Potential predators are identified by the proportion of the *Calanus*-produced fatty acids (20:1, 22:1 isomers) in their total FA pool and their trophic level in Aug.–Sept. 2020. Polar cod (*Boreogadus saida*) liver (L) and muscle (M) were analysed separately. Individuals identified to species or genus level are copepods (C), fish (F), amphipods (A), pteropods (P), decapods (D) and euphausiids (E). In brackets, the number of replicate samples is given. The potential predators were colour-coded based on their anomalies from a homogeneous vertical distribution in Aug.–Sept. 2020, with the strongest positive anomalies occurring at 50–0 m (dark blue), 200–50 m (light blue), 500–200 m (yellow) or 1000–500 m (orange). None of the

potential predators showed a positive anomaly at 2000–1000 m. The box plots show the median (central line), the upper and lower quartiles (box), the minimum and maximum 25% of scores (lower and upper whiskers) and outliers (asterisks).
**b** Average vertical distribution of nine potential predators in Nov. 2019–Mar. 2020 (*n* = 14, black) and Aug.–Sep. 2020 (*n* = 9, green) are presented as the anomaly from an even distribution. Groups/taxa are sorted from top-left to bottom-right by their aggregation in increasingly deeper water. The error bars represent the standard error across the multiple sampling events. The symbols are placed in the middle of each stratum. The total abundances (ind m$^{-2}$) and total carbon biomass (mg C m$^{-2}$) of each group/taxon within the 2000 m water column are given in the left-hand corner of the panel for Nov.–Mar. (black) and Aug.–Sep. (green).

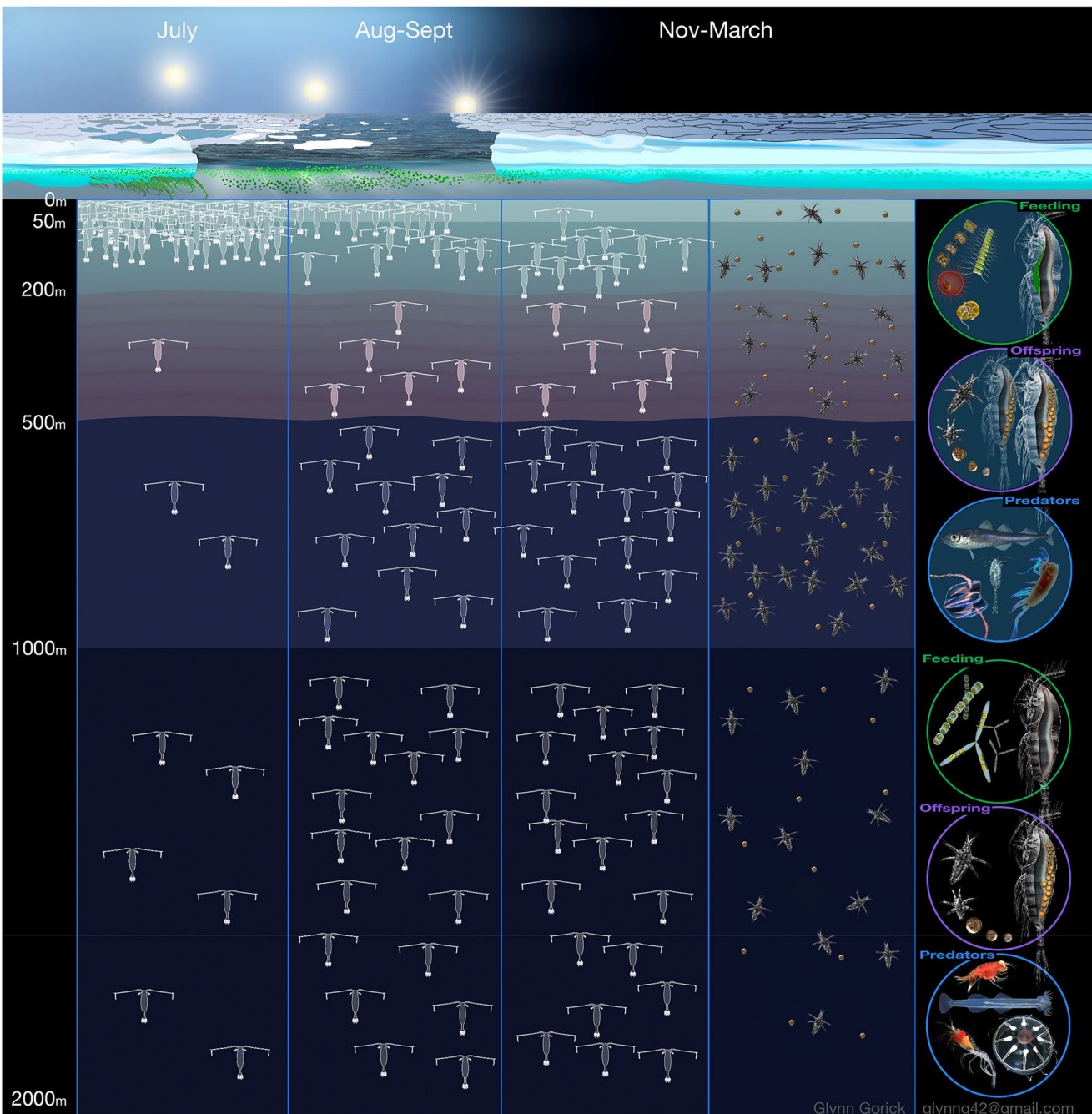

**Fig. 8 | Conceptual model of the seasonal vertical dispersal of *Calanus hyperboreus* in the Central Arctic Ocean during the summer-winter transition, with related aspects of feeding history, predator avoidance and recruitment.** The depicted copepods represent subadult (CV) and adult females (AF) of *Calanus hyperboreus*, while the copepod eggs and *Calanus* nauplii were not identified to species level but coincide with the winter reproduction period of *C. hyperboreus*. For each season and life stage, 50 specimens illustrate the vertical distribution observed during MOSAiC based on the relative abundance data (%) presented in Table S1. In summer, >80% of CV and AF accumulate in the upper 200 m water column, while in winter, the population spreads evenly across the 2000 m water column. During the winter dispersal, ~30% of AF and ~10% of CV reside in the upper 200 m (depicted as 10 specimens, 20%), while ~31% of AF and ~51% of CV migrate to waters deeper than 1000 m (depicted as 20 specimens, 41%). The circles to the right indicate differences in the feeding history, reproduction and types of predators between the upper (circles 1–3) and the deeper ocean (circles 4–6). In the upper 200 m, females mainly fed on non-diatom taxa during late summer, numbers of mature females and eggs are high during winter and key predators include other copepod species that feed on the offspring (e.g. *Metridia longa*, *Paraeuchaeta* spp.), polar cod (*Boreogadus saida*) and ctenophores. In deeper water, trophic markers indicate a feeding history on ice-associated diatoms in late summer, the number of mature females and eggs is lower and key predators are cnidarians, chaetognaths and deep-sea amphipods and shrimps.

migratory *C. acutus* disperses over nearly 1000 m during winter (300–1300 m)[51], which likely aids predator avoidance through the absence of layers of concentrated, defenceless copepods. Our study in the CAO shows slightly negative anomalies for most potential *Calanus* predators at depths below 1000 m, but even there, cnidarians, deep-sea amphipods, shrimps and chaetognaths are still occurring. The developmental stage that benefits most from AF wintering at depth seem to be *Calanus* nauplii, as indicated by their positive anomalies between 1000–500 m (Fig. 2c). Near the surface, nauplii likely face higher mortality from omnivorous and carnivorous copepods, such as *Metridia longa* and *Paraeuchaeta* spp.[52,53],

which accumulate here during winter (Fig. 7). Thus, AF that reduce their own predation risk by migrating to depth, also give their offspring a longer time away from key predators while developing and ascending. The success of this strategy will depend on the nauplii having sufficient lipid reserves to fuel the ascent until food becomes available. The AF appeared in slightly higher concentrations in the upper water column but had similar body mass and lipid levels at the end of winter regardless of depth (Fig. 3). Based on the abundance (ind m$^{-3}$) timelines there was no evidence of major winter mortality for the adult and subadult *Calanus* population in our data (Fig. 3).

There are other potential reasons why vertical dispersal throughout the whole water column may confer advantages over a translocation to depth. Broader dispersal can reduce the spread of epi- and endoparasites that *Calanus* carry in the high Arctic[54]. Vertical dispersal might also lead to horizontal dispersal as life stages enter different water masses and thereby enhance the probability that part of the population encounters more favourable conditions elsewhere[41]. Whatever the key advantages may be, the spreading out of a population throughout several vertical kilometres of water is not a trivial task and the next section explores some potential mechanisms.

Physical forcings (e.g. vertical currents, downwelling plume fronts, internal waves) can play an important role for the copepods' vertical distribution, for instance, in the Norwegian Coastal Current[55]. However, in the CAO, there is no down- or upwelling. Stratification and its changes, combined with the vertical temperature distribution, are the only physical processes acting here[56]. Therefore, active swimming is the primary way to reach vertical displacement, and this can be supported or impaired by density differences between the copepods and the surrounding seawater. The stratification of the seawater can change from day to day and may thereby form corridors of reduced density that promote downward swimming or corridors of increased density that aid upward swimming (Fig. 5).

The copepods themselves can also regulate their density, for instance via the ion composition in their haemolymph[57], their wax ester content[58] or their PUFA proportions[59]. However, our study shows clear differences in tissue density despite similar lipid, wax ester and PUFA contents of the copepods, and draws attention to another lipid component — cholesterol. The positive linear correlation between the maximum tissue density and the copepods' cholesterol content (Fig. 6d) requires further consideration. Cholesterol has a higher density than wax esters or water and occurs in high amounts in the swimbladder membrane of deep-sea fish[60]. It significantly alters the phase behaviour of cell membranes when incorporated into the lipid bilayers at high concentrations (typically >25–30 mol%)[61]. Rather than transferring from a 'crystalline' to a 'liquid-disordered' phase under temperature rise, cholesterol promotes the formation of a 'liquid-ordered' (rafted) phase[62,63]. This liquid-ordered phase has several properties that can lead to increased molecular density at higher temperatures: firstly, lipid molecules are packed tightly, secondly, the thermal expansion of the membrane is reduced, and thirdly, water molecules are expelled during formation of the liquid-ordered phase[61,63,64].

This incorporation of cholesterol seems to be especially relevant in the CAO, where warmer Atlantic Water lies below the very cold Polar Surface Water (Fig. 6a). This warmer layer can create a barrier for *Calanus* spp. as the buoyant force on the wax esters changes primarily with temperature, not pressure[34,65]. Thus, copepods that enter the warm Atlantic layer may become more buoyant and either be unable to cross this layer or require a long time to swim against the uplift. However, for those *C. hyperboreus* that were sampled from warmer waters in Aug.–Sept. 2020, we found the tissue density to increase, rather than decrease, at temperatures similar to the Atlantic Water layer. This suggests that *C. hyperboreus* has developed mechanisms to aid its descent through the Atlantic layer, likely due to the incorporation of cholesterol and other density-regulating components. The co-occurring *C. glacialis* did not show strong density increases at higher temperatures (Fig. 6), which might explain why the species is rarely found below 200 m in the CAO (Figs. 2c and S3), but migrates below 200 m in locations where temperatures continuously decrease with depth[66].

It is assumed that seasonally migrating copepods overwinter motionless near their point of neutral buoyancy[58] to reduce metabolic costs and avoid predators. With the in situ water density increasing from 1025 to 1037 kg m$^{-3}$ over 2000 m, small differences in the copepods' density could lead to a winter dispersal over hundreds of metres (see *Calanoides acutus*)[51]. How these differences in density occur remains uncertain. However, in calanoid copepods, cholesterol and PUFA derive mainly from dietary intake[67,68], which suggests that small differences in the feeding history of *C. hyperboreus* may translate into different points of neutral buoyancy and therefore contribute to winter dispersal.

Partial migration, where populations of animals are composed of a mixture of resident and migratory individuals, has increasingly been documented with the technological advances of tracking animal movements and is now accepted as a widespread phenomenon amongst migratory taxa from invertebrates to fish, birds and mammals[69–71]. However, the ultimate mechanisms that determine individual differences in migratory tendency remain controversial. Intraspecific competition over resources is considered a key mechanism, where smaller subordinate individuals are outcompeted by larger ones and start to migrate. This is called the 'competitive release' hypothesis[69,72] and may explain why subadult *Calanus* (CV) were slightly more prominent at depth and adult *Calanus* (AF) at the surface[19] (Fig. 2c, this study). However, other factors might also play a role, such as individual differences in food preferences or vulnerability to predators, and genetic variations in the threshold for a migratory tendency[69].

For partial migration to be maintained over time, the two strategies must yield either equivalent fitness returns (an evolutionary stable state) or the relative benefits of each strategy differ according to circumstances but are overall balanced, known as a conditional strategy[69,73,74]. It is generally predicted that migration confers survival benefits due to the avoidance of predators, harsh weather or starvation, while residency promotes breeding success through better access to resources, such as habitat or food[69,74,75]. Thus, in partially migratory populations, both migration and residency may offer complementary fitness benefits to the population[1,73]. Our observations on *C. hyperboreus* align with this theory: the resident individuals had a higher PUFA content, indicating better access to resources (here high-quality food), while the migratory individuals released their eggs into a habitat with lower predation pressure, leading to higher offspring (nauplii) survival. The joint benefit of the two strategies might come from the wide dispersal of offspring, which enhances the likelihood that some of them will escape predation and find food.

Climate change is a key factor that could perturb the cost/benefit trade-off in migration strategies. A study on 340 migrating bird species across Europe found that species that scatter across wider areas in winter are more resilient to climate change, while those that aggregate in small areas are more vulnerable and likely to decline[76]. Thus, partial migration is a positive predictor of population trends in European birds[76]. However, many species are also changing their migratory behaviour under climate change and human disturbances[2]. In response to milder winters and extended feeding seasons, for example, some populations of white stork and bowhead whale now spend longer times at their summer feeding grounds or even become fully resident[77,78]. In the Arctic, loss of sea ice, Atlantification, increased primary production and changes in sea-ice algae and phytoplankton phenology[79,80] may trigger a changed migratory behaviour in copepods[27,81]. However, the direction of change could be debated; while earlier and/or later algal blooms[48,82,83] may favour increasing residency in *C. hyperboreus*, stronger winter storms that can build with the thinning or loss of sea ice[84] may trigger copepod descent. Our study provides a valuable baseline of *C. hyperboreus* seasonal migration in the remote, ice-covered CAO against which future studies can assess the effect of climate change.

Zooplankton (including copepods) provide the primary pathways of carbon sequestration to the deep ocean, as they contribute to both the gravitational carbon flux via their faecal pellets and the active carbon transport via their vertical migration[85]. On a global average, faecal pellets are responsible for 60% of the total carbon export, physical mixing for 20% and sinking phytoplankton and vertically migrating zooplankton for 10% each,

while in high latitudes and coastal upwelling regions the contribution of the migrant pump almost doubles[85]. The latter reflects the important role of lipid-rich copepod migrant species in those regions. However, such carbon export estimates often assume a rigid migration behaviour that is not always met in the natural environment. Our study shows that about one-third of the subadult and adult *C. hyperboreus* population is not migrating to sequestration depth of 500 m, and similar proportions might be true for other seasonally migrating copepod species.

Interestingly, the proportion of the population that resides in the upper 200 m has not changed since the early ice-drift expeditions in the 1950s (Fig. 2b), which suggests that during MOSAiC, the winter conditions in the CAO were still similar. Key factors relevant for the future are the continued reduction in sea ice[86] and strengthening and warming of the Atlantic Water layer[87]. The former will likely result in a break-up of stratification and the start of winter deep mixing in the CAO[88], while the latter will enhance the inflow of potential *Calanus* predators such as planktivorous myctophids and squids[89]. This could lead to North Atlantic conditions, where the dominant *Calanus* species (here *C. finmarchicus*) often overwinters below the thermocline (650 m), avoiding planktivorous predators, including mesopelagic fish, that occupy intermediate depths (400–600 m)[90]. However, even in the North Atlantic, *C. finmarchicus* shows a large vertical spread during winter with some part of the population remaining in the upper water column or occupying shallow habitats[90]. Thus, *C. finmarchicus* in the North Atlantic shows a similar behavioural asynchrony as we encountered for *C. hyperboreus* in the CAO.

In conclusion, we propose that this large individual variability in overwintering depth is driven by differences in feeding history and the accumulation of density-regulating components (e.g. cholesterol, PUFA, metal ions) in combination with physical forcings (e.g. stratification, temperature, downwelling, upwelling). While some individuals will incur reduced fitness at their overwintering depth due to predation or lack of resources, this bet-hedging strategy promotes long-term population survival in fluctuating environments.

## Materials and methods
### The MOSAiC expedition
The MOSAiC expedition, in 2019–2020, represents the first year-round interdisciplinary study of atmosphere, sea ice, ocean, ecosystem, and biogeochemical processes during the transpolar drift across the CAO, with a unique opportunity for intensive field sampling[91–94]. While the zooplankton was sampled during most of the expedition, this study focusses on stratified net hauls from 2000 m to the surface, that took place in winter (Nov.–Mar. 2019/2020, n = 14) and late summer (Jul.–Sept. 2020, n = 12). During both seasons, RV *Polarstern* drifted in the CAO (≥85°N), except for three sampling events in the Fram Strait in July 2020.

### Hydrographic data
Full-depth temperature and salinity profiles were collected daily, from the ship and/or from the ice, for most of the expedition. We compute the potential density from the temperature and salinity profiles following the TEOS10 standard equation of state[95,96].

### Vertical profiles of zooplankton abundance and community structure
The MultiNet (Hydrobios MultiNet 'Midi'; 150 μm mesh size, mouth area: 0.25 m²) sampled the following five depth intervals: 2000–1000, 1000–500, 500–200, 200–50 and 50–0 m and was equipped with a calibrated electronic flow metre measuring the volume of filtered seawater (m³) for each sample. Immediately after sampling, the catch was preserved in a 4% formaldehyde–seawater solution buffered with hexamethylenetetramine and stored at room temperature until quantitative analysis at the Alfred Wegener Institute (AWI). The samples were divided with a Motoda plankton splitter up to aliquots of 1/256, depending on the number of organisms present. Large and rare taxa, including *C. hyperboreus* AF and CV, were counted from the entire sample. To calculate mesozooplankton

abundances (individuals m⁻³) for the different depth intervals, the counts (n) per subsample and the filtered volume (V; m³) as measured by the flowmeter were used[97]:

$$\text{Abundance} = n * \text{split factor}/V \quad (1)$$

The anomaly from an even distribution of a zooplankton taxon or developmental stage throughout the water column was calculated by subtracting the proportion of filtered water in each of the five depth strata from the proportion of specimens in the same depth strata There are five strata: 2000–1000 m (S1, extent: 1000 m), 1000–500 m (S2, extent: 500 m), 500–200 m (S3, extent 300 m), 200–50 m (S4, extent 150 m) and 50–0 m (S5, extent 50 m). Abundance1 is the abundance in strata 1 (no m⁻²). AbundanceT is the total abundance in 2000 m (no m⁻²). Depth1 is the extent of strata 1 (=1000 m). DepthT is the total sampling depth (=2000 m).

$$\text{Anomaly1}(\% \text{ difference}) = (\text{Abundance1} * 100/\text{AbundanceT}) \\ - (\text{Depth1} * 100/\text{DepthT}) \quad (2)$$

A positive anomaly indicates that animals were more abundant in that stratum than would have been expected from an even distribution throughout the water column.

### Extracting historical data of *Calanus hyperboreus* vertical distribution in the CAO
To compare the vertical distribution of *C. hyperboreus* and *C. glacialis* AF and CV during MOSAiC with observations from previous cruises, we extracted data from a database compiled by Kvile et al. and published online (doi:10.18739/A2KD1QK3Q)[9]. This database includes stage-specific abundances (ind m⁻³) of *C. hyperboreus* and *C. glacialis* from 51 different sources sampled in the Arctic Ocean from 1935 to 2016 ('historical data 1935–2016'). All data are provided with basic background information such as date, location, type of net, mesh size and upper/lower depth of the net. For our purpose, we extracted (1) winter data with at least two sampling depths (one of about 200–0 m depth, and one of about 1000-200 m depth), (2) winter data below 1000 m depth, (3) year-round abundances in the upper 200 m and (3) year-round abundances below 500 m depth.

### Distinguishing between mature and non-mature *Calanus hyperboreus* females
For all formalin-preserved AF from MultiNet samples, we distinguished between mature females carrying dark-brown oocytes that often fill a large part of the prosoma, and non-mature or senescent females, that contained either transparent oocytes (non-mature) or oocytes that were only visible in the ovary (senescent), based on the method developed by Niehoff and Hirche (1996)[98].

### Collection of specimens for biochemical analysis
For biochemical analysis, *C. hyperboreus* and potential predators were sampled weekly from the upper (200–0 m) and deeper ocean (2000–200 m) by vertical tows of a ring-net (mouth area 1 m², mesh size 150 μm) with a mechanical messenger (Hydrobios), and from underneath the ice (0 m and 10 m) by horizontal tows of a net (0.24 m², 150 μm) attached to a remotely operated vehicle (M500, Ocean Modules, Sweden)[99] with average towing speed of 0.26 m s⁻¹ and ~15 min diving time. A ring-net of larger mesh size (1 m², 1000 μm) was deployed from 1000–0 m to collect macrozooplankton. Juvenile polar cod (*Boreogadus saida*) were collected from cracks and holes in the ice using hand-held spoons and sieves[50]. The polar cod were sampled and processed according to and within laws, guidelines, and policies of the German Animal Welfare Organization. No specific permissions were required. The fish collected are neither endangered nor protected in the central Arctic waters and coastal waters of the Svalbard Archipelago. Polar cod were killed immediately after collection.

In August and September 2020, additional zooplankton specimens for biochemical analysis were picked directly from the MultiNet casts, to allow for higher vertical resolution (2000–1000, 1000–500, 500–200, 200–50 and 50–0 m). The catch was sorted in trays placed on crushed ice to avoid a change in temperature. Active specimens were gently transferred into petri dishes and pooled for each biochemical sample depending on their body size, e.g. 10 *C. hyperboreus* AF, 20 *C. hyperboreus* CV, 40 *C. glacialis* AF. Specimens were briefly dipped into ultrapure water to remove saltwater, Digital images were taken by Leica M125 or Wild M5 microscopes, and the samples were transferred into glass vials for storage at -80 °C until further analysis.

## Dry mass and copepod prosome length

All samples for biochemical analysis were freeze-dried at the AWI for 24–48 h and sent to the University of Plymouth, UK for subsequent preparations. Here, each sample was weighed using an analytical balance (Mettler Toledo, XP 504, $d = 0.1$ mg), gently broken up with a spatula and partitioned into subsamples for three purposes: (1) bulk stable isotope analysis of carbon (C) and nitrogen (N), (2) lipid analysis (including total lipid content, FA and FAlc composition, $\delta^{13}$C values of FA and FAlc, sterol composition, HBI concentrations) and (3) density measurements[100].

For *C. hyperboreus* AF and CV collected by MultiNet casts in August and September 2020, the prosome length was measured on the Digital images using the software package ImageJ v.1.53 k[101].

## Bulk stable isotope and carbon-nitrogen analyses

Carbon and nitrogen bulk isotopic compositions were determined to investigate differences in copepod food sources and trophic level between seasons and/or sampling depths. Subsamples of homogenised tissue ($1.25 \pm 0.3$ mg) were transferred into pre-weighed tin capsules and the dry mass was estimated via the mass difference. The tin caps were closed, compacted and sent to the Littoral, Environment and Societies Joint Research Unit stable isotope facility (CNRS — University of La Rochelle, France). There, the samples were analysed with a continuous flow isotope ratio mass spectrometer (Delta V Plus with a Conflo IV interface, Thermo Scientific, Bremen, Germany) interfaced with an elemental analyser (EA Isolink, Thermo Scientific, Milan, Italy). Results are reported in per mil (‰) in the δ notation as deviations from standards: atmospheric $N_2$ for $\delta^{15}$N and Vienna Pee Dee Belemnite for $\delta^{13}$C using the formula:

$$\delta^{13}C \text{ or } \delta^{15}N = [(R_{sample}/R_{standard}) - 1] \times 1000 \quad (3)$$

where R is $^{13}$C/$^{12}$C or $^{15}$N/$^{14}$N, respectively. Normalisation was done using USGS61 and USGS63 (US Geological Survey, Reston, VA, USA) based on their assigned carbon and nitrogen isotope-delta values and standard uncertainties. The uncertainty of the reported isotope-delta values was evaluated as the standard deviation of repeated measurements ($n = 5$) for each reference material (i.e. USGS61 and USGS63) within a single group of analyses. Uncertainty did not exceed 0.10‰ for both $\delta^{13}$C and $\delta^{15}$N values. The trophic level (TL) of *Calanus* species and of their potential predators were calculated using a $\delta^{15}$N trophic fractionation factor of 3.4‰ and the mean $\delta^{15}$N value of the suspended particulate organic matter in Aug.–Sep. 2020 ($\delta^{15}$N: 5.0‰, $n = 48$)[102] as a baseline following the formula by Zanden and Rasmussen[103]:

$$TL = 1 + (\delta^{15}N_{consumer} - \delta^{15}N_{suspended\ particulate\ organic\ matter})/3.4 \quad (4)$$

Calibration for the total carbon and nitrogen determination was done daily with an Acetanilide standard. The carbon and nitrogen contents were expressed as per individual for *C. hyperboreus* and all studied potential predators, except Cnidaria. For the latter, we used an indirect approach via volume estimates from ZooScan images[104] and subsequent carbon conversion for Cnidaria according to Kiørboe[105].

## Lipid analyses

Lipid content, FA and FAlc and their respective carbon isotopic composition, as well as sterol composition, were determined to investigate energy storage, food sources and trophic relationships between *C. hyperboreus* and its potential predators. Three internal standards were added to the dried animal samples: tricosanoic acid methyl ester (23:0) for FA analysis; 9-octyl-8-heptadecene for HBIs ($m/z$ 350.3) and 5α-androstan-3β-ol for sterols (m/z 333). Total lipids were extracted in dichloromethane : methanol (2:1, v:v) for 10 min in a sonication bath. Thereafter, the sample was centrifuged (2500 rpm, 2 min) and the lipid-containing liquid phase was transferred into a new vial. The procedure was repeated twice, and the pooled lipid extract was cleaned with 0.88% potassium chloride solution via centrifugation and removal of the debris-containing layer. The cell-free lipid extract was transferred into a pre-weighed vial, evaporated to dryness under $N_2$-atmosphere and weighed. The lipid content is expressed as percentage of dry mass (% DM). Thereafter, lipids were redissolved in 1 ml of dichloromethane and divided into two 0.5 ml subsamples, one for FA and FAlc analyses at the AWI and one for HBI and sterol analysis at the University of Plymouth.

At the AWI, samples were converted into fatty acid methyl esters (FAME) and wax ester-derived free FAlc by transesterification using a solution of 3% concentrated sulphuric acid in methanol and heating for 4 h at 80 °C[106]. Subsequently, FAME and FAlc were quantified using a gas chromatograph (6890 N, Agilent Technologies, USA) with a DB-FFAP capillary column (60 m, 0.25 mm I.D., 0.25 μm film thickness) supplied with a splitless injector and a flame ionisation detector using temperature programming (160–240 °C). Helium was used as a carrier gas. The detection limit based on the certified reference material (Supelco 37 Component FAME mix, Supelco, Germany) was 10–20 ng per component. Clarity chromatography software system (version 8.8.0, DataApex) was used for chromatogram data evaluation. FA and FAlc are presented in shorthand notation, i.e. $A:B(n-x)$, where: $A$ indicates the number of carbon atoms in the straight FA chain, $B$ represents the number of double bonds, $n$ represents the terminal methyl group and $x$ denotes the position of the first double bond from the terminal end. The portions of individual FA or FAlc are expressed as mass percentages of the TFA or total FAlc content. We calculated the number of ATP molecules that can be gained via beta-oxidation based on the average number of carbon atoms and the average number of double bonds within the TFA or FAlc pool (http://amazingbiotech.in/biochemical-calculator/).

For the compound-specific stable isotope analysis, FAMEs were separated from the wax ester-derived FAlc via column chromatography with silica gel (6%, deactivated). The FAME fraction was eluted with hexane : dichloromethane (9:1, v:v), FAlc with hexane: acetone (1:1, v/v). Carbon isotopic compositions were determined for abundant FA and FALc using a GC-c-IRMS system, equipped with a Trace GC Ultra gas chromatograph, a GC Isolink and Delta V Plus isotope ratio mass spectrometer, connected via a Conflo IV interface (Thermo Scientific Corporation, Germany). The FAMEs, dissolved in hexane, were injected in splitless mode and separated on a DB-FFAP column (60 m, 0.25 mm I.D., 0.25 μm film thickness). The $\delta^{13}$C values of the individual FAMEs were calibrated by analysing the certified standard FAMEs 14:0 (certified: $\delta^{13}$C value −29.98‰, measured: $\delta^{13}$C value −29.54‰) and 18:0 (certified: $\delta^{13}$C value −23.24‰, measured: $\delta^{13}$C value −23.29‰) at regular intervals (~every five samples). The analytical error was ±0.3‰ for both 14:0 and 18:0 (representing 1 standard deviation of 10 analyses each). Furthermore, for quality assurance and analytical precision of the determined carbon stable isotope ratios, the laboratory standard 23:0 was measured intermittently during the sample runs with an analytical error of ±0.4‰ (representing the standard deviation of 10 analyses).

For *C. hyperboreus* AF and CV collected by MultiNet casts in August and September 2020, separated fractions of polar and neutral lipids were additionally analysed for their FA and FAlc. Polar lipids represent mainly membrane components, while neutral lipids are storage lipids (for Arctic

*Calanus* spp., primarily wax esters). The separation of polar and neutral lipids was carried out at AWI using column chromatography on small glass columns (Pasteur pipettes) filled with silica gel 60 (0.040–0.063 mm, Merck). After conditioning with $2 \times 2 - 3$ mL hexane:dichloromethane (50:50, v:v), up to 500 µL of total lipids in dichloromethane were added on top of the column. Subsequently, neutral lipids were collected with 3 mL dichloromethane:methanol (75:25, v:v) and polar lipids were obtained with 3 mL methanol:water (90:10, v:v). Both lipid fractions were evaporated to dryness with a stream of pure nitrogen, redissolved in a small volume of hexane, and stored at -20 °C for further processing at the University of Bremen. Here, FA in the polar and neutral lipid fractions were converted into FAMEs and wax ester-derived free FAlc by transesterification using a solution of 3% concentrated sulphuric acid in methanol and heating for 4 h at 80 °C, in analogue to the method used for TFA[106]. FAME and FAlc were quantified using a gas chromatograph (7890 A, Agilent Technologies, USA) equipped with a DB-FFAP capillary column (30 m, 0.25 mm I.D., 0.25 µm film thickness) running a temperature programme (oven temperature 80–240 °C) with helium as carrier gas. Samples were injected in solvent vent mode by a programmable temperature vaporiser injector detected by flame ionisation and identified by comparing retention times with those from standards of known composition (Supelco 37 Component FAME mix, Supelco, Germany, as well as a copepod mixture established as laboratory standard for FAlc) using the Agilent ChemStation software. The detection limit was 1–2 ng per component.

In addition to the FA and FAlc, two HBIs and five sterols were analysed that are specific to certain marine diatom species or other microalgae, and can provide further information on the feeding history of polar zooplankton[107,108]: IP$_{25}$ (C$_{25}$ monoene; *m/z* 350.3), IPSO$_{25}$ (C$_{25}$ diene; *m/z* 348.3), epi-brassicasterol (24-methylcholesta-5,22E-dien-3β–ol; *m/z* 470), ß-sitosterol (24-ethylcholest-5-en-3β–ol; *m/z* 396), chalinasterol (24-methylcholesta-5,24(28)-dien-3β–ol; *m/z* 470), campesterol (24-methylcholest-5-en-3β–ol; *m/z* 382) and cholesterol (cholest-5-en-3β–ol; *m/z* 458). After saponification with 20% potassium hydroxide in water:methanol (1:9,-v:v, 70 °C; 60 min), HBIs and sterols were extracted with hexane ($3 \times 2$ mL) and purified by open-column chromatography (SiO$_2$) using hexane as solvent for HBIs and subsequently, hexane : methylacetate (4:1,-v:v) as solvent for sterols. Sterol fractions were derivatised with N,O-bis(trimethylsilyl)trifluoroacetamide (BSTFA, 70 °C, 1 h). HBIs and sterols were analysed using a gas chromatograph (Agilent 7890A GC), coupled to a mass selective detector (Agilent 5975 mass spectrometry), fitted with an Agilent HP-5ms column with auto-splitless injection. Identification of individual HBIs and sterols was achieved by comparison of their retention index and mass spectrum with those obtained from purified standards[109]. Quantification of IP$_{25}$, IPSO$_{25}$ and sterols was achieved by integrating individual ion responses in selected-ion monitoring (SIM) mode and normalising these to the corresponding peak area of the internal standard and an instrumental response factor obtained from purified standards[110].

## Density measurements

For *C. hyperboreus* AF, CV and *C. glacialis* AF sampled in August and September 2020, the density of the freeze-dried body tissue was measured with a pycnometer (BELSORP-max, Microtrac MRB). This pycnometer uses helium gas to fill the voids around solid structures for high-precision volume measurements[111]. In the first step, the empty sample cell (Model 010-20002-0-0; 0.5 cm³) was immersed in a thermostatic bath and the space volume of the cell was measured across six temperatures between −2 °C and 3 °C to supply a temperature-specific 'blank' volume. The temperature of the thermostatic bath was maintained via an open circulating bath (Julabo, UK) filled with a water-glycol-based fluid (Thermal G, Julabo, UK). For each temperature, four measurements were taken, each based on 12 individual runs. Then the temperature was increased by 1 °C, with a 70-min pause to allow for stabilisation of the new temperature within the sample cell. These

'blank' measurements were carried out three times, at the beginning, middle and end of the lab working period. The standard deviation for the three blank measurements varied between 0.003 and 0.008% for the six temperatures.

In the second step, dried *Calanus* tissue was added to the sample cell and the space volume of the cell was measured for the same temperatures as the blank. At the end, the sample cell was weighed with and without animal tissue using a high-precision digital balance (Mettler Toledo, XP 504, $d = 0.1$ mg). The density of the tissue was obtained from the quantity of tissue divided by the difference in volumetric capacity between the blank sample cell and the cell with tissue.

The instruments' measuring accuracy can be as high as ±0.02% (BELSORP, Instruction Manual Ver. 1.3.4), but generally drops with low sample mass. To accumulate sufficient mass (usually 15–38 mg), samples had to be compiled based on species, developmental stage, and sampling depth. Between 5 and 18 biochemical subsamples contributed to one sample for density measurements, representing 50 to several hundred specimens. For *C. glacialis*, we had two samples, one with specimens from the upper 100 m, and one from the upper 200 m water column. For *C. hyperboreus* CV, there was one sample from the upper (0-200 m) and one from the deeper ocean (500–2000 m). Due to the larger size of *C. hyperboreus* AF, there were 7 samples available, covering the upper ocean (0–100 m, 0-200 m), the Atlantic layer (200–500 m), the deeper ocean (200–2000 m, 500–1000 m, 1000–2000 m) and a reference station north of Svalbard with stronger influence of Atlantic water (50–500 m).

## Statistics

To interpret the detailed biochemical data from *C. hyperboreus* samples in Aug.–Sept. 2020, we used two different statistic approaches: PCA and box plots with Mann–Whitney *U*-tests in Minitab v. 19. The PCA is based on percentage data for lipid content, wax ester content, FA, FAlc and phytosterols, and δ¹³C and δ¹⁵N values, and illustrates the overall differences in the biochemical composition between *C. hyperboreus* living in different depth strata. The box plots accompanying the PCA show the exact data of the biochemical data compiled for the two 'surface' (200–50 m, 50–0 m) and 'depth' strata (2000–1000 m, 1000–500 m). The box plots show the summary of a data set: the median (central line), lower and upper quartile including the middle 50% of the data (box), minimum and maximum 25% of scores (lower and upper whiskers) and outliers (dots outside the whiskers). To determine whether any aspects of their biochemical composition differ between surface and depth-collected *C. hyperboreus*, the central tendency within the two groups was compared with the non-parametric Mann–Whitney *U*-test. This test can replace the two-sided *t*-test when the sample size is small (<15 observations) and the data are not normally distributed. To calculate the Mann–Whitney *U*-test, the rankings of the individual values are determined and added up for each of the two groups. The *W*-values give the rank sum of the first group. *P* values < 0.05 indicate that the null hypothesis, saying that the central tendency of the two groups (surface and depth) does not differ, must be rejected. We compiled the *W*-values and *p* values for all tested biochemical aspects in Table S2, and in Fig. 4 box plots, significant differences are marked with an asterisk.

## Reporting summary

Further information on research design is available in the Nature Portfolio Reporting Summary linked to this article.

## Data availability

The data that support the findings of this study are openly available in PANGAEA and the UK Polar Data Centre: https://doi.org/10.1594/PANGAEA.959965 (hydrographic data, ships CTD), https://doi.org/10.1594/PANGAEA.959966 (hydrographic data, Ocean City CTD). https://doi.pangaea.de/10.1594/PANGAEA.980472 (zooplankton abundances), https://doi.org/10.5285/e8792e69-c9ae-4d54-a0a0-622005f325ad

(zooplankton trophic marker composition), https://doi.pangaea.de/10.1594/PANGAEA.980518 (stable isotope compositions).

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

## Acknowledgements

'MOSAiC' data presented in this manuscript were produced as part of the international Multidisciplinary drifting Observatory for the Study of the Arctic Climate (MOSAiC) with the tag MOSAiC20192020 and the Project_ID: AWI_PS122_00. We thank all those involved in the expedition of the RV *Polarstern* during MOSAiC in 2019–2020 (AWI_PS122_00) as listed in Nixdorf et al.[112]. Martina Vortkamp, Lena Eggers, Anja Nicolaus, Tim Barnes and Lucy Stephenson are curating the associated MOSAiC samples and metadata. We thank Judith Peters and Richard Broughton for discussing the outcomes of our density measurements on *Calanus* tissue. We enjoyed working with Glynn Gorick on the conceptual model. K.S., A.A. and S.T.B. were funded through the UK Natural Environment Research Council's (NERC) contribution to MOSAiC, the SYM-PEL project (NE/S002502/1). C.H.'s contribution to MOSAiC was funded through the Swedish Research Council (Starting Grant 2018-03859). S.S. and F.L.S. were funded by the Dutch Research Council grant 866.18.003, with contributions of the European Commission (EC), European Climate, Infrastructure and Environment Executive Agency (CINEA), Framework Contract EASME/EMFF/2018/003, Specific Contract EASME/EMFF/2018/1.3.2.2/03/SI2.805469, and of the Netherlands Ministry of Agriculture, Nature and Food Quality (LNV) grant WOT-04-009-047.04. B.L. was supported by a Fellowship at the Hanse-Wissenschaftskolleg Institute for Advanced Study, Delmenhorst, Germany. A.C. was funded by the German Ministry for Education and Research (BMBF; Grant 03F0917A MOSAiC 3). C.J.A., R.G.C., C.E.G., K.M.S. and R.J. were funded by the US National Science Foundation Office of Polar Programs (OPP-1824447 and OPP-1824414). Three anonymous reviewers provided constructive comments that improved the manuscript.

## Author contributions

K.S., M.G., H.F., B.N., W.H., A.A., C.H., B.L., A.A.F., F.L.S. and K.Ø.K. developed the concept of this study. N.H., G.C., R.G.C., C.J.A., S.S., K.M.S., K.S., A.A.F. and C.J.M.H. conducted the field sampling. A.C., N.K., B.N., C.E.G. and R.J. carried out the taxonomic analysis, maturity staging and size measurements. M.G., N.W., M.W., W.H., S.D., K.S. and S.T.B. conducted or supervised the lipid analyses. B.L., G.G. and C.J.M.H. carried out or supervised the bulk stable isotope analysis. K.S., G.L. and K.J. performed or supervised the tissue density analysis. C.H. and S.S. advised on physical oceanography aspects. K.S. carried out the data analysis and initial drafting of the manuscript. All co-authors commented on the manuscript.

## Competing interests

The authors declare no competing interests.
