## [Peer Review file · Communications Earth & Environment]

Seasonal vertical migration of large polar copepods reinterpreted as dispersal mechanism throughout the water column

Corresponding Author: Dr Katrin Schmidt

Version 0:

Decision Letter:

Dear Dr Schmidt,

Your manuscript titled "Seasonal vertical migration of polar copepods is a winter dispersal rather than translocation to depth" has now been seen by 3 reviewers, and we include their comments at the end of this message. They find your work of interest, but some important points are raised. We are interested in the possibility of publishing your study in Communications Earth & Environment, but would like to consider your responses to these concerns and assess a revised manuscript before we make a final decision on publication.

We therefore invite you to revise and resubmit your manuscript, along with a point-by-point response that takes into account the points raised. Please highlight all changes in the manuscript text file.

Please submit your point-by-point responses as a separate file, distinct from your cover letter where you can add responses to the Editors' comments that you do not want to be made available to the reviewers. Word files are preferred. We recommend that any figures, tables or graphs that are included in the response to reviewers are also included in the main article or Supplementary Information.

Please use the following link to submit your revised manuscript, point-by-point response to the referees' comments (which should be in a separate document to any cover letter), a tracked-changes version of the manuscript (as a PDF file) and the completed checklist:

Link Redacted

We hope to receive your revised paper within six weeks; please let us know if you aren't able to submit it within this time so that we can discuss how best to proceed. If we don't hear from you, and the revision process takes significantly longer, we may close your file. In this event, we will still be happy to reconsider your paper at a later date, as long as nothing similar has been accepted for publication at Communications Earth & Environment or published elsewhere in the meantime.

Please do not hesitate to contact us if you have any questions or would like to discuss these revisions further. We look forward to seeing the revised manuscript and thank you for the opportunity to review your work.

Best regards,

Jose Luis Iriarte Machuca, PhD

Editorial Board Member
Communications Earth & Environment

Alienor Lavergne
Senior Editor
Communications Earth & Environment

EDITORIAL POLICIES AND FORMATTING

Editorial Policy: [Policy requirements](https://www.nature.com/documents/nr-editorial-policy-checklist.pdf) (Download the link to your computer as a PDF.)

- Behavioural and social science
- Ecological, evolutionary & environmental sciences
- Life sciences

<https://www.nature.com/documents/nr-reporting-summary.zip>

Furthermore, please align your manuscript with our format requirements, which are summarized on the following checklist: [Communications Earth & Environment formatting checklist](https://www.nature.com/documents/commsj-phys-style-formatting-checklist-article.pdf)

and also in our style and formatting guide [Communications Earth & Environment formatting guide](https://www.nature.com/documents/commsj-phys-style-formatting-guide-accept.pdf) .

*** DATA: Communications Earth & Environment endorses the principles of the Enabling FAIR data project (<http://www.copdess.org/enabling-fair-data-project/>). We ask authors to make the data that support their conclusions available in permanent, publically accessible data repositories. (Please contact the editor if you are unable to make your data available).

All Communications Earth & Environment manuscripts must include a section titled "Data Availability" at the end of the Methods section or main text (if no Methods). More information on this policy, is available at <http://www.nature.com/authors/policies/data/data-availability-statements-data-citations.pdf>.

If a community resource is unavailable, data can be submitted to generalist repositories such as [figshare](https://figshare.com/) or [Dryad Digital Repository](http://datadryad.org/). Please provide a unique identifier for the data (for example a DOI or a permanent URL) in the data availability statement, if possible. If the repository does not provide identifiers, we encourage authors to supply the search terms that will return the data. For data that have been obtained from publically available sources, please provide a URL and the specific data product name in the data availability statement. Data with a DOI should be further cited in the methods reference section.

REVIEWER COMMENTS:

Reviewer #1 (Remarks to the Author):

This manuscript advances a novel understanding of the vertical migratory behavior of *Calanus hyperboreus* in the Arctic. Through multifaceted data streams, the authors argue that this copepod's vertical behaviour are more consistent with a "dispersal" strategy across the pelagic realm than a classic "habitat translocation" (habitat shift).

The manuscript is well-written, and they bring in a diversity of data to support their conclusions. For the species in focus, the results are novel and align with an emerging paradigm that highlights individual variability of zooplankton vertical migrations.

However, the text needs to be streamlined and figures need to be further tuned.

I have made the following observations regarding the text and figures. The authors (hereafter, "you") may use these for improving the clarity of the manuscript further.

Abstract

L35-37: "Partial migration as seen in *C. hyperboreus* has previously been described for other copepod species, but the non-migrants are usually ignored in population models and biogeochemical budgets": Great that you touched this aspect!

Introduction

L84 (repeated in L96): "Several theories have been proposed...": Theories or hypotheses?

L89: This is an abrupt transition from the hypotheses listed above and the MOSAIC program. I enjoyed reading the introduction but felt like hitting a speed-bump in this transition. Just add one or two lines mentioning the weaknesses of the above-mentioned hypotheses (e.g., not being widely tested throughout the spatial domain of *Calanus*, thus leading to considerable variability in observations across time and space; also, less focus on *C. hyperboreus*). This potential research gap can then be used to pitch the suitability of MOSAIC dataset on *C. hyperboreus*.

L96-100: This text is out of place and is a premature exposure of results in the introduction. Consider removing it and keeping the last sentence(s) of the introduction a bit more open (such as the hypotheses you are going to test).

Methods

L436: "splitter up to aliquots of 1/256": Just curious, what were present in such high densities to split samples up to 8 times?

L439-440: Consider adding the formula into a standard equation form (not as inline text).

L441-445: Consider converting this text to an equation.

L452: "...attached to a remotely operated vehicle": Require more information, such as what vehicle was used, towing speed, distance between the vehicle and the net opening, whether the vehicle was propulsion driven (jet or thruster) and potential impact on the propulsion turbulence (turbulent wake) on the sampling.

L457: "warming": Perhaps, use a better term.

L462-463: "...home laboratories": Perhaps, remove the "home" part. It reads a bit awkward.

L477: Cite ImageJ (its always good to cite open-source apps).

L489: Write the equation separately (not as an in-text equation, as it stands right now). Do the same for that in L497-498.

L614-624: Font change – please check!

Results

L104-110: This section seems to be out of place. It is a conceptual model that is derived from your data – and thus it should be placed in support of the discussion. Otherwise, you are presenting a summary of the results prematurely before presenting the underlying data. Start the results chapter at L112 instead.

Subchapter titles: Again, these subtitles are a bit coercive on the reader. What you want in the results chapter is to keep things as objectively as possible and do all the interpretation in the discussion. Change these subchapter topics to something simple, such as "vertical distribution of *Calanus* during the study period" or something...

L114: "net sampling profiles": consider using "vertical profiles" or "vertical net hauls".

Fig. 2: In line 2 of the caption, change the depth strata notation from deeper depth – to – shallower depth (50-0, 200-50 etc.). This is because the net is hauled from bottom-up.

Fig. 2: Panel (b): the bottom row of figures of panel b are somewhat intersecting with the top row x-labels. Adjust this. Also, make sure that the y-labels are right-aligned. They are easier to read when the labels are closer to the y-axis line.

Fig. 2: Panel (c): the grey-color grids in the two figures are cutting through the points. Grid must be drawn below the points – otherwise, it is distracting to read the plot. Also, is the y-axis in log-scale? If so, the axis scaling is wrong. If you want to keep

this y-axis scaling (= equal distance between tickmarks) then, use the corresponding log-transformed abundance values in the tick marks. Or, if you want to keep to this y-axis labels (= keep the original data on a log-transformed y-axis), then put the tickmarks in the correct distance from each other. For example, distance between 0.001 and 0.1 is not similar to that between 1 and 10.

L122: Remove cross reference to Figure 1 – add a pointer to actual data, not the conceptual model at this stage! Do this change elsewhere (e.g., L 133).

L127: “data base”: I noted that this term pops out here and in Figure 2. I have not encountered any description of a database in the Methods chapter, apart from mentioning that data from previous years were used. Also, use “database” not “data base”.

L143, 145, 148: reverse the depth bin notation (e.g., 1000-500 m)

Figure 3: In panel A, can you add the proportional/percentage variance explained to the PC1 and PC2 axes labels?

Figure 3: The vertical axis label of panel D is cutting into the panel ID. Adjust this! See a similar intersection in Figure 4 (Y-label of panel “I” is cutting into the right figure margin of “H”)

L199-201: Move to discussion.

Discussion

L405-408: “With the current knowledge, we are unable to predict how seasonal vertical migration of *C. hyperboreus* will change throughout a warming Arctic. However, continued strengthening and warming of the Atlantic Water layer (Wang et al. 2020), might contribute to reduced deep seasonal migrations”: Is this not a bit contradictory given that you mention in L410-411 that there should be more visual predation pressure on *Calanus*? What we see in the north Atlantic waters (e.g., in the Norwegian Sea) is that *C. finmarchicus* overwinters below the permanent thermocline at around 650 m or below. Such deeper migration (translocation – as termed in your case) likely makes them avoid planktivorous predators, including mesopelagic fish that occupy intermediate depths (400-600 m). Shouldn't this partly be what we will be seeing in the CAO with the multifaceted climate change effects?

General: One thing I would like to see in the discussion a bit more is the role of physical forcing in vertical distribution of these copepods. How do water currents, stratification, watermass exchange, up/downwelling contribute to autumn/winter dispersal of copepods? Also, to what extent that patchiness and sampling bias therein could have impacted your conclusions? I am not saying they were, but, build a case that such biases were not significant.

Overall: Very nice points mentioned in the discussion. The case you build on partial migration highlights the individual variability in the migration behaviour, that ultimately leads to a behavioural asynchrony.

Reviewer #2 (Remarks to the Author):

This manuscript describes year-round vertical distribution of *Calanus hyperboreus* in the central Arctic Ocean and examines the various mechanisms that may be responsible for this distribution. The authors found that *C. hyperboreus* does not perform a mass vertical migration but instead disperse throughout the water column during the winter, possibly due to differences in density and/or diet. The authors hypothesise that this is bet-hedging strategy allowing them to avoid predation while maximising resources. The paper is well written, the results are relevant and novel, the conclusions are supported by data, and it undoubtedly deserves publication. I do, however, have some major and minor comments regarding the presentation of the data. Once these are addressed, I believe the manuscript will be an important contribution in the field of winter polar marine ecology.

My main major criticism is the choice of the authors to present their data in non-chronological order - summer followed by winter/spring, when in reality they sampled vice versa. This choice hinges on the assumption that their summer/winter samples from a single season are representative of any summer/ any winter, anywhere in the Arctic Ocean (since they also did not sample in the same area, but along a drift trajectory), and that the seasonal transitions will be identical year-to-year. I do not believe that this is a fair assumption to make, and think that the data should be shown and interpreted with the caveat that it is the results of one year of sampling, in very specific locations. With this I am by no means belittling their effort - seasonal data from the central Arctic is extremely rare, and this was a massive undertaking. But I think showing any kind of

“time series” in a scientific studies needs to respect the actual timeline when the data was collected, and this includes showing the gaps where they exist (for example, in this case April-June).

I am also questioning the authors' choice to exclude data between April-June, especially since this data exists. Even though they are focusing on summer vs. winter, this could clearly show the transition between the two states. In March the copepods were dispersed, but in July they were already concentrated in the surface layer - what happened in between? Same with body composition/lipid content - in Figure 4 you can see it dropping during march and then it goes back up again by august.

There is one other question I find intriguing, and I did not see discussed in the manuscript. We see *C. hyperboreus* in deep, oceanic waters - not just in the central Arctic but also, for example, in the deep north Atlantic (e.g., Norwegian Sea) or Greenland Sea. The common belief is that they require deep waters in order to overwinter, unlike, for example, *C. glacialis*, which is why we typically do not see them in shallower waters. How do your results line up with their broader geographic distribution patterns, if it turns out that they can overwinter in the shallows as well?

The other minor comments are listed below.

Title - the title is a bit misleading, since you are discussing only one species, not all polar copepods

Abstract

Line 29 - biomass dominant

Line 30 - this is your main result, and I would expand on it a bit more, with some more concrete numbers and dates.(e.g., “between Nov-Mar, the copepods were evenly distributed throughout the water column, with 20% remaining in the upper 200m” - since throughout the manuscript you are looking at animals above/below 200m it makes more sense to mention here rather than 500m

Line 31 - this is another very important result, and again, I would write something more concrete about what you found regarding the density differences

Line 33 - remove ‘female’s’ and change to ‘female specimens’ or ‘adult females’

Introduction

Lines 96-100 - This belongs in the results, not introduction. Here, you should write your study goals and/or hypotheses.

Results

It was not clear to me whether this section is supposed to be a combined results/discussion section, or a results section followed by a separate discussion section. At the moment, it is somewhere in between, and needs to be re-worked in both cases. In the first case, the discussion (page 10 onwards) needs to be integrated into the appropriate results sub-sections. In the latter case, the current parts of the section that are not direct results of this study (e.g., comparison with the work of Kvile, 2019) need to be moved there.

On your Figure 2, your September points are wildly scattered across the study area, presumably because the ship stopped drifting and began steaming southwards. Were these stations showing the same patterns in all their characteristics? Is it fair to pool them (as I assume you have done in all the figures and analyses)?

Lines 104-108. All but the last sentence in this paragraph belong in the figure caption, not the main text, and not as ingress to the results. Instead, refer to Figure 1 as needed throughout the text, and perhaps summarise these findings again in discussion or conclusions/summary.

Line 119 - “start re-aggregating near the surface in March” Where do you show this? On your supplementary figure S1 your March samples do not look that much different from the winter ones. This also contradicts your decision to pool Nov-Mar in all you analyses and plots.

Line 120 - “across 14 winter sampling events” - were they all the same? There was no seasonal progression/variability between dates/locations?

Line 122 - this is either discussion, or, if it is part of your results, you need to describe this data and statistical analysis in the methods

Line 131 - remove comma after Both

Line 192 - wouldn't a regression analysis be more appropriate here, since you are comparing 2 numeric variables (temperature vs density)?

Line 193 - could this be explained spatially? You mentioned the one station north of Svalbard, but where do all the others come from?

Lines 170-171:

Discussion:

Line 280 - how big of an issue are visual predators in the CAO under multi-year ice in the polar night?

Line 297 - How did you determine this?

Line 335 - is this unique for this species? Are there any other examples in animals where tissue density increases with temperature?

Methods:

Lines 424 - As mentioned above, I think this is a misleading way to present your data.

Lines 449-450 - how did you take stratified net tows using the ring net? Did you have some kind of closing mechanism?

Figures:

In all figures, please re-work captions for clarity. Make them succinct, without extra information that belongs in results or methods. For example, in Fig. 5 do not include "no or minor changes" etc. - the reader will see that by looking at the figure, that belongs in the results text. You also do not need to write how you prepared the tissue samples - that belongs in methods.

On figure 4, instead of explaining a-f/g-i in the caption first collectively and then individually, you can make sub-headers on the figure itself ("Body characteristics", "Abundance"), and in the caption just define each panel a single time.

You also need to decipher all used acronyms in every figure caption (AF, FA, PUFA, CAO).

Figure 1:

This is a beautiful graphic, but it needs to be reworked a bit.

First, please re-plot in chronological order - you sampled Nov-March before July.

Second, in addition to your copepods densities also include numbers (%) for each layer - for example, in Nov-Mar, was it 30 or 50 or 79% of the population below 1000m? At the moment it is really impossible for me to say.

Third, the circles on the right are confusing, and in my opinion, do not add much to the figure. Can you make separate vertical panels (can be a little offset from the rest) showing main food sources and predators in the upper vs. lower ocean? The "offspring" circle is the same, as far as I can tell, and is already shown with the eggs/nauplii panel.

Figure 2:

Please make an insert showing your study area on a broader geographic scale. Please add place names (islands, land masses, ocean basins and ridges).

Do the colours mean months? That is not clear. Please also add years to the months, or indicate the direction of the drift, so it is apparent that Nov/Dec were sampled before Jun/July. Describe also the stations (Aug/Sep) that are not on the drift line

Figure 3:

Please re-order the panels so they are in chronological order. I think the panel eggs/nauplii can also be moved from supplementary into the main text.

Figure 4:

Since here you are showing an actual time series, this figure is particularly misleading, as your sampling points are not in chronological order. Please re-plot (including gaps for the missing months between them)

Even if they are chronological, it is generally not good practice to connect points with lines, since that implies a linear transition from one point to the next, and you do not actually know the dynamics between your two points, and especially since you are also moving through space and not only time. This is particularly apparent in your abundance plots which jump wildly from date-to-date. Consider plotting this figure with paired bar plots instead of points/lines.

Figure 5:

5b - In the caption it says *C. glacialis*? I thought we were talking about *C. hyperboreus*?

Could 5b be shown as a regression instead of an ANOVA? You can use a mixed model and include your samples as a random factor.

5c-d. This is confusing to me. I would collapse these two panels into one, instead of artificially separating them into groups - there are not so many groups that the reader can't distinguish between them. Re-color in a colour scheme that logically goes from shallowest to deepest.

Reviewer #3 (Remarks to the Author):

I have written an evaluation report that is attached below.

In brief, the manuscript reports valuable results on vertical distribution of the dominant copepod *Calanus hyperboreus* and other zooplankton in the CAO, along with data on lipid chemistry and trophic markers. The observations on *C. hyperboreus* are interpreted to reflect partial migration where one part of the population remains in the upper water layer in winter. The notion is one of winter dispersal rather than translocation to depth, as stated in the title. A weakness of the manuscript is that it reflects poorly the many previous studies of *Calanus hyperboreus* and other zooplankton in the CAO, including from earlier US and Soviet ice drift stations. These, and other studies in subarctic waters, have demonstrated migration into deep water, with dispersal over a wide range of depth, which in the CAO includes the upper water layer under the ice in mid-winter. In my view, the issue is not translocation to depth or dispersal, but rather translocation and dispersal. The question for the CAO is why the dispersal includes the upper layer.

There is a strong emphasis on the generality of partial migration including for birds and mammals. I would have liked to see more focus and emphasis on the specific conditions of the CAO habitat for *Calanus hyperboreus*, building from the knowledge gained by previous studies such as Dawson (1978) and Rudyakov (1983).

Communications Earth & Environment is committed to improving transparency in authorship. As part of our efforts in this direction, we are now requesting that all authors identified as 'corresponding author' create and link their Open Researcher and Contributor Identifier (ORCID) with their account on the Manuscript Tracking System prior to acceptance. ORCID helps the scientific community achieve unambiguous attribution of all scholarly contributions. You can create and link your ORCID from the home page of the Manuscript Tracking System by clicking on 'Modify my Springer Nature account' and following the instructions in the link below. Please also inform all co-authors that they can add their ORCIDs to their accounts and that they must do so prior to acceptance.

Version 1:

Decision Letter:

Dear Dr Schmidt,

Your revised manuscript titled "Seasonal vertical migration of polar copepods is a winter dispersal rather than translocation to depth" has now been seen by the previous 3 reviewers, and we include their comments at the end of this message. They find your work of continued interest, but still some important points are raised. We are interested in the possibility of publishing your study in Communications Earth & Environment, but would like to consider your responses to these concerns and assess a revised manuscript before we make a final decision on publication.

In addition, we invite you to consider the following editorial thresholds:

- 1) acknowledge and discuss the possible role of translocation in seasonal polar copepod migration,
- 2) refine the discussion to be specific to *Calanus* and the Central Arctic Ocean, and
- 3) fully justify your approach towards measuring density.

We therefore invite you to revise and resubmit your manuscript, along with a point-by-point response that takes into account the points raised. Please highlight all changes in the manuscript text file.

Please submit your point-by-point responses as a separate file, distinct from your cover letter where you can add responses to the Editors' comments that you do not want to be made available to the reviewers. Word files are preferred. We recommend that any figures, tables or graphs that are included in the response to reviewers are also included in the main article or Supplementary Information.

Please use the following link to submit your revised manuscript, point-by-point response to the referees' comments (which should be in a separate document to any cover letter), a tracked-changes version of the manuscript (as a PDF file) and the completed checklist:

Link Redacted

We hope to receive your revised paper within six weeks; please let us know if you aren't able to submit it within this time so that we can discuss how best to proceed. If we don't hear from you, and the revision process takes significantly longer, we may close your file. In this event, we will still be happy to reconsider your paper at a later date, as long as nothing similar has been accepted for publication at Communications Earth & Environment or published elsewhere in the meantime.

Please do not hesitate to contact us if you have any questions or would like to discuss these revisions further. We look forward to seeing the revised manuscript and thank you for the opportunity to review your work.

Best regards,

Jose Luis Iriarte, PhD
Editorial Board Member
Communications Earth & Environment

Alice Drinkwater, PhD
Associate Editor
Communications Earth & Environment
Consulting Editor
Communications Sustainability

EDITORIAL POLICIES AND FORMATTING

Editorial Policy: [Policy requirements](https://www.nature.com/documents/nr-editorial-policy-checklist.pdf) (Download the link to your computer as a PDF.)

- Behavioural and social science
- Ecological, evolutionary & environmental sciences
- Life sciences

<https://www.nature.com/documents/nr-reporting-summary.zip>

Furthermore, please align your manuscript with our format requirements, which are summarized on the following checklist: [Communications Earth & Environment formatting checklist](https://www.nature.com/documents/commsj-phys-style-formatting-checklist-article.pdf)

and also in our style and formatting guide [Communications Earth & Environment formatting guide](https://www.nature.com/documents/commsj-phys-style-formatting-guide-accept.pdf) .

*** DATA: Communications Earth & Environment endorses the principles of the Enabling FAIR data project (<http://www.copdess.org/enabling-fair-data-project/>). We ask authors to make the data that support their conclusions available in permanent, publically accessible data repositories. (Please contact the editor if you are unable to make your data available).

All Communications Earth & Environment manuscripts must include a section titled "Data Availability" at the end of the Methods section or main text (if no Methods). More information on this policy, is available at <http://www.nature.com/authors/policies/data/data-availability-statements-data-citations.pdf>.

If a community resource is unavailable, data can be submitted to generalist repositories such as [figshare](https://figshare.com/) or [Dryad Digital Repository](http://datadryad.org/). Please provide a unique identifier for the data (for example a DOI or a permanent URL) in the data availability statement, if possible. If the repository does not provide identifiers, we encourage authors to supply the search terms that will return the data. For data that have been obtained from publically available sources, please provide a URL and the specific data product name in the data availability statement. Data with a DOI should be further cited in the methods reference section.

REVIEWER COMMENTS:

Reviewer #1 (Remarks to the Author):

The authors have addressed the concerns I raised on the initial submission (R0) and the manuscript reads a lot better as a result. The authors have incorporated most of the reviewer comments (from all reviewers) and made fair justifications where some comments/suggestions could not be addressed. I appreciate the authors' efforts in reviewing the manuscript, and I do not have any further major comments to the manuscript at this stage.

Reviewer #2 (Remarks to the Author):

The manuscript has increased substantially since the previous version, and the authors have done an admirable job addressing many issues raised by myself and the other two reviewers. I particularly appreciated the incorporation of the historical data from the drifting ice stations and SHIBA. Nonetheless, I still have a few remaining comments that I believe are rather important and should be addressed prior to publication.

1.) The title. I still believe that the authors are doing both themselves and potential readers a disservice by naming the paper "vertical migration of polar copepods" when in reality their data focuses mainly on *Calanus hyperboreus* in the CAO, even if other copepods and polar regions are briefly mentioned in a literature context. Perhaps consider changing to "a key Arctic copepod" or at the very least "large herbivorous copepod", since nobody is arguing that common, abundance-dominant species like *Oithona* or *Oncaea* are seasonally translocating to depth. Alternatively, consider something like "Vertical distribution patterns of key *Calanus* in the CAO support an active winter dispersal strategy in large herbivorous polar copepods"

2.) Some key literature is still missing, as was rightfully brought up by Reviewer #3. For example, the early and more recent works of Kosobokova; Kvile et al. 2018; as well as some important studies from the Greenland Sea. There are not so many publications about *C. hyperboreus* in the Arctic that they all can't be at least mentioned -- if you are claiming to redefine established paradigms, first you need to establish the state of the field.

3.) The distinction between Results vs Discussion (and even Introduction!) is still very blurred - some contents that belongs in Discussion is still in the Results section, and vice versa. This should be addressed.

More specific comments:

Figure 1. I am not sure that this figure belongs in the introduction, since it is not simply "setting the stage" but an actual meta-analysis of existing literature. I would put this figure last, and reference it in the discussion instead. In the caption, say clearly in the first line "based on literature values"

Line 118 onwards: Start the results with presenting your own data first, then followed by literature compilation.

Line 193 onwards (section 3.4). This is very interesting information, but most of this section as currently written belongs in the discussion. Here, you just describe the characteristics of the physical environment that you think are relevant.

Line 265. "Norwegian Sea is at the fringe of its habitat" Many would argue that the deep Arctic Ocean is also at the "fringe of its habitat" (e.g., Kvile et al., 2018, <https://doi.org/10.1111/gcb.14419>) If you think otherwise, you need to justify your opinion.

Line 276. "Within their core habitat". Related to the previous point, what is the core habitat for *C. hyperboreus*? How do you define "core habitat"?

Line 337. See also this recent paper that supports this hypothesis: <https://doi.org/10.3354/meps14665>

Line 356. "We found a positive linear correlation". This phrasing should be in results. Here, you say something like "The positive trend between cholesterol and tissue density suggests..."

Fig 3. I get the appeal of this figure, but statistically speaking, it is not correct. Using simple linear regression for this data violates the most important assumption of SLR: independence. In your case, the non-independence is both spatial and temporal - you are not sampling in the same location, but drifting through space. To correct for this, you would need to introduce both a temporal autocorrelation correction (e.g., CAR1) and somehow incorporate the spatial variability into your model. Additionally, this analysis is not described anywhere in the methods section.

An alternative would be to just show this data as bar plots with whiskers for each month (similar to your Figure 1, with red and blue side by side as deep and shallow), and leave statistics out of it - that will still make the trends you observed quite visible.

Figure 6. D. is this trend line for the averages or for the raw data? Instead of showing the standard errors, just show the actual data points (keeping your colors and symbols)

Reviewer #3 (Remarks to the Author):

Review of manuscript – 01 April 2025

Seasonal vertical migration of polar copepods is a winter dispersal rather than translocation to depth

Authors: Katrin Schmidt and co-workers.

Submitted to journal: Communications Earth & Environment

General

This is the second time I see this manuscript. I provided comments to a first version with my review dated 17 January 2025. This is now a revised version of the manuscript. I raised four major points in my first review, and regrettably, I find that only one of them has been adequately addressed.

This point was a poor reflection of previous literature and knowledge. The authors have now included more information from previous studies. In the Discussion, they now refer to the study by Østvedt (1955) on *Calanus hyperboreus* collected at weather station M in the Norwegian Sea. They rightfully point out that this location is at the southern fringe of the habitat for the Arctic *Calanus hyperboreus*. I don't have the publication by Østvedt at hand, but the species does reproduce in warmer conditions as seen in this study from a deep fjord just outside Bergen: Matthews, J. B. L., Hestad, L. & Bakke, J. L. W. 1978: Ecological studies in Korsfjorden, western Norway. The generations and stocks of *Calanus hyperboreus* and *C. finmarchicus* in 1971-1974. *Oceanologica Acta* 1. 277-284.

The authors have also now included results from Soviet ice drift stations in 1950-1956 (Rudakov 1983) and the SHEBA ice drift in 1997-98 (Ashjian et al. 2003). Very usefully, results from these previous studies are now shown along with the new results from MOSAiC in Fig. 2b. The new results are broadly similar to the old results, which "suggests that during MOSAiC the winter conditions in the CAO were still similar" (to the conditions during the 1950s) (lines 447-449).

The concept of partial migration

A second point I raised in my first review was on the notion of dispersal rather than translocation. In my view, the previous literature has shown migration (or translocation) to depth AND dispersal. This can be understood in an ecological sense as migrating into the dark deep to avoid visual predators and spread out with dispersion to reduce the likelihood of being found by non-visual invertebrate predators.

The present authors make a contrast between these two aspects as reflected in the title – winter dispersal rather than translocation to depth. I take issue with this. The data from MOSAiC (as well as earlier results) show a translocation to depth as the difference between the vertical distributions in summer and winter, as shown in Fig. 2 b and c. I understand the authors as suggesting that this is merely a dispersal phenomenon and not a migration, basically a concept of migration OR dispersal.

We don't know (at least not well) what the mechanisms of migration are, nor what the mechanisms for dispersal are. They could both reflect active swimming and/or changes in buoyancy. I have colleagues (Kjell Arne Mork and others) who work with profiling Argo floats equipped with video that records copepods in the Norwegian Sea. They document a rapid descent of *Calanus finmarchicus* from the surface layer to below 500 m depth in a few weeks' time around mid-summer. This suggests active migration, and it is possible that the visual records may provide information on the behavior of these descending copepods. As far as I know, this remains to be investigated.

The data on *Calanus hyperboreus* from MOSAiC in the CAO suggests a descent of adult females and stage V copepodites in late summer and autumn at this high latitude, in August-September suggested by Fig. 2b. The winter dispersal is not that easy to see, but the anomaly plots in Fig. 2c suggest a slight increase in concentration of *C. hyperboreus* between late summer/autumn (August-September) and winter (November-March). This would have been easier to see if data were shown with absolute and not relative units. Also, August-September is the time of the seasonal descent from the surface layer (as shown in Fig. 2b), and the contrast between the profiles for late summer/autumn and winter are therefore not ideal to document the seasonal descent and dispersal at depth, not the least because of the very coarse vertical resolution of the sampling. Apparently, Polarstern left the CAO in summer to return to resume sampling in August. Therefore, the dispersal of *Calanus hyperboreus* is not well documented in this study.

The interpretation that a fraction of the population of adult females and CVs remain stationary as a resident component in the surface layer during winter is a dubious claim not well supported by the data. The trendlines in Fig. 2b suggest that there is an increasing proportion in the upper 200 m between autumn and mid-winter. It is therefore possible that the two processes of descent (or translocation or migration to depth) and dispersal are countering each other. As the polar night is coming, the dispersal may include the upper layer under the sea ice, perhaps reflecting the lack of a light cue for the copepods to keep at depth as they would do at lower latitudes where there is light in winter. I have some evidence for *Calanus* species in the central Barents Sea to support this interpretation (published in the grey literature). It is of interest to note that the results from SHEBA suggest a decline in the proportion of both adult females and CVs of *C. hyperboreus* between mid-winter and late winter. SHEBA drifted at lower latitudes in the Amerasian Basin where winter darkness would be of shorter duration compared to MOSAiC.

In my view, it is not warranted to suggest partial migration for the population of *Calanus hyperboreus* in the CAO based on the MOSAiC results. The data is consistent with a concept of dislocation to depth AND dispersion, which in the high-latitude location with heavy sea ice in the CAO, includes dispersion all the way to the surface layer. It is in a way illogical to suggest the copepods are not translocating to depth but simply dispersing (which can only be effective in one direction – downwards) and then claim that a component of the population is not migrating but remains stationary in the upper layer. Is it implied that these copepods are not dispersing and thus have a different behavior from the rest of the population which does disperse? The lengthy discussion of bet-hedging and the ecological advantages of partial migration with reference to latitudinal migrations of birds and mammals, are not particularly relevant in the context of this paper.

Habitat and life cycle of *Calanus hyperboreus*

My third main comment in the first review was that there was too much emphasis on generalities and too little on specifics of the CAO. This relates to the previous point on partial migration. The paper could have focused more on the specific results on the distribution, reproduction and biochemistry of *Calanus hyperboreus*, and it could be shortened by removing the more speculative and not well substantiated parts.

There is no mentioning of the multi-year life cycle of *Calanus hyperboreus* in the CAO, which would be relevant in relation to the discussion of trade-offs between finding food and avoiding predators. *C. hyperboreus* may require up to 4 years to mature, and the species appear to be only partially successful in reproducing in the lower latitude portions of the CAO, and to be partly an expatriate in the high latitude central part of the CAO. The CAO is therefore not core habitat for *Calanus hyperboreus* (as claimed on page 11, line 276). On the contrary, this is on the fringe of its distribution, and the species is only marginally successful in the CAO. The core areas are the Greenland Sea and Baffin Bay.

The abundance of *Calanus hyperboreus* in the high latitude part of the CAO is typically 500-1000 copepodites m⁻² (Skjoldal, H. R. (Ed.). 2022. Ecosystem assessment of the Central Arctic Ocean: Description of the ecosystem. ICES Cooperative Research Reports Vol. 355. 341 pp. <https://doi.org/10.17895/ices.pub.20191787>). The abundances from the MOSAiC study fall generally in this range (Fig. 2c).

The presence of eggs and nauplii in winter suggests that some of the adult females were reproducing. I believe it is very likely that these were eggs and nauplii of *Calanus hyperboreus*, although it was unfortunate that they were not identified (e.g. from size measurements of the eggs). The abundance of eggs was ~3000 m⁻², compared to ~500 adult females m⁻² (Fig. 2c). This gives an apparent egg production of 6 eggs per female, which is relatively low. This could reflect the low proportion of mature females, which appears to be ~10 % from comparison of Fig. 3 a and b. This suggests that only a small fraction of the adult females was actively reproducing at this high latitude location in the CAO. By comparison, Dawson (1978) found that 20-30 % of the females were mature in late winter in his sampling from the T-3 ice drift station.

Adult females generally produce eggs from stored body reserves, without the need to feed. This is what allows them to spawn at depth in winter. There is some increase in the number of mature females in the upper layer during winter (Fig. 3b), which is implied to reflect winter feeding, supported by a trend in trophic markers in Fig. 3h. Winter feeding by *Calanus hyperboreus* is surprising to me. There is a reference to stomach content analysis based on personal communication from one of the coauthors. It would be useful if this evidence was provided, although I understand it is being prepared for publication in another paper. The evidence from the trophic markers is indirect. Also, without food, there would be a process of maturation driven by internal reserves. Therefore, an increase in proportion of mature females (still at a low level) needs not imply winter feeding.

A key point related to Fig. 3b is the low proportion of adult females that mature. This suggests unsuccessful reproduction by *Calanus hyperboreus* at this high latitude location, in line with an interpretation that the species is largely an expatriate in this part of the CAO. This is an aspect that could have been included in the discussion.

Density measurements

The fourth point I raised in my first review was on the validity and ecological relevance of the measured tissue density.

Density was measured on freeze-dried and crushed copepods (gently broken up with a spatula and partitioned into subsamples) with a pycnometer using helium gas to fill the voids in the freeze-dried and homogenized samples. There is just one general reference to Webb (2001), and no information that this is a valid method for determining density of zooplankton. With freeze-drying, water is removed from the copepods and there will be spaces in the dried material. These spaces are affected by shrinking in the drying process, and by the crushing and homogenization of the samples. I don't see how the original shape of the copepods (or crushed parts of the copepods) is inflated with helium gas back to the original volume that corresponded to the weight of the subsamples used for measurements of density. The results from these measurements of density look strange, characterized by very large variation from 0.6 to 1.7 g cm³ (Fig. 6c).

I am hesitant to accept these results before it is demonstrated that the method is valid for the purpose of investigating buoyancy of copepods and other zooplankton. Am I missing something here?

Some other issues

I have difficulty understanding Fig. 8, the conceptual model. What are the small individuals in the upper 200 m in July? Are they meant to be young copepodite stages (and were any such observed?), or are they CV and adult females shown with smaller symbols? Does the increase in number of symbols below 500 m between July and August-September illustrate the seasonal downward migration? Does the scheme overall suggest successful reproduction, and is there an implied multi-year life cycle?

In section 3.5 on vertical distribution of potential predators, there is a consideration on predation risk which I find too simplistic. On lines 254-256, it is stated that only the deepest stratum below 1000 m is potentially a zone of lower predation risk for *Calanus*. However, predation risk is not only about the presence of potential predators but also about their density. There is evidence to suggest that predation is low overall, which allows *Calanus hyperboreus* to exist in relatively high abundance as, at least partially, an expatriate with a multiyear life span.

Communications Earth & Environment is committed to improving transparency in authorship. As part of our efforts in this direction, we are now requesting that all authors identified as 'corresponding author' create and link their Open Researcher and Contributor Identifier (ORCID) with their account on the Manuscript Tracking System prior to acceptance. ORCID helps the scientific community achieve unambiguous attribution of all scholarly contributions. You can create and link your ORCID from the home page of the Manuscript Tracking System by clicking on 'Modify my Springer Nature account' and following the instructions in the link below. Please also inform all co-authors that they can add their ORCIDs to their accounts and that they must do so prior to acceptance.

Version 2:

Decision Letter:

Dear Dr Schmidt,

Your manuscript titled "Reinterpreting seasonal vertical migration of large polar copepods as dispersal mechanism throughout the water column" has now been seen by the editorial team. We are delighted to say that we are happy, in principle, to publish a suitably revised version in Communications Earth & Environment.

We therefore invite you to revise your paper one last time to edit your manuscript to comply with our format requirements and to maximise the accessibility and therefore the impact of your work.

EDITORIAL REQUESTS:

*****Please take care to match our formatting and policy requirements. We will check revised manuscript and return manuscripts that do not comply. Such requests will lead to delays. *****

SUBMISSION INFORMATION:

OPEN ACCESS:

Communications Earth & Environment is a fully open access journal. Articles are made freely accessible on publication. For further information about article processing charges, open access funding, and advice and support from Nature Research, please visit <https://www.nature.com/commsenv/open-access>

Link Redacted

Best regards,

Alice Drinkwater, PhD
Associate Editor
Communications Earth & Environment
Consulting Editor
Communications Sustainability

Review of manuscript

Seasonal vertical migration of polar copepods is a winter dispersal rather than translocation to depth

Authors: Katrin Schmidt and co-workers.

Submitted to journal: Communications Earth & Environment

General evaluation

The manuscript describes results on the large calanoid copepod *Calanus hyperboreus* obtained during the MOSAiC expedition with RV Polarstern, which was frozen into and drifted with the pack ice of the Central Arctic Ocean (CAO) during the winter 2019/2020 (September 2019–October 2020). The MOSAiC expedition was unique in the breadth and sophistication of multidisciplinary observations across atmosphere, ocean and biology. It is to be noted that it was not the first of its kind as it followed a long tradition of winter and year-round observations from drifting ice camps by US and Soviet scientists from before and in the decades after WWII.

The manuscript provides useful observations of the dominant zooplankton species *Calanus hyperboreus* in the CAO, which add to our knowledge and are certainly of broad interest to polar ecologists. This concerns especially the observations on vertical distribution and lipid chemistry of copepodite stage CV and adult females. However, I find some difficulties in the way the results are presented and interpreted. This relates to:

1. The notion of dispersal rather than translocation.
2. Scant reflection of previous literature and knowledge.
3. Too much emphasis on generality and too little on CAO specifics.
4. The validity and ecological relevance of the measured tissue density.

Before I go into these four points in more detail, I will draw up a brief sketch of the species and its habitat. *Calanus hyperboreus* is distributed in the Arctic Mediterranean Sea (which is the Nordic Seas plus the CAO, openly connected through the Fram Strait) and the subarctic gyres of the NW Atlantic. Baffin Bay and the Greenland Sea are the two core areas for the species, both areas characterized by seasonal sea ice and a lack of visual-feeding planktivorous fish.

Calanus hyperboreus can have a one-year life cycle in the ice-free parts of the distribution area but requires more than one year in ice-covered waters at high latitudes. Thus, in the CAO, it may require up to 4 years to complete its life cycle. *Calanus hyperboreus* can reproduce successfully in the southern part of the CAO where there are more light and less heavy ice conditions, but it may at least partly be an expatriate in the central part of the CAO.

The water masses in the CAO are mostly of Atlantic origin. The Atlantic water flows into the CAO via the two prongs of the Atlantic gateway (the Fram Strait and across the Barents Sea

shelf) and circulates cyclonic (counterclockwise) around the basins of the CAO with a residence time of <10 to ~50 years. In this large-scale circulation, *Calanus hyperboreus* goes through its life cycle and performs its seasonal vertical migration. Due to the sluggish circulation and long residence time of water, and the multiyear and multigenerational development of *Calanus hyperboreus*, we should expect a heterogenous history of its population in the CAO.

With this as a general backdrop of information, I revert to the main issues I raised earlier.

1. Dispersal versus translocation to depth

The notion I have from reading literature is that *Calanus hyperboreus* has a seasonal migration to overwinter at depth, where it disperses over a wide water column. See e.g. Østvedt (1955) who described an annual cycle of *Calanus hyperboreus* (and *C. finmarchicus*) at Weather station 'M' in the Norwegian Sea. Conover (1988) described it as an interzonal migrant like the *Neocalanus* species in the subarctic gyres of the North Pacific. This pattern of large-scale downward migration with dispersal at depth has been documented even more extensively for *Calanus finmarchicus* (see Melle et al. 2014 for a review).

So, for me, these large calanoid copepods perform an extensive vertical migration with translocation to depth and dispersal. Putting these two things as alternatives (translocation or dispersal) is to me a wrong emphasis and interpretation. Therefore, already the title of the paper is problematic. In the present case with *Calanus hyperboreus* in the CAO, the issue is more why the dispersal extends all the way from the mesopelagic layer up through the upper layer to the surface. This is what should be discussed, and to me an obvious factor is the strong winter darkness at high latitudes (close to the North Pole) and under thick ice.

2. Previous literature and knowledge

There have been many studies of zooplankton and their vertical distribution in the CAO. There are no references, for example to the extensive work by Ksenia Kosobokova, both her early and later work. Kosobokova worked with zooplankton samples collected from Soviet ice-drift stations, e.g., Kosobokova (1978) described the diurnal variation in vertical distribution of *Calanus hyperboreus* and *C. glacialis* in the CAO. Later work included detailed studies and descriptions of community composition and vertical distribution of zooplankton in the CAO, in collaboration with Hans-Jürgen Hirche, Russ Hopcroft, and others – e.g., Kosobokova and Hopcroft 2010, and Kosobokova, Hirche and Hopcroft 2011. Reference to these and other papers would have provided a much better backdrop of what is already known about vertical distribution and migration of *Calanus hyperboreus* and other species of zooplankton in the CAO. A report prepared by the ICES/PICES/PAME joint working group on integrated ecosystem assessment of the CAO (WGICA) provides a summary and review of both older and newer literature on zooplankton (Skjoldal, H. R. (Ed.). 2022. Ecosystem assessment of the Central Arctic Ocean: Description of the ecosystem. ICES Cooperative Research Reports Vol. 355. 341 pp. <https://doi.org/10.17895/ices.pub.20191787>).

Specifically, Dawson (1978) and Rudyakov (1983) described seasonal vertical distribution of *Calanus hyperboreus* under the ice in the CAO, including sampling through the winter period. There is no reference to Rudyakov, but there is reference to Geynrikh et al. (1983) where Rudyakov along with Kosobokova were coauthors. Dawson (1978) and Geynrikh et al. (1983) are referenced in the manuscript but in a wrong context: “Previous studies at lower latitudes likewise found *C. hyperboreus* CV and AF near the surface in winter (Dawson 1978, Geinrikh et al. 1983, Kvile et al. 2019), ...”. This is wrong. Dawson (1978) reported results from nearly two years of sampling (February 1970–January 1972) from the US T-3 Ice Island in the northern part of the Canada Basin at 84–85°N. Rudyakov (1983) reported results on *C. hyperboreus* sampled over several years from the Russian North Pole ice-drift stations (SP-2–5, 1950–1956) in the Canada and Makarov basins. This can hardly be said to be lower latitudes but comparable to MOSAiC.

3. Generalities versus CAO specifics

The manuscript generalizes the pattern of partial seasonal migration, interpreted for *Calanus hyperboreus*, to the wider cases of geographical (latitudinal) migrations by birds and mammals. While this might be an interesting comparison and analogy in a general review, it seems to me to be somewhat far-fetched in the present case of discussing the specific observations of seasonal vertical distribution of *Calanus hyperboreus* in high latitude parts of the CAO.

I was expecting to see a gain of knowledge on vertical distribution and migration, building from the observations of Dawson (1978) and Rudyakov (1983). Dawson observed a complex pattern for adult females, with many individuals found in the upper layer (0–100 m) under the ice during mid-winter (December–February), and then descending to depths of 200–300 m during spring. My own thinking has been that this is related to the transition from the darkness of mid-winter, when there is no light cue to keep the dispersed copepods away from the upper layer, to a response to the strong increase in light around the time of spring equinox. One reason why this matter is the presence of visual predators (polar cod *Boreogadus saida* and Arctic cod *Arctogadus glacialis*) up under the ice in winter, described by Andryashev et al. (1979).

The way the data is presented is mainly to show the contrast between vertical distribution (and associated lipid chemistry and trophic markers) between the winter (November–March) and summer (August–September) situations. The data are averaged for these two time periods, and the bi-weekly resolution that lies in the data is not used. There might be a reason for this, but it would have been of interest to see if there were some patterns in the upper part of the water column under the ice, as was observed by Dawson (1978).

4. Tissue density

The density of freeze-dried and homogenized bodies of *Calanus hyperboreus* was determined by a pycnometer using helium gas to fill the voids around solid structures in the material. There are no references to applications, and it is not clear if this method has been used with

zooplankton previously, and what relevance tissue density determined in this way has in relation to *in situ* buoyancy. Before a method like this is used in the context of migration mechanisms, it should be validated against other methods for determining density.

5. Some other issues

The vertical distribution of *Calanus hyperboreus* is shown with relative numbers as anomalies in Fig. 1b and in Fig. S1. I understand that this is to show deviations from evenly dispersed distributions. However, the results would have been more directly comparable to other studies if they were given as numbers per m³, at least for the vertical profiles for each sampling occasion in Fig. S1. Integrated numbers (per m²) are given as average values on the panels in Fig. 1 (which is useful) but are not included on the panels of Fig. S1. Average values per m² for each depth interval are given in Table S1 for the summer and winter periods.

Anomalies are used for the vertical distribution of potential predators in Fig. 6. I would have preferred real numbers rather than relative numbers also here. Integrated numbers per m² are given for both abundance and biomass on each panel, which is useful. I note that the biomass values are very high for two of the taxa, ~4 g C m⁻² for *Paraeuchaeta* and ~6 g C m⁻² for chaetognaths. These values would be high anywhere and especially in the low productive CAO. I suspect they are wrongly high due to too high values used for average individual weights to convert numbers to biomass.

In consideration of potential predators, I find that two taxa are missing but which should have been mentioned for completeness. This is the Arctic cod *Arctogadus glacialis*, which is possibly a pelagic species in the CAO, and the group of siphonophores which are known to be present with several species in the mesopelagic layer of the CAO (e.g. Raskoff et al. 2010).

Specific comments

- Line 69 – Pond (2012) is not in the reference list. Pond et al. 2012 is.
- Lines 89-93 – There are two problems with this sentence. The first part ('first to sample zooplankton in highest Arctic latitudes') may give a misleading impression, while the second part ('a unique opportunity to understand the seasonal vertical migration of the biomass-dominant copepod *Calanus hyperboreus*') is not followed through with the results as they are presented.
- Line 112 - Use of 'Arctic' here is somewhat ambiguous. Does it refer to *Calanus hyperboreus* as an Arctic species, or to *C. hyperboreus* in the high Arctic part of its distribution?
- Line 115 – This may not be quite accurate since it appears that some of the September samples were collected from the Nansen Basin, south of the Gakkel Ridge – Fig. 1a. The samples in August and September seem to be collected away from the drift route of the ship. I assume the Polarstern was mobile at this time, but this is not described in the manuscript.
- Line 118 – 'descend' (not 'descent')

- Line 129 – ‘while the congener *Calanus* species *C. glacialis* reproduced in summer.’ How was this observed – egg production, presence of young copepodites?
- Line 172 – I believe the start of the sentence should be reversed, like this: ‘In late summer compared to winter,’
- Line 181 – Fig. 4g-i. Strictly speaking, this should be Fig. 4h, i, since the sentence is about eggs and nauplii.
- Line 181 – ‘*C. hyperboreus* reproduced between Nov.-May’. Is this a general statement, or was *Calanus hyperboreus* observed to reproduce? If the latter, how was reproduction observed – egg production, presence of males, early copepodites?
- Line 181 - It is a weak point that eggs and nauplii were not identified to species. However, I find the assumption to be robust.
- Line 264 – ‘prioritize food uptake over predator avoidance’. I assume from the context that this refers to the winter situation. Did the females which were collected near the surface feed, e.g. did they have stomach content?
- Lines 287-288 – ‘still abundant’. Abundant is here a relative term. The abundances are overall low compared e.g. to subarctic seas.
- Lines 288-289 – ‘The developmental stage that benefits most from AF wintering at depth seem to be *Calanus* nauplii..’ This should be rephrased. *Calanus hyperboreus* are known to spawn at depth during winter, so this observation is in line with what has been previously reported.
- Lines 369-372 –It is uncertain whether observations in Aug-Sept can be taken to reflect resident vs migratory components. The individuals in the upper layer in August-September may migrate to deep water (and disperse) later in the season and thus be part of the migratory component. There is some, but not very pronounced, difference in NL-PUFA (Fig. 3f) but not in PL-PUFA which occur in higher quantities (Fig. 3e).
- Line 386 – Berge (not Berges)
- Lines 407-408 – Note that there is deep seasonal migration through the warmer Atlantic water in the Norwegian Sea (Østvedt 1955), and overwintering depth is deeper there than in the colder waters of the Greenland Sea (Hirche).
- Line 417 – ‘the first year-round interdisciplinary study of atmosphere, sea ice, ocean, ecosystem, and biogeochemical processes during the transpolar drift across the Central Arctic Ocean (CAO)’. I am not sure this is true. There should be reference to the previous extensive ice-drift stations, and to expeditions such as the Canadian/US SHEBA.
- Line 458 – ‘Live’ instead of ‘lively’
- Line 471 – ‘gently broken up with a spatula’. How well does this homogenize the sample material? Has this been studied?
- Line 489 – $\delta^{13}\text{C}$ – C should not be superscript.
- Line 494 – should be USGS63
- Line 619 – ‘living at in’. Delete ‘at’.
- Fig. 4 – There is information on lipid composition and stable isotopes for POM from the Chl maximum layer. It is not clear what this POM represents. Does it make sense to speak about a Chl maximum layer in winter?

- Fig. 5 – there is reference to one station north of Svalbard. There are several stations N of Svalbard in Fig. 1a. Could the one used here be identified?
- Fig. 6 – the line with TL values should be explained in the legend.
- Fig. 6 – what are the numbers on the panel for Cnidaria – abundance or biomass?
- Table S1 – What are the errors - SD or SE?

Reply to the reviewers' comments:

SUMMARY:

We would like to thank the three reviewers for their very constructive comments that triggered a lot of further thinking and helped to improve the manuscript substantially. In response to their comments, we have added two new figures.

The first (Fig. 1) shows that in polar regions all the major seasonally migrating copepod species retain part of their population in the upper 200 m during winter (addressing comments by Rev. 2 and Rev. 3). The second (Fig. 5) illustrates some physical constraints on the copepods' seasonal vertical migration during MOSAiC (Rev. 1). We have also largely extended the inclusion of previous work in the CAO (Rev. 2, Rev. 3), by adding a new Panel in Fig. 2 comparing results from MOSAiC with those from previous ice drift expeditions in the CAO (Severny Polyus 1950-1956 and SHEBA 1997/1998) and discussing the work by Dawson (1978). In the revised version, we also discuss the *Calanus hyperboreus* vertical distribution in the Norwegian Sea (Rev. 2), based on the historical data by Østvedt (1955), and the *C. finmarchicus* vertical distribution in the North Atlantic (Rev. 1), based on the data by Heath et al. (2000).

Another major focus of our revision was to gain a mechanistic understanding of the tissue density increase that we saw for some *Calanus* samples at higher temperatures (Rev. 2 and Rev. 3). A literature survey and discussions with an expert (Judith Peters) draw attention to the role of cholesterol. The revised manuscript now includes a panel on the relationship between the maximum tissue density of *Calanus* and their cholesterol content (Fig. 6). We also supply evidence for winter feeding (Rev. 3) based on the trophic markers IP₂₅ and IPSO₂₅ (Fig. 3h) and a personal communication by K. Shoemaker who analysed gut DNA of MOSAiC winter specimens. Finally, all abundance data of *C. hyperboreus* and their potential predators are now included in the Supplements (Rev. 3).

The only request we were unable to follow, is the detailed inclusion of summer data, neither in Fig. 3 (Rev. 2) nor in the *Introduction* and *Discussion* (Rev. 3). Between mid-Mar 2020 and end-July 2020, no deep nets (below 1000m) were sampled during MOSAiC, so we have no summer data from the deeper ocean to show here. Therefore, the *Introduction* and *Discussion* of this manuscript focusses on the winter months and the autumn descent. The suggestion by Rev. 3 to remove the discussion on other seasonal migrants such as birds and mammals was not followed, as such cross-considerations did benefit the understanding of our data within a wider ecological and biogeochemical context. We had positive feedback on the pitch of our *Introduction* and *Discussion* (i.e. looking beyond the CAO), by Rev. 1 and many of our co-authors. Overall, we aimed to balance our revision equally across the three reviews in terms of time allocation and the incorporation of new text and data. The journal allows an overall manuscript length of 5000 words, a limit that we have now slightly exceeded.

REVIEWER COMMENTS:

Reviewer #1 (Remarks to the Author):

This manuscript advances a novel understanding of the vertical migratory behavior of *Calanus hyperboreus* in the Arctic. Through multifaceted data streams, the authors argue that this copepod's vertical behaviour are more consistent with a "dispersal" strategy across the pelagic realm than a classic "habitat translocation" (habitat shift).

The manuscript is well-written, and they bring in a diversity of data to support their conclusions. For the species in focus, the results are novel and align with an emerging paradigm that highlights individual variability of zooplankton vertical migrations.

However, the text needs to be streamlined and figures need to be further tuned.

I have made the following observations regarding the text and figures. The authors (hereafter, "you") may use these for improving the clarity of the manuscript further.

Abstract

L35-37: "Partial migration as seen in *C. hyperboreus* has previously been described for other copepod species, but the non-migrants are usually ignored in population models and biogeochemical budgets": Great that you touched this aspect!

Thank you!

Introduction

L84 (repeated in L96): "Several theories have been proposed...": Theories or hypotheses?

We changed to 'hypotheses'. Thank you!

L89: This is an abrupt transition from the hypotheses listed above and the MOSAIC program. I enjoyed reading the introduction but felt like hitting a speed-bump in this transition. Just add one or two lines mentioning the weaknesses of the above-mentioned hypotheses (e.g., not being widely tested throughout the spatial domain of *Calanus*, thus leading to considerable variability in observations across time and space; also, less focus on *C. hyperboreus*). This potential research gap can then be used to pitch the suitability of MOSAIC dataset on *C. hyperboreus*.

We followed your advice and finish the *Introduction* by, first, mentioning some research gaps, then introducing the potential of the MOSAIC expedition to close those gaps and finally explaining the three major aspects that we will investigate to progress the topic (historical data, potential drivers, and associated population recruitment).

L96-100: This text is out of place and is a premature exposure of results in the introduction. Consider removing it and keeping the last sentence(s) of the introduction a bit more open (such as the hypotheses you are going to test).

We removed this sentence.

Methods

L436: "splitter up to aliquots of 1/256": Just curious, what were present in such high densities to split samples up to 8 times?

It was small copepods like *Oithona*, copepod eggs and nauplii.

L439-440: Consider adding the formula into a standard equation form (not as inline text).

We changed this. Thank you!

L441-445: Consider converting this text to an equation.

We changed this. Thank you!

L452: "...attached to a remotely operated vehicle": Require more information, such as what vehicle was used, towing speed, distance between the vehicle and the net opening, whether the vehicle was propulsion driven (jet or thruster) and potential impact on the propulsion turbulence (turbulent wake) on the sampling.

Please note, the animals from the ROV tows were used for biochemical analysis not for abundance estimates (the heading of the section is: 'Collection of specimens for biochemical analysis'). However, we now included the make of the vehicle, towing speed and duration, and a reference paper for further details on L520/521.

L457: "warming": Perhaps, use a better term.

We changed to '...to avoid a change in temperature'. Now on L532

L462-463: "...home laboratories": Perhaps, remove the "home" part. It reads a bit awkward.

We finish the sentence '...until further analysis'. L536

L477: Cite ImageJ (its always good to cite open-source apps).

We cited Abramoff et al. (2004). L550

L489: Write the equation separately (not as an in-text equation, as it stands right now). Do the same for that in L497-498.

We changed this. Thank you!

L614-624: Font change – please check!

Thanks! This has been adjusted. L691

Results

L104-110: This section seems to be out of place. It is a conceptual model that is derived from your data – and thus it should be placed in support of the *discussion*. Otherwise, you are presenting a summary of the results prematurely before presenting the underlying data. Start the results chapter at L112 instead.

We have followed this advice. The text has been removed and the Figure is now a conceptual model and part of the *Discussion* (Fig. 8).

Subchapter titles: Again, these subtitles are a bit coercive on the reader. What you want in the results chapter is to keep things as objectively as possible and do all the interpretation in the discussion. Change these subchapter topics to something simple, such as "vertical distribution of Calanus during the study period" or something...

We agree. All subtitles have been changed to simple headings that do not include data interpretation.

L114: “net sampling profiles”: consider using “vertical profiles” or “vertical net hauls”.

We changed to ‘vertical profiles’.

Fig. 2: In line 2 of the caption, change the depth strata notation from deeper depth – to – shallower depth (50-0, 200-50 etc.). This is because the net is hauled from bottom-up.

We agree, and changed accordingly throughout the manuscript.

Fig. 2: Panel (b): the bottom row of figures of panel b are somewhat intersecting with the top row x-labels. Adjust this. Also, make sure that the y-labels are right-aligned. They are easier to read when the labels are closer to the y-axis line.

We improved the layout of the figures.

Fig. 2: Panel (c): the grey-color grids in the two figures are cutting through the points. Grid must be drawn below the points – otherwise, it is distracting to read the plot.

We removed the grid lines.

Also, is the y-axis in log-scale? If so, the axis scaling is wrong. If you want to keep this y-axis scaling (= equal distance between tickmarks) then, use the corresponding log-transformed abundance values in the tick marks. Or, if you want to keep to this y-axis labels (= keep the original data on a log-transformed y-axis), then put the tickmarks in the correct distance from each other. For example, distance between 0.001 and 0.1 is not similar to that between 1 and 10.

We now present the abundances as LOG_{10} values. These panels are now in the Supplements (Fig. S2).

L122: Remove cross reference to Figure 1 – add a pointer to actual data, not the conceptual model at this stage! Do this change elsewhere (e.g., L 133).

Yes, we removed those cross references to the conceptual model.

L127: “data base”: I noted that this term pops out here and in Figure 2. I have not encountered any description of a database in the Methods chapter, apart from mentioning that data from previous years were used. Also, use “database” not “data base”.

We now have a dedicated part of the Material and Methods that explains the use of the ‘database’. It’s under 2.3.2. ‘Extracting historical data of *Calanus hyperboreus* vertical distribution in the CAO’ starting on L498. Instead of ‘database’ we call it now ‘historical data 1935-2016’, to make it clearer.

L143, 145, 148: reverse the depth bin notation (e.g., 1000-500 m)

Done. Thank you!

Figure 3: In panel A, can you add the proportional/percentage variance explained to the PC1 and PC2 axes labels?

Yes, we added 43.6% to PC1 and 17.6% to PC2.

Figure 3: The vertical axis label of panel D is cutting into the panel ID. Adjust this! See a similar intersection in Figure 4 (Y-label of panel “I” is cutting into the right figure margin of “H”)

We improved the layout of the figures.

L199-201: Move to discussion.

Thanks, this sentence has been removed.

Discussion

L405-408: “With the current knowledge, we are unable to predict how seasonal vertical migration of *C. hyperboreus* will change throughout a warming Arctic. However, continued strengthening and warming of the Atlantic Water layer (Wang et al. 2020), might contribute to reduced deep seasonal migrations”: Is this not a bit contradictory given that you mention in L410-411 that there should be more visual predation pressure on *Calanus*? What we see in the north Atlantic waters (e.g., in the Norwegian Sea) is that *C. finmarchicus* overwinters below the permanent thermocline at around 650 m or below. Such deeper migration (translocation – as termed in your case) likely makes them avoid planktivorous predators, including mesopelagic fish that occupy intermediate depths (400-600 m). Shouldn't this partly be what we will be seeing in the CAO with the multifaceted climate change effects?

We used this information for the end of our *Discussion*, writing:

‘Interestingly, the proportion of the population that resides in the upper 200 m has hardly changed since the early ice drift expeditions in the 1950s (Fig. 2b), which suggests that during MOSAiC the winter conditions in the CAO were still similar. Key factors relevant for the future are the continued reduction in sea ice (Heuzé and Jahn 2024) and strengthening and warming of the Atlantic Water layer (Wang et al. 2020). The former will likely result in a break-up of stratification and the start of winter deep mixing in the CAO (Heuzé and Liu 2024), while the latter will enhance the inflow of potential *Calanus* predators such as planktivorous myctophids and squid (Snoeijs-Leijonmalm et al. 2022). This could lead to North Atlantic conditions, where the dominant *Calanus* species (here *C. finmarchicus*) often overwinters below the thermocline (650 m) avoiding planktivorous predators, including mesopelagic fish, that occupy intermediate depths (400-600 m) (Heath et al. 2000). However, even in the North Atlantic, *C. finmarchicus* shows a large vertical spread during winter with some part of the population remaining in the upper water column and occupying shallow habitats (Heath et al. 2000). Thus, *C. finmarchicus* in the North Atlantic shows a similar behavioural asynchrony as we encountered for *C. hyperboreus* in the CAO.’

We hope that this reflects your suggestion.

General: One thing I would like to see in the discussion a bit more is the role of physical forcing in vertical distribution of these copepods. How do water currents, stratification, watermass exchange, up/downwelling contribute to autumn/winter dispersal of copepods?

Also, to what extent that patchiness and sampling bias therein could have impacted your conclusions? I am not saying they were, but, build a case that such biases were not significant.

We addressed this advice by adding a figure to the manuscript that shows the daily stratification and day-to-day stratification change during the MOSAiC expedition (Fig. 5) and explain the two panels in the Results (L191-L200).

In the *Discussion* under '(2) Mechanisms that enable vertical dispersal of copepods', we mention the role of physical forcing (e.g. vertical currents, downwelling plume fronts, internal waves) – however those do not occur in the ice-covered CAO (L342). Nevertheless, changes in the stratification as presented in Fig. 5 can indeed support or hinder upward and downward swimming of the copepods. Especially at the end of summer, we see 'blue' corridors of reclining day-to-day stratification that indicate a reduction in seawater density, that would make downward migration easier.

Overall: Very nice points mentioned in the discussion. The case you build on partial migration highlights the individual variability in the migration behaviour, that ultimately leads to a behavioural asynchrony.

Thank you!

Reviewer #2 (Remarks to the Author):

This manuscript describes year-round vertical distribution of *Calanus hyperboreus* in the central Arctic Ocean and examines the various mechanisms that may be responsible for this distribution. The authors found that *C. hyperboreus* does not perform a mass vertical migration but instead disperse throughout the water column during the winter, possibly due to differences in density and/or diet. The authors hypothesise that this is bet-hedging strategy allowing them to avoid predation while maximising resources. The paper is well written, the results are relevant and novel, the conclusions are supported by data, and it undoubtedly deserves publication. I do, however, have some major and minor comments regarding the presentation of the data. Once these are addressed, I believe the manuscript will be an important contribution in the field of winter polar marine ecology.

Thank you!

My main major criticism is the choice of the authors to present their data in non-chronological order - summer followed by winter/spring, when in reality they sampled vice versa.

We agree with the reviewer. We now present the winter data first (Results '3.2. Changes in abundance and body conditions of *Calanus hyperboreus* over the winter months'). The data are presented in the new Fig. 3 and include only the winter sampling events (Nov. 2019-Mar 2020). All individual data points are presented separately, and we test if there is a trend over time (i.e. from early winter to early spring). The late summer data are presented subsequently in Fig. 4 (Result '3.3. Feeding history and lipid stores of resident and migrating *C. hyperboreus* in late summer'). Here we focus on vertical differences in a range of trophic biomarkers.

This choice hinges on the assumption that their summer/winter samples from a single season are representative of any summer/ any winter, anywhere in the Arctic Ocean (since

they also did not sample in the same area, but along a drift trajectory), and that the seasonal transitions will be identical year-to-year. I do not believe that this is a fair assumption to make, and think that the data should be shown and interpreted with the caveat that it is the results of one year of sampling, in very specific locations.

We now compare the results from the MOSAiC expedition with those from previous ice drift expeditions in the same area (Severny Polyus 1950-1956) and near the Beaufort Sea (SHEBA 1997/1998), that show very similar results regarding the seasonal cycle of *C. hyperboreus* AF and CV found in the upper 200m water column (Fig. 2b), when taking into account some minor differences in net sampling depths. Also, when just focussing on summer-winter differences, the inclusion of other polar species confirms that a variable, but often substantial proportions of individuals remain in the upper water column during winter, i.e. the results found here in the CAO match previous findings on other species elsewhere (new Fig. 1). What we find is a partial migration and that seems not just to be a finding for *C.h.* in the CAO, but seems to be true across species and regions (Fig. 1), as long as these polar species are not outside their core habitat and avoid the surface waters due to higher temperatures. We discuss the Norwegian Sea example based on the data by Østvedt that Rev. 3 mentioned.

With this I am by no means belittling their effort - seasonal data from the central Arctic is extremely rare, and this was a massive undertaking. But I think showing any kind of "time series" in a scientific studies needs to respect the actual timeline when the data was collected, and this includes showing the gaps where they exist (for example, in this case April-June).

Unfortunately, we have no deep net samples (neither for taxonomy nor biomarker) from mid-Mar to end-July. We have therefore no data to cover this part of the year in a consistent manner alongside the winter data. Therefore, this study focusses on the winter months and the time of seasonal descent in late summer (Aug-Sep).

I am also questioning the authors' choice to exclude data between April-June, especially since this data exists. Even though they are focusing on summer vs. winter, this could clearly show the transition between the two states. In March the copepods were dispersed, but in July they were already concentrated in the surface layer - what happened in between? Same with body composition/lipid content - in Figure 4 you can see it dropping during march and then it goes back up again by august.

The aim of this study is to compare animals sampled at depth vs those at the surface. For the very few summer sampling events between mid-Mar and end-July, we have only data from the upper 200 m (biomarker) or surface and intermediate depth (no deep casts beyond 800-1000 m). We feel that this study would not benefit from presenting those very patchy summer data (but for completeness, the data are included in the datasets submitted to PANGAEA and the Polar Data Centre).

There is one other question I find intriguing, and I did not see discussed in the manuscript. We see *C. hyperboreus* in deep, oceanic waters - not just in the central Arctic but also, for example, in the deep north Atlantic (e.g., Norwegian Sea) or Greenland Sea. The common belief is that they require deep waters in order to overwinter, unlike, for example, *C. glacialis*, which is why we typically do not see them in shallower waters. How do your results line up with their broader geographic distribution patterns, if it turns out that they can overwinter in the shallows as well?

We think that the Norwegian Sea is not a core habitat for C.h., but adults or subadults get advected there and might even produce eggs, but the offspring does not recruit to the next generation of adults. This has also been suggested by Rudyakov (1983) when discussing the data from Østvedt (1955). We see the same in Antarctic krill at South Georgia – where they can't recruit, nevertheless you can find very high abundances due to advection and even egg production, but no significant numbers of offspring that survive into later stages. It can be the higher temperatures that hinder optimal development (interestingly krill also 'sit' at depth at South Georgia during summer). Getting back to the Norwegian Sea, the paper by Østvedt (1955) did not mention *C. glacialis*, which suggests to us that C.h. gets there with deeper, colder water (that C.g. does not inhabit) and just survives on their reserves. That C.g. does not or only rarely occurs, confirms that it must be the fringe of where C.h. can live.

Title - the title is a bit misleading, since you are discussing only one species, not all polar copepods

We now included a figure showing other major polar copepods *C. acutus*, *C. propinquus*, *Rhincalanus gigas*, *C. simillimus*, *C. finmarchicus* and *C. glacialis* (Fig. 1), and included *C. glacialis* data throughout the manuscript (e.g. Fig. 1, Fig. 2c, Fig. 6, Fig. 7a, Fig. S2). In the *Discussion* we again mention *C. finmarchicus* and *C. glacialis*, alongside *C. hyperboreus*. Thus, other polar copepod species take now a significant part of the manuscript.

Abstract

Line 29 - biomass dominant

We added 'biomass'. Thank you.

Line 30 - this is your main result, and I would expand on it a bit more, with some more concrete numbers and dates. (e.g., "between Nov-Mar, the copepods were evenly distributed throughout the water column, with 20% remaining in the upper 200m" - since throughout the manuscript you are looking at animals above/below 200m it makes more sense to mention here rather than 500m'

We changed the text as the reviewer suggested. L30/31

Line 31 - this is another very important result, and again, I would write something more concrete about what you found regarding the density differences

We agree and wrote: 'The vertical position of the copepods seems to align with differences in their lipid composition (e.g. cholesterol) that can enhance their tissue density via phase transition at certain temperatures.'

Line 33 - remove 'female's' and change to 'female specimens' or 'adult females'

We now wrote 'adult females'.

Introduction

Lines 96-100 - This belongs in the results, not introduction. Here, you should write your study goals and/or hypotheses.

We agree and changed the text accordingly, mentioning our approach and research aim.

Results

It was not clear to me whether this section is supposed to be a combined results/discussion section, or a results section followed by a separate discussion section. At the moment, it is somewhere in between, and needs to be re-worked in both cases. In the first case, the discussion (page 10 onwards) needs to be integrated into the appropriate results sub-sections. In the latter case, the current parts of the section that are not direct results of this study (e.g., comparison with the work of Kvile, 2019) need to be moved there.

In this manuscript, we have a '*Results*' and a '*Discussion*' part. However, we consider the comparison between MOSAiC data and historical data as part of the '*Results*', as we are extracting the original data from a database and present them in a new way. For instance, here (Fig. 2b) we calculate the part of the population that remains in the upper 200m water column. Such percentages are not presented in the original papers by Rudyakov (1983) and Ashjian et al. (2003). Also, when comparing the absolute abundances of C.h. and C.g. during MOSAiC with the historical data (Fig. S2), we first extract the data from the Kvile et al. data base and plot them together with the MOSAiC data for two relevant water depths (above 200 m and below 500 m). Such data are not presented in Kvile et al. (2019), but are included in the authors' data base. Therefore, referring to Kvile et al. (2019) in the *Discussion* would not be possible, because the reader would not find such data in the paper by Kvile et al. (2019). Please note, we now call the database 'historical data 1935-2016', which we feel is a better term. The process of extracting the historical data is now described in the '*Material and Methods*': '2.3.3 Extracting historical data of *Calanus hyperboreus* vertical distribution in the CAO' L498

On your Figure 2, your September points are wildly scattered across the study area, presumably because the ship stopped drifting and began steaming southwards. Were these stations showing the same patterns in all their characteristics? Is it fair to pool them (as I assume you have done in all the figures and analyses)?

In Fig. 2, which is now Fig. S2, we presented the abundance (ind m⁻³) in the upper 200 m, which for MOSAiC comes from the 50-0 m net (higher values) and the 200-50 m net (lower values). In other figures, those data are again presented separate (Fig. 2c) or summed (Fig. 2b) or weighted (Fig. 3), depending on what is presented in the Figure – 'percentage in upper 200 m' or 'ind m⁻³ in upper 200 m' or 'any abundances found in the upper 200 m' (Fig. S2).

Lines 104-108. All but the last sentence in this paragraph belong in the figure caption, not the main text, and not as ingress to the results. Instead, refer to Figure 1 as needed throughout the text, and perhaps summarise these findings again in discussion or conclusions/summary.

We agree. Rev. 1 did suggest presenting this figure as a conceptual model in the *Discussion*, which we are happy to follow.

Line 119 - "start re-aggregating near the surface in March" Where do you show this? On your

supplementary figure S1 your March samples do not look that much different from the winter ones. This also contradicts your decision to pool Nov-Mar in all your analyses and plots.

We agree. Because the mid-March value is the last in this time series, we can't say for sure if the specimens started to re-aggregate, or if this was just within the natural winter variability. We therefore removed the sentence.

Line 120 - "across 14 winter sampling events" - were they all the same? There was no seasonal progression/variability between dates/locations?

We now added the standard deviation to each percentage value mentioned in the text and show the individual data in the new Fig. S2. As mentioned before, with our data, set a seasonal progression cannot unambiguously be determined. 'Across the fourteen winter sampling events, $\sim 20 \pm 11\%$ of the subadult/adult population remained in the upper 200 m water column, $\sim 11 \pm 7\%$ resided at 500-200 m, $27 \pm 9\%$ at 1000-500 m and $41 \pm 16\%$ at greater depth (2000-1000 m) (Fig. S2).'

Line 122 - this is either discussion, or, if it is part of your results, you need to describe this data and statistical analysis in the methods

The extraction of these data is now described in the *Material and Methods* ('2.3.2. Extracting historical data of *Calanus hyperboreus* vertical distribution in the CAO'). The text is changed to 'The abundances (ind m⁻³) of *C. hyperboreus* and *C. glacialis* encountered during MOSAiC overlap with the compiled historical data (1935-2016) from the CAO (0.1 to 10 ind m⁻³) (Fig. S2)', which appreciates the spread of both datasets.

Line 131 - remove comma after Both

This sentence is not included in the revised version of the manuscript.

Line 192 - wouldn't a regression analysis be more appropriate here, since you are comparing 2 numeric variables (temperature vs density)?

We think that our data are too few to prove (or reject) a linear relationship between temperature and density. Some tissue samples showed the highest density at 1°C and some at 2°C or 3°C. We therefore removed the ANOVA result here. Based on further reading on the topic of temperature and density change in lipids, we came across a potential role of cholesterol. We therefore present now the relationship between maximum tissue density and cholesterol content, which is significant positive across our data set (Fig. 6).

Line 193 - could this be explained spatially? You mentioned the one station north of Svalbard, but where do all the others come from?

Yes, the origin of the copepods might play a role. The station north of Svalbard is in the western Nansen Basin (therefore we now call the station 'wNB') and has a 'much' warmer Atlantic layer (see Fig. 6a), while the other 6 stations were all much further North (85°-89°N, Aug-Sep'20 in Fig. 2). Again, we think that our data are too few to entirely resolve this variability here, but it shows an important pathway for the further understanding of vertical dispersal. Here we don't want to overinterpret our data, but make the reader aware that density regulating lipids (e.g. cholesterol) seem to play a role.

Lines 170-171:

Discussion:

Line 280 - how big of an issue are visual predators in the CAO under multi-year ice in the polar night?

MOSAIC has shown that the sea ice in the CAO can be highly mobile during winter and often breaks up, which would at least allow moonlight (star light, northern lights) to penetrate the upper water column. Such ice activities could also stimulate bioluminescence. Thus, any aggregation near the ice most likely leads to a higher predation risk than overwintering at depth. See also the further *Discussion* mentioning omnivorous/ carnivorous copepods and *Themisto* in winter surface waters.

Line 297 - How did you determine this?

This is based on the overall abundance data over the winter months. We now wrote: 'Based on the abundance (ind m⁻³) timelines there was no evidence of major winter mortality for the adult and subadult *Calanus* population in our data (Fig. 3).'

Line 335 - is this unique for this species? Are there any other examples in animals where tissue density increases with temperature?

We had a thorough look at the literature and found one mechanism that can lead to higher tissue density with rising temperature. This increase in density is sensitive to the concentration of cholesterol in the biomembranes of the animals. That animals use cholesterol in buoyancy control is known for deep-sea fish (Phleger 1998). We now adjusted the figure (new Fig. 6) to illustrate the role of cholesterol for *Calanus hyperboreus* and described the mechanism in the *Discussion* L349-L377

Methods:

Lines 424 - As mentioned above, I think this is a misleading way to present your data.

We agree and have removed this sentence.

Lines 449-450 - how did you take stratified net tows using the ring net? Did you have some kind of closing mechanism?

Yes, we used a mechanical messenger from Hydrobios that goes down the wire and releases a closing mechanism. This is now described in the *Material and Methods* L. 518

Figures:

In all figures, please re-work captions for clarity. Make them succinct, without extra information that belongs in results or methods. For example, in Fig. 5 do not include "no or minor changes" etc. - the reader will see that by looking at the figure, that belongs in the

results text. You also do not need to write how you prepared the tissue samples - that belongs in methods.

We agree and have re-worked all figure captions.

On figure 4, instead of explaining a-f/g-i in the caption first collectively and then individually, you can make sub-headers on the figure itself (“Body characteristics”, “Abundance”), and in the caption just define each panel a single time.

Yes, we have simplified the caption of Fig. 4 (now Fig. 3) and explained every panel just once.

You also need to decipher all used acronyms in every figure caption (AF, FA, PUFA, CAO).

Yes, we did so.

Figure 1:

This is a beautiful graphic, but it needs to be reworked a bit.

Please note, this image has been produced by a professional artist (Glynn Gorick) and any subsequent changes can only be done by him, with additional payment. We therefore requested only the essential changes.

First, please re-plot in chronological order - you sampled Nov-March before July.

Rev. 1 suggested to present this figure as a conceptual model in the *Discussion* rather than in the *Result* part. We followed this advice. As a conceptual model, we have the ‘freedom’ to present the data as a seasonal cycle from summer over the autumn descent and then through the winter, even our actual sampling was not in that order.

Second, in addition to your copepods densities also include numbers (%) for each layer - for example, in Nov-Mar, was it 30 or 50 or 79% of the population below 1000m? At the moment it is really impossible for me to say.

Originally, we only presented the AF in this figure, but have now changed to include both AF and CV. Therefore, the number of individuals in each panel have been corrected to present the average percentages of AF and CV, with the actual numbers been given in the figure caption.

Third, the circles on the right are confusing, and in my opinion, do not add much to the figure.

As mentioned in the figure caption, the three circles represent the 3 topics that we studied alongside the vertical distribution: feeding history, offspring production and predators. All three topics revealed differences between the upper and deeper ocean as explained in the figure caption.

Can you make separate vertical panels (can be a little offset from the rest) showing main food sources and predators in the upper vs. lower ocean? The “offspring” circle is the same, as far as I can tell, and is already shown with the eggs/nauplii panel.

The 'offspring' circle refers to gonad development and the vitality of the offspring (addressing the common idea that only AF at depth develop their gonads and produce offspring). We updated the upper circle, showing a second mature female, to indicate the higher numbers of mature females in the upper ocean (Fig. 3b).

Figure 2:

Please make an insert showing your study area on a broader geographic scale. Please add place names (islands, land masses, ocean basins and ridges).

We included another map showing the locations of MOSAiC and the two previous drift expeditions in the CAO (SP-5 and SHEBA) against the maximal winter sea ice extend during MOSAiC. Land masses and ocean basins etc are labelled.

Do the colours mean months? That is not clear.

Yes. But we changed now to use the same colours as in Fig. 2c (black, red, green), which makes it clearer.

Please also add years to the months, or indicate the direction of the drift, so it is apparent that Nov/Dec were sampled before Jun/July.

Years were added.

Describe also the stations (Aug/Sep) that are not on the drift line

We now write: 'Separate station during Aug'20 and Sep'20 were sampled while the vessel was in transit.'

Figure 3

Please re-order the panels so they are in chronological order. I think the panel eggs/nauplii can also be moved from supplementary into the main text.

We now show the three seasons in the same graph to easier recognise the differences (Fig. 2c), in line with the presentation of the potential predators (Fig. 7b). The panel on eggs/nauplii has been moved into the main text.

Figure 4 (did you mean Fig 3?)

Since here you are showing an actual time series, this figure is particularly misleading, as your sampling points are not in chronological order. Please re-plot (including gaps for the missing months between them)

Due to the lack of suitable summer data, we now show only the winter data – which were all sampled in high frequency and along the drift line. The winter data is the most consistent seasonal data set we have collected during MOSAiC (due to various logistic challenges in spring and summer) and of especial value due to rare winter sampling in the CAO.

Even if they are chronological, it is generally not good practice to connect points with lines, since that implies a linear transition from one point to the next, and you do not actually know the dynamics between your two points, and especially since you are also moving through space and not only time. This is particularly apparent in your abundance plots which jump

wildly from date-to-date. Consider plotting this figure with paired bar plots instead of points/lines.

We agree with the reviewer and now plot the individual data with a trend line focussing on temporal changes over the winter months.

Figure 5:

5b - In the caption it says *C. glacialis*? I thought we were talking about *C. hyperboreus*?

Yes, for comparison, we have included *C. glacialis*. This is now explained at the end of the Introduction: 'Where possible, the congener *Calanus* species *C. glacialis* was examined for comparison.'

Could 5b be shown as a regression instead of an ANOVA? You can use a mixed model and include your samples as a random factor.

We now show a regression between the copepod's maximum density under the tested range of temperatures and their cholesterol content (Fig. 6d). The ANOVA was removed.

5c-d. This is confusing to me. I would collapse these two panels into one, instead of artificially separating them into groups - there are not so many groups that the reader can't distinguish between them. Re-color in a colour scheme that logically goes from shallowest to deepest.

We followed the reviewer's advice to colour-code the samples (not by depth, but by temperature within their depth strata) and plotted them all in the same panel. Thank you for the advice.

Reviewer #3 (Remarks to the Author):

I have written an evaluation report that is attached below.

In brief, the manuscript reports valuable results on vertical distribution of the dominant copepod *Calanus hyperboreus* and other zooplankton in the CAO, along with data on lipid chemistry and trophic markers. The observations on *C. hyperboreus* are interpreted to reflect partial migration where one part of the population remains in the upper water layer in winter. The notion is one of winter dispersal rather than translocation to depth, as stated in the title.

Thank you!

A weakness of the manuscript is that it reflects poorly the many previous studies of *Calanus hyperboreus* and other zooplankton in the CAO, including from earlier US and Soviet ice drift stations.

We agree with the reviewer and sincerely apologize. We have now selected two previous ice drift studies: the Soviet Severny Polyus (1950-1956) and the US SHEBA (1997/1998) for comparison with the MOSAiC data. The winter location of the three expeditions is shown in Fig. 2a and the actual data in Fig. 2b. We chose these two expeditions as they are the only that continuously deployed a surface net (down to ~200 or 250 m) and a deeper net (from 200 to 1000 or 1500 m) throughout the winter, and are therefore to some extent comparable to net samples taken during MOSAiC.

These, and other studies in subarctic waters, have demonstrated migration into deep water, with dispersal over a wide range of depth, which in the CAO includes the upper water layer

under the ice in mid-winter. In my view, the issue is not translocation to depth or dispersal, but rather translocation and dispersal.

We think that 'dispersal' is the primary winter behaviour as it is common in all biomass-dominant lipid-storing seasonal migrant species (including *Calanus glacialis*, that is sometimes incorrectly referred to as a non-migrant species), while 'the translocation to depth' is more variable – both, among species as well as for the same species across different locations. The 'dispersal' holds always true (see Fig. 1).

The question for the CAO is why the dispersal includes the upper layer.

Because the CAO is a core habitat for C.h. while in some subarctic regions and the North Atlantic, the warm surface waters may 'prevent' C.h. from upwards migration. We reached this conclusion partly through comparison with Antarctic krill (see text earlier, in reply to Rev. 2). Krill show a similar vertical restriction near the sea bed at South Georgia in summer, where it is also the warm temperatures that seem to prevent occupation of the surface waters by this stenotherm species.

There is a strong emphasis on the generality of partial migration including for birds and mammals.

We think that ecological consideration about drivers of seasonal migration can be transferred across species and might encourage us or the reader (or the reviewer) to investigate new hypothesis also for copepods.

I would have liked to see more focus and emphasis on the specific conditions of the CAO habitat for *Calanus hyperboreus*, building from the knowledge gained by previous studies such as Dawson (1978) and Rudyakov (1983).

We think that the two should not exclude each other – we can discuss birds and mammals and still look at previous studies in the CAO (Rev. 3), and consider physical forcing in the CAO (Rev. 1) and observations from the Norwegian Sea (Rev. 2). We tried to follow all of the advice, rather than excluding one for the benefit of another. Seasonal migration is a very complex behaviour and therefore it requires complex approaches to study and complex considerations to understand it. Looking at previous data is a very valuable avenue, but there are also other avenues worth exploring (diet and body composition, physical forcing, predators, risk-reward considerations etc.)

General evaluation

The manuscript describes results on the large calanoid copepod *Calanus hyperboreus* obtained during the MOSAiC expedition with RV Polarstern, which was frozen into and drifted with the pack ice of the Central Arctic Ocean (CAO) during the winter 2019/2020 (September 2019–October 2020). The MOSAiC expedition was unique in the breadth and sophistication of multidisciplinary observations across atmosphere, ocean and biology. It is to be noted that it was not the first of its kind as it followed a long tradition of winter and year-round observations from drifting ice camps by US and Soviet scientists from before and in the decades after WWII.

Yes, and we sincerely apologize for not mentioning the previous work. Two of these expeditions (Severny Polyus and SHEBA) have now a very prominent place in our manuscript, with their location being shown on the first map (Fig. 2a) and in the first and second data plot (Fig. 1 and Fig. 2b).

The manuscript provides useful observations of the dominant zooplankton species *Calanus hyperboreus* in the CAO, which add to our knowledge and are certainly of broad interest to polar ecologists. This concerns especially the observations on vertical distribution and lipid chemistry of copepodite stage CV and adult females.

Thank you!

However, I find some difficulties in the way the results are presented and interpreted.

This relates to:

1. The notion of dispersal rather than translocation.
2. Scant reflection of previous literature and knowledge.
3. Too much emphasis on generality and too little on CAO specifics.
4. The validity and ecological relevance of the measured tissue density.

We reply to those points further down.

Before I go into these four points in more detail, I will draw up a brief sketch of the species and its habitat. *Calanus hyperboreus* is distributed in the Arctic Mediterranean Sea (which is the Nordic Seas plus the CAO, openly connected through the Fram Strait) and the subarctic gyres of the NW Atlantic. Baffin Bay and the Greenland Sea are the two core areas for the species, both areas characterized by seasonal sea ice and a lack of visual-feeding planktivorous fish.

Calanus hyperboreus can have a one-year life cycle in the ice-free parts of the distribution area but requires more than one year in ice-covered waters at high latitudes. Thus, in the CAO, it may require up to 4 years to complete its life cycle. *Calanus hyperboreus* can reproduce successfully in the southern part of the CAO where there are more light and less heavy ice conditions, but it may at least partly be an expatriate in the central part of the CAO.

We found mature females, eggs, nauplii and CI-III, CIV-V in the Amundsen Basin during MOSAiC, suggesting it to be a core habitat.

The water masses in the CAO are mostly of Atlantic origin. The Atlantic water flows into the CAO via the two prongs of the Atlantic gateway (the Fram Strait and across the Barents Sea shelf) and circulates cyclonic (counterclockwise) around the basins of the CAO with a residence time of <10 to ~50 years. In this large-scale circulation, *Calanus hyperboreus* goes through its life cycle and performs its seasonal vertical migration. Due to the sluggish circulation and long residence time of water, and the multiyear and multigenerational development of *Calanus hyperboreus*, we should expect a heterogeneous history of its population in the CAO.

With this as a general backdrop of information, I revert to the main issues I raised earlier.

1. Dispersal versus translocation to depth

The notion I have from reading literature is that *Calanus hyperboreus* has a seasonal migration to overwinter at depth, where it disperses over a wide water column. See e.g. Østvedt (1955) who described an annual cycle of *Calanus hyperboreus* (and *C. finmarchicus*) at Weather station 'M' in the Norwegian Sea.

We think that the Østvedt (1955) study in Norwegian Sea might have contributed to the common idea that all *C. hyperboreus* CV and AF overwinter at depth, because here indeed ~90% of the AF and CV were found below 1000 m and almost 100% below 600 m. However, even in May and June, 50-70% of the population stood below 600 m. This is not just

seasonal migration, but also avoidance of the warm surface waters in a location that is not *C. hyperboreus* core habitat. We now describe our interpretation of the observations by Østvedt (1955) at the beginning of the Discussion, L. 257

Conover (1988) described it as an interzonal migrant like the *Neocalanus* species in the subarctic gyres of the North Pacific. This pattern of large-scale downward migration with dispersal at depth has been documented even more extensively for *Calanus finmarchicus* (see Melle et al. 2014 for a review).

In the revised version, we now mention *C. finmarchicus* based on a paper by Heath et al. (2000) from the North Atlantic/ Norwegian Sea (L 451). Here we see major dispersal during winter with some part of the population overwintering in surface waters of the deep basins or in shallower waters on the shelf. Again, part of the population stays in the upper water column during winter if it is their core habitat.

So, for me, these large calanoid copepods perform an extensive vertical migration with translocation to depth and dispersal.

We must respectfully disagree here, and think that, these large calanoid copepods disperse (spread out) during winter. How deep they go might vary between species and locations, but some percentage of the population stays near the surface (see our new Fig. 1 --- it is the same principle for all the species considered; aggregation near the surface in summer and spreading out in winter). The dispersal has ecological benefits – less competition for remaining resources, less tractable for predators (as explained in the *Discussion* – ‘Benefits of dispersal and the ecological context’). ‘Translocation to depth’ excludes *C. glacialis* and maybe even *C. propinquus*, while ‘dispersal’ covers the behaviour of all of them.

Putting these two things as alternatives (translocation or dispersal) is to me a wrong emphasis and interpretation.

We think that ‘dispersal’ is the key mechanism, while ‘translocation to depth’ does not always apply.

Therefore, already the title of the paper is problematic. In the present case with *Calanus hyperboreus* in the CAO, the issue is more why the dispersal extends all the way from the mesopelagic layer up through the upper layer to the surface. This is what should be discussed, and to me an obvious factor is the strong winter darkness at high latitudes (close to the North Pole) and under thick ice.

We found ~40% of the *C.h.* population (AF and CV) below 1000 m depth during winter, Rudyakov (1983) found similar values (18-35%). The question is: How can this be explained with ‘strong winter darkness under thick ice’? Why are 40% still moving so far away from the surface? This can’t be explained with the specific conditions in the CAO, it can, however, be explained in terms of ‘dispersal’ as a bet-hedging strategy, -- spreading out vertically and thereby spreading the risk of predation and the chance to find food. It’s not strictly true, that there are no predators near the surface in the Arctic, there are lots of species that can feed on *Calanus* eggs and nauplii (see the paper by Darnis et al. 2019). We should not only think of the visual predators that might feed on adult *C.h.*, but also consider all the copepods near the surface that feed on eggs or nauplii. We argue that spreading the risks and benefits ensures long-term survival under uncertain conditions, regardless of whether the habitat is in the CAO or in the Antarctic or anywhere on land.

2. Previous literature and knowledge

There have been many studies of zooplankton and their vertical distribution in the CAO. There are no references, for example to the extensive work by Ksenia Kosobokova, both her early and later work. Kosobokova worked with zooplankton samples collected from Soviet ice-drift stations, e.g., Kosobokova (1978) described the diurnal variation in vertical distribution of *Calanus hyperboreus* and *C. glacialis* in the CAO. Later work included detailed studies and descriptions of community composition and vertical distribution of zooplankton in the CAO, in collaboration with Hans-Jürgen Hirche, Russ Hopcroft, and others –e.g., Kosobokova and Hopcroft 2010, and Kosobokova, Hirche and Hopcroft 2011. Reference to these and other papers would have provided a much better backdrop of what is already known about vertical distribution and migration of *Calanus hyperboreus* and other species of zooplankton in the CAO. A report prepared by the ICES/PICES/PAME joint working group on integrated ecosystem assessment of the CAO (WGICA) provides a summary and review of both older and newer literature on zooplankton (Skjoldal, H. R. (Ed.). 2022. Ecosystem assessment of the Central Arctic Ocean: Description of the ecosystem. ICES Cooperative Research Reports Vol. 355. 341 pp. <https://doi.org/10.17895/ices.pub.20191787>).

Thank you for this valuable historical context, and we realise that we did not give this sufficient emphasis in our initial manuscript. All the mentioned work by Kosobokova and co-authors was actually included in the 'historical data (1935-2016)', that was published by Kvile et al. (2019) using previously compiled data sets by Hopcroft and others, and new compilations. We should have mentioned these important source data much more explicitly. In the initial version of the manuscript, we did not explain this 'historical data' in the Methods, however, we extracted data from this compilation for our Fig. 2, which is now in the Supplements (Fig. S3). This data compilation is now explained in the Methods ('2.3.2. Extracting historical data of *Calanus hyperboreus* vertical distribution in the CAO'). It includes 51 individual data sets that unfortunately, we can't mention here individually. In this manuscript, we focus on winter data and give now special emphasis to Rudjavov (1983), Ashjian et al. (2003) and Dawson (1978).

Specifically, Dawson (1978) and Rudyakov (1983) described seasonal vertical distribution of *Calanus hyperboreus* under the ice in the CAO, including sampling through the winter period. There is no reference to Rudyakov, but there is reference to Geynrikh et al. (1983) where Rudyakov along with Kosobokova were coauthors. Dawson (1978) and Geynrikh et al. (1983) are referenced in the manuscript but in a wrong context: "Previous studies at lower latitudes likewise found *C. hyperboreus* CV and AF near the surface in winter (Dawson 1978, Geinrikh et al. 1983, Kvile et al. 2019),...". This is wrong. Dawson (1978) reported results from nearly two years of sampling (February 1970–January 1972) from the US T-3 Ice Island in the northern part of the Canada Basin at 84–85°N. Rudyakov (1983) reported results on *C. hyperboreus* sampled over several years from the Russian North Pole ice-drift stations (SP-2–5, 1950–1956) in the Canada and Makarov basins. This can hardly be said to be lower latitudes but comparable to MOSAiC.

Yes, we greatly thank the reviewer for making us aware of these winter publications and encouraging us to include those data. We used the 'historical data (1935-2016)' by Kvile et al. (2019) to extract the Severny Polyus data (1950-1956) and the SHEBA data (1997/1998) and plotted the part of the population that remains in the upper 200m for all three ice drift expeditions --- SP, SHEBA and MOSAiC. It's a great result (Fig. 2b) – it shows overall a very consistent pattern across AF and CV. We also looked at the Dawson (1978) data. However, this dataset did not include many deep nets and was therefore not suitable for the comparison with the other three paper. We discuss the Dawson (1978) paper from L339.

3. Generalities versus CAO specifics

The manuscript generalizes the pattern of partial seasonal migration, interpreted for *Calanus hyperboreus*, to the wider cases of geographical (latitudinal) migrations by birds and mammals. While this might be an interesting comparison and analogy in a general review, it seems to me to be somewhat far-fetched in the present case of discussing the specific observations of seasonal vertical distribution of *Calanus hyperboreus* in high latitude parts of the CAO.

We think that seasonal migration has developed multiple times during animal evolution because it poses some benefit to the populations, but this benefit might change under climate change. Considering such general ecological principles and learning from other fields can bring the perspective needed for this broad-readership journal. This perspective can also help to understand our *Calanus* data, develop hypothesis and test them across the various other species or locations. Rev. 1 seemed in favour of such wider considerations and many of the co-authors gave likewise positive feedback, thus, we think it will engage readers who are not primarily interested in copepods or the CAO.

I was expecting to see a gain of knowledge on vertical distribution and migration, building from the observations of Dawson (1978) and Rudyakov (1983). Dawson observed a complex pattern for adult females, with many individuals found in the upper layer (0–100 m) under the ice during mid-winter (December–February), and then descending to depths of 200–300 m during spring. My own thinking has been that this is related to the transition from the darkness of mid-winter, when there is no light cue to keep the dispersed copepods away from the upper layer, to a response to the strong increase in light around the time of spring equinox. One reason why this matter is the presence of visual predators (polar cod *Boreogadus saida* and Arctic cod *Arctogadus glacialis*) up under the ice in winter, described by Andryashev et al. (1979).

We were careful to use and discuss the data by Dawson (1978), as during the first winter (1970/1971), he only sampled the upper 300 m water column and during the second winter (1971/1972), the *C.h.* abundances were generally very low. Thus, he might have missed part of the *C.h.* population during the first winter, and encountered low abundances throughout the second winter. However, the data from the SHEBA expedition show low proportions of *C.h.* in surface waters during the equinox, which might be in line with what the author describes. On the other hand, the MOSAiC and the Severny Polyus data do not confirm especially low proportions of *C.h.* in surface waters during the equinox. We therefore did not discuss this phenomenon in the manuscript.

The way the data is presented is mainly to show the contrast between vertical distribution (and associated lipid chemistry and trophic markers) between the winter (November–March) and summer (August–September) situations. The data are averaged for these two time periods, and the bi-weekly resolution that lies in the data is not used. There might be a reason for this, but it would have been of interest to see if there were some patterns in the upper part of the water column under the ice, as was observed by Dawson (1978).

We now present the fourteen winter sampling events as individual data points, either as abundance data (ind m⁻³, Fig. 3) or as vertical distribution data (% in 50-0m, % in 200-50m etc., Fig. S2). After mid-Mar 2020, sampling became very patchy – i.e. sampling events became rare, it was partly away from the drift line (when the ship relocated to Svalbard) and also the depth strata changed, with deep nets being missed altogether. Therefore, we are not presenting the spring-summer data here. However, there are data from the ROV net that was deployed under the ice almost year-round and those data will be submitted for

publication by Cornils et al. shortly. Hopefully, the reviewer will find answer to those questions in the ROV net data set.

4. Tissue density

The density of freeze-dried and homogenized bodies of *Calanus hyperboreus* was determined by a pycnometer using helium gas to fill the voids around solid structures in the material. There are no references to applications, and it is not clear if this method has been used with zooplankton previously, and what relevance tissue density determined in this way has in relation to *in situ* buoyancy. Before a method like this is used in the context of migration mechanisms, it should be validated against other methods for determining density.

There are several publications that have previously suggested a special role of lipids, either wax esters (e.g. Yayanos et al. 1978, Visser and Jonasdottir 1999) or polyunsaturated fatty acids (Pond et al. 2012), for the buoyancy regulation in copepods. Here we follow on from these publications looking (1) if animals sampled at different depths in the CAO vary in the density of their lipid-rich tissues – despite similar overall lipid and wax ester content, and (2) how the density of these lipids changes when passing a temperature range similar to what copepods experience when migrating from the surface CAO through the Atlantic layer and then to greater depth. So, our aim was a mechanistic understanding of changes in density under certain conditions (here especially the warmer Atlantic layer) rather than estimating the true *in situ* density (which is technical very difficult due to potential changes in the water content as soon as the animals are lifted by the net). Our data show that animals from different depths and a different *in situ* temperature range vary in the cholesterol content and that this can lead to different tissue density. Thus, our conclusion was that the cholesterol content might play an important role in density control and we provide a mechanism for this based on a literature survey and discussing our results with Judith Peters who has authored some of these papers L.349-377.

A pycnometer was used for two reasons, first, it allows to detect very small differences in tissue volume (a benefit when working with small copepods) and, second, it can be placed in a thermostat and therefore polar temperatures can be simulated (a benefit when working with *C.h.*)

5. Some other issues

The vertical distribution of *Calanus hyperboreus* is shown with relative numbers as anomalies in Fig. 1b and in Fig. S1. I understand that this is to show deviations from evenly dispersed distributions. However, the results would have been more directly comparable to other studies if they were given as numbers per m³, at least for the vertical profiles for each sampling occasion in Fig. S1. Integrated numbers (per m²) are given as average values on the panels in Fig. 1 (which is useful) but are not included on the panels of Fig. S1. Average values per m² for each depth interval are given in Table S1 for the summer and winter periods.

We now give integrated numbers (ind m⁻²) for all panels in Fig. S1. We also show individual winter abundances (ind m⁻³) in Fig. 3a-d, and winter relative distributions for each sampling date in Fig. S2.

Anomalies are used for the vertical distribution of potential predators in Fig. 6. I would have preferred real numbers rather than relative numbers also here. Integrated numbers per m² are given for both abundance and biomass on each panel, which is useful.

We now give predator abundances (ind m⁻³) for each sampling date and depth strata in Table S2.

I note that the biomass values are very high for two of the taxa, ~4 g C m⁻² for *Paraeuchaeta* and ~6 g C m⁻² for chaetognaths. These values would be high anywhere and especially in the low productive CAO. I suspect they are wrongly high due to too high values used for average individual weights to convert numbers to biomass.

We thank the reviewer for spotting these mistakes. We mixed the values for ostracods and *Paraeuchaete*. All values have now been checked again and considered correct.

In consideration of potential predators, I find that two taxa are missing but which should have been mentioned for completeness. This is the Arctic cod *Arctogadus glacialis*, which is possibly a pelagic species in the CAO, and the group of siphonophores which are known to be present with several species in the mesopelagic layer of the CAO (e.g. Raskoff et al. 2010).

We now write: 'Based on their total estimated biomass, cnidarians, chaetognaths and other copepod species are likely the main *Calanus* predators in the CAO (Fig. 7b), while ctenophores and Arctic cod (*Arctogadus glacialis*) were not considered here.'

Siphonophores are included in the 'Cnidaria' group.

Specific comments

-Line 69 –Pond (2012) is not in the reference list. Pond et al. 2012 is.

Thank you for spotting this! We changed this to Pond et al. (2012)

-Lines 89-93 –There are two problems with this sentence. The first part ('first to sample zooplankton in highest Arctic latitudes') may give a misleading impression, while the second part ('a unique opportunity to understand the seasonal vertical migration of the biomass-dominant copepod *Calanus hyperboreus*') is not followed through with the results as they are presented.

We removed this sentence. In the revised version, we only say: 'The Multidisciplinary drifting Observatory for the Study of Arctic Climate (MOSAiC) expedition provided an ideal opportunity to address these research gaps.'

-Line 112 -Use of 'Arctic' here is somewhat ambiguous. Does it refer to *Calanus hyperboreus* as an Arctic species, or to *C. hyperboreus* in the high Arctic part of its distribution?

We changed the subtitle to: '3.1. Seasonal differences in the vertical distribution of *C. hyperboreus* and *C. glacialis* in the CAO'

-Line 115 –This may not be quite accurate since it appears that some of the September samples were collected from the Nansen Basin, south of the Gakkel Ridge—Fig. 1a.

We agree and changed this to 'CAO ≥85°N'.

The samples in August and September seem to be collected away from the drift route of the ship. I assume the Polarstern was mobile at this time, but this is not described in the manuscript.

We now write in the figure caption: 'Separate station during Aug'20 and Sep'20 were sampled while the vessel was in transit.'

-Line 118 –'descend' (not 'descent')

We removed this sentence.

-Line 129 –'while the congener *Calanus* species *C. glacialis* reproduced in summer.' How was this observed –egg production, presence of young copepodites?

We now write: '..., mature *C. glacialis* were seen in summer.'

-Line 172 –I believe the start of the sentence should be reversed, like this: 'In late summer compared to winter,....'

In the revised version, we are not comparing the summer and winter data anymore. The sentence has been removed.

-Line 181 –Fig. 4g-i. Strictly speaking, this should be Fig. 4h, i, since the sentence is about eggs and nauplii.

In the revised version, we now write: 'The abundances of copepod eggs significantly increased in the upper 200 m over the winter, while the abundances of *Calanus* nauplii increased in both the upper and especially the deeper ocean (Fig. 3c,d).

-Line 181 –'C. hyperboreus reproduced between Nov.-May'. Is this a general statement, or was *Calanus hyperboreus* observed to reproduce? If the latter, how was reproduction observed –egg production, presence of males, early copepodites?

We now show the abundance of mature AF (Fig. 3b), which follows a bell-shaped curve from Nov to May, thereafter we found no mature *C.h.* However, the sentence mentioned by the reviewer is not included anymore in the revised version.

-Line 181 -It is a weak point that eggs and nauplii were not identified to species. However, I find the assumption to be robust.

OK, thanks.

-Line 264 –'prioritize food uptake over predator avoidance'. I assume from the context that this refers to the winter situation. Did the females which were collected near the surface feed, e.g. did they have stomach content?

Yes, there was evidence for winter feeding via gut DNA sequencing (Shoemaker pers. comm.) and via trophic marker concentrations (Fig. 3h). The gut DNA analysis is part of another study that is currently prepared for submission by Shoemaker et al., while the trophic marker analysis was part of this study. We now write at the end of the overall *Discussion*: 'Both stomach content analysis (Shoemaker, personal communications) and trophic marker concentrations (Fig. 3h) indicate winter feeding.' L.367

-Lines 287-288 –'still abundant'. Abundant is here a relative term. The abundances are overall low compared e.g. to subarctic seas.

We agree and changed to '... are still occurring.'

-Lines 288-289 –'The developmental stage that benefits most from AF wintering at depth seem to be *Calanus* nauplii,..'. This should be rephrased. *Calanus hyperboreus* are known to spawn at depth during winter, so this observation is in line with what has been previously reported.

We found more mature females and eggs at surface (Fig. 3b, c), but more nauplii at depth (Fig. 3d), which suggests that the spawning in deeper water benefits primarily the nauplii (not so much the females or eggs). Considering such details, I think our observation is novel and worth mentioning here.

-Lines 369-372 –It is uncertain whether observations in Aug-Sept can be taken to reflect resident vs migratory components. The individuals in the upper layer in August-September may migrate to deep water (and disperse) later in the season and thus be part of the migratory component.

When comparing the vertical distribution in Nov 2019-Mar 2020 and Aug-Sept 2020, there is very little difference. In late summer, there were 50% of the AF and 65% of CV below 500 m depth and in winter it was 55% AF and 82% CV. This means that at the time of our sampling in late summer, most 'migrants' would already have left the surface. This can also be seen by the historical data (SHEBA and SP, Fig. 2b): after Sept, the surface population remains relatively constant at 30% (CV) to 50% (AF), when only the upper 1000 m are considered.

There is some, but not very pronounced, difference in NL-PUFA (Fig. 3f) but not in PL-PUFA which occur in higher quantities (Fig. 3e).

Thanks for this comment. It made us aware that the y-axis might be misleading. Instead of (% TFA) it should be (% PL-TFA) and (% NL-TFA). The two fractions, PL (membranes) and NL (wax esters), were analysed separately, but NL is overall the bigger fraction (>80% of total lipids is within wax esters, see Fig. S4), and therefore the differences in the NL-composition are significant and also reflected in the total lipids. In the PC (Fig. 4a), we see all 22:6(n-3), 20:5(n-3) and 18:4(n-3) (right panel) to align with the AF at the surface (left panel). Both are in blue colour. In the figure caption, we now added the words 'membranes, <20% total lipids' after PL and 'wax esters, >80% total lipids' after NL.

-Line 386 –Berge (not Berges)

Thank you, we changed this.

-Lines 407-408 –Note that there is deep seasonal migration through the warmer Atlantic water in the Norwegian Sea (Østvedt1955), and overwintering depth is deeper there than in the colder waters of the Greenland Sea (Hirche).

The temperature profile shown by Østvedt (1955, Fig. 3) has warm water of 9°C at the surface and then it cools down over the upper 600 m to 0°C, and -1°C even deeper. Thus, the overall drop in temperature helps to increase the density of the copepod during their downward migration (making the migration easier). However, in the CAO, the copepods start at -1.7°C and after 200 m they reach water that is almost 4° warmer (2°C), which would naturally expand their lipids and make them more buoyant (making the migration harder). Thus, the problem is not crossing the Atlantic water, but moving from the coldest water at the surface into warmer water, a situation that is (probably) only found in the CAO.

-Line 417 –‘the first year-round interdisciplinary study of atmosphere, sea ice, ocean, ecosystem, and biogeochemical processes during the transpolar drift across the Central Arctic Ocean (CAO)’. I am not sure this is true. There should be reference to the previous extensive ice-drift stations, and to expeditions such as the Canadian/US SHEBA.

We removed this sentence and gave both the SHEBA and Severny Polyus expeditions a prominent place in the revised version of this manuscript.

-Line 458 –‘Live’ instead of ‘lively’

We changed to ‘active’ specimens.

-Line 471 –‘gently broken up with a spatula’. How well does this homogenize the sample material? Has this been studied?

The problem with using a mortar and ‘grinding’ very fatty samples like *Calanus hyperboreus* tissue, is that the lipids stick to the mortar and can’t be recovered for the sample. Gently breaking the sample with a spatula seemed a better alternative with very little loss of lipids. During our initial test phase, we compared a range of published procedures for lipid extraction and the one we final used, gave most consistent results.

-Line 489 – $\delta^{13}\text{C}$ –C should not be superscript.

Thank you. We changed this.

-Line 494 –should be USGS63

Thank you. We changed this.

-Line 619 –‘living at in’.Delete ‘at’.

Thank you. We changed this.

-Fig. 4 –There is information on lipid composition and stable isotopes for POM from the Chl maximum layer. It is not clear what this POM represents. Does it make sense to speak about a Chl maximum layer in winter?

Based on comments of Rev. 2, we changed Fig. 4 (now Fig. 3) to include only winter data. For simplicity, we have also removed the POM data.

-Fig. 5 –there is reference to one station north of Svalbard. There are several stations N of Svalbard in Fig. 1a. Could the one used here be identified?

We now named the station 'western Nansen Basin' ('wNB') on the map of Fig. 2 and throughout the text.

-Fig. 6 –the line with TL values should be explained in the legend.

We now write 'Trophic level' at the figure margin and in the caption, it says:
'Potential predators are identified by the proportion of the *Calanus*-produced fatty acids (20:1, 22:1 isomers) in their total FA pool and their trophic level in Aug.-Sept. 2020.' The calculation of the trophic level is explained in the Material and Methods.

-Fig. 6 –what are the numbers on the panel for Cnidaria –abundance or biomass?

We now show two numbers also for Cnidaria, the first being abundance and the second biomass. e.g. in winter: 674/ 695 and spring 710/ 288

-Table S1 –What are the errors -SD or SE?

These are SD. We added this in Table S1.

Reply to the reviewers' comments:

SUMMARY:

We would like to thank the three reviewers sincerely for taking the time to go through our manuscript again.

All three reviewers acknowledged our substantial efforts in revising the manuscript after their first review. While Reviewer #1 was happy after this first revision round, Reviewer #2 had some minor issues remaining, but Reviewer #3 felt that three of her/his four major points had not been addressed sufficiently. We now followed the advice of both Reviewer #2 and #3 and changed the *Title*. We also added one further display item (Fig. S6) that shows the occurrence of all developmental stages during the MOSAiC expedition in the CAO ($\geq 85^\circ\text{N}$). Even though this is not a key aspect of our manuscript, it confirms that the CAO, as we encountered it in 2019/2020, was a more suitable habitat for *Calanus hyperboreus* to fulfil its life cycle than the Norwegian Sea and that our study was not conducted in a 'fringe region'.

Apart from changing the *Title*, adding this display item, a few additional references and some corrections to the text (see below), we felt that there is little more we can do to this manuscript. The three out-standing points raised by Rev. #3 are addressed in this rebuttal letter but have not led to major changes/ removal of text. Here the additional round of reviewer comments did not change our opinion.

Overall, we would like to thank all the reviewers once again for their continued interest in our manuscript. Their questions and advice having triggered a lot of thinking, literature study and data exploration for the authors. This was a very valuable experience.

REVIEWER COMMENTS:

Reviewer #1 (Remarks to the Author):

The authors have addressed the concerns I raised on the initial submission (R0) and the manuscript reads a lot better as a result. The authors have incorporated most of the reviewer comments (from all reviewers) and made fair justifications where some comments/suggestions could not be addressed. I appreciate the authors' efforts in reviewing the manuscript, and I do not have any further major comments to the manuscript at this stage.

We thank Reviewer #1 for acknowledging our efforts in revising the manuscript.

Reviewer #2 (Remarks to the Author):

The manuscript has increased substantially since the previous version, and the authors have done an admirable job addressing many issues raised by myself and the other two reviewers. I particularly appreciated the incorporation of the historical data from the drifting ice stations and SHIBA.

We thank Reviewer #2 for acknowledging our efforts in revising the manuscript.

Nonetheless, I still have a few remaining comments that I believe are rather important and should be addressed prior to publication.

1.) The title. I still believe that the authors are doing both themselves and potential readers a disservice by naming the paper "vertical migration of polar copepods" when in reality their data focuses mainly on *Calanus hyperboreus* in the CAO, even if other copepods and polar regions are briefly mentioned in a literature context. Perhaps consider changing to "a key Arctic copepod" or at the very least "large herbivorous copepods", since nobody is arguing that common, abundance-dominant species like *Oithona* or *Oncaea* are seasonally translocating to depth. Alternatively, consider something like "Vertical distribution patterns of key *Calanus* in the CAO support an active winter dispersal strategy in large herbivorous polar copepods"

We agree, and have now revised the manuscript title to: 'Reinterpreting seasonal vertical migration of large polar copepods as dispersal mechanism throughout the water column'. This title takes into account that the journal's instructions for authors allows a strict maximum of 15 words for the title. We agree with Reviewer #2 the need to clarify that this manuscript is about 'large' copepods, but avoid the term 'herbivorous' as *Calanus hyperboreus* are well known to feed omnivorously. The gut DNA analysis carried out during MOSAiC confirms the role of heterotrophic food especially in winter. We also think that other polar species are presented in this manuscript beyond being 'briefly mentioned in a literature context' (e.g. our new Fig. 1) Further down the list of comments, Reviewer #2 described our data compilation from other polar copepods as an 'actual meta-analysis of existing literature' (Fig. 1), which indicates that it is more than a brief mentioning. Moreover, results from *C. glacialis* are presented alongside those of *C. hyperboreus* for most of the studied parameter. We therefore do not reduce our study to *C. hyperboreus* in the *Title*. The revised *Title* does not oppose 'dispersal' and 'translocation to depth' anymore, which is in line with suggestions by Reviewer #3. Instead, we now say 'dispersal mechanism throughout the water column'.

2.) Some key literature is still missing, as was rightfully brought up by Reviewer #3. For example, the early and more recent works of Kosobokova; Kvile et al. 2018; as well as some important studies from the Greenland Sea. There are not so many publications about *C. hyperboreus* in the Arctic that they all can't be at least mentioned -- if you are claiming to redefine established paradigms, first you need to establish the state of the field.

Yes, we now cite Kvile et al. 2018 and Hirche et al. 2024 (see Reviewer #2's comment below) within the *Discussion* of whether *C. hyperboreus* is a resident or expatriate in the CAO (Line 290).

3.) The distinction between Results vs Discussion (and even Introduction!) is still very blurred - some contents that belongs in Discussion is still in the Results section, and vice versa. This should be addressed.

We do not entirely share this view of Reviewer #2. We noticed that there was also some discrepancy between Reviewer #1 and #2 regarding the best place for the conceptual model (now Fig. 8); Rev.1 suggested to move it to the *Discussion*, while Reviewer #2 wanted it to remain in the *Results*. This illustrates that the incorporation of certain display items is not always a clear cut. We hope the reviewers can bear with us on the tricky issue how to present the extensive new datasets from MOSAiC, older compilations from elsewhere and the more general schematics of seasonal vertical distribution.

More specific comments:

Figure 1. I am not sure that this figure belongs in the introduction, since it is not simply "setting the stage" but an actual meta-analysis of existing literature. I would put this figure

last, and reference it in the discussion instead. In the caption, say clearly in the first line "based on literature values"

As we discuss above, there are some conflicting opinions among authors and multiple reviewers on where to put display items. We think that the literature compilation in Fig. is a key introductory point of the manuscript, that illustrates the generality of the topic and can therefore open our manuscript to researchers that don't work on *Calanus hyperboreus*, but maybe on related species in the Southern Ocean or on different *Calanus* species in the North Atlantic and Arctic. Presenting this figure at an early stage likely increases the readership and therefore the overall research impact. We would like to keep this figure in the *Introduction*.

Line 118 onwards: Start the results with presenting your own data first, then followed by literature compilation.

See comment above. We prefer presenting the literature compilation in the *Introduction* to appeal to a wide readership.

Line 193 onwards (section 3.4). This is very interesting information, but most of this section as currently written belongs in the discussion. Here, you just describe the characteristics of the physical environment that you think are relevant.

This section (3.4) was implemented in response to a request by Reviewer #1, and he/she has subsequently acknowledged the incorporation of these data. It seems wrong, to subsequently remove these data. In the *Results*, we show the requested physical data derived from the MOSAiC cruise (Reviewer #1) and in the *Discussion* we discuss these data and build thereby the basis for our 'density' discussion.

Line 265. "Norwegian Sea is at the fringe of its habitat" Many would argue that the deep Arctic Ocean is also at the "fringe of its habitat" (e.g., Kvile et al., 2018, <https://doi.org/10.1111/gcb.14419>) If you think otherwise, you need to justify your opinion.

The Norwegian Sea is at the fringe of the *C. hyperboreus* habitat for thermal reasons, while the CAO might be at the fringe due to food shortage. Too warm surface waters can affect the vertical distribution, especially if food concentrations and therefore further energy intake are low, as would be the case in winter. Therefore, we wrote 'avoid the warm Atlantic water...'.and gave the temperature in brackets. We also point out that *C. hyperboreus* is a cold-water species.

See also the comment by Reviewer #3 under 'General', marked in bold. '**They rightfully point out that this location (weathership M in the Norwegian Sea) is at the southern fringe of the habitat for the Arctic *Calanus hyperboreus*.**'

The paper by Hirche 1997, that compared *Calanus hyperboreus* developmental stage composition in the Greenland Sea Gyre (~0°C) and Westspitzbergen Current (~5-8°C) at the same sampling period, clearly shows such an effect of higher temperatures on younger developmental stages. There are hardly any CI and CII occurring in the WSC, but many in GSG.

Line 276. "Within their core habitat". Related to the previous point, what is the core habitat for *C. hyperboreus*? How do you define "core habitat"?

We used the term 'core habitat' in relation to temperature, but realise now that this caused confusion as there is also the aspect of food availability to consider. We now write 'a water column with suitable low temperatures as found in the CAO'.

Line 337. See also this recent paper that supports this hypothesis:

<https://doi.org/10.3354/meps14665>

Thanks for making us aware of this interesting paper. We now included this reference.

Line 356. "We found a positive linear correlation". This phrasing should be in results. Here, you say something like "The positive trend between cholesterol and tissue density suggests..."

We changed this to 'The positive linear correlation between the maximum tissue density and the copepods' cholesterol content (Fig. 6d) requires further considerations.'

Fig 3. I get the appeal of this figure, but statistically speaking, it is not correct. Using simple linear regression for this data violates the most important assumption of SLR: independence. In your case, the non-independence is both spatial and temporal - you are not sampling in the same location, but drifting through space. To correct for this, you would need to introduce both a temporal autocorrelation correction (e.g., CAR1) and somehow incorporate the spatial variability into your model. Additionally, this analysis is not described anywhere in the methods section.

An alternative would be to just show this data as bar plots with whiskers for each month (similar to your Figure 1, with red and blue side by side as deep and shallow), and leave statistics out of it - that will still make the trends you observed quite visible.

Unfortunately, this alternative is not in line with the journal's requirement that individual data should be shown if n-numbers are low (< 10). We have now removed the R² and p-values and just provided a dashed trend line. As Reviewer #2 stated, we are looking at trends here.

Figure 6. D. Is this trend line for the averages or for the raw data? Instead of showing the standard errors, just show the actual data points (keeping your colors and symbols)

The trendline is for the averages. The 11 average cholesterol values derived from ~150 individual samples (usually 10-20 samples per presented average), while the density was measured on one pooled sample per species and sampling depth. Presenting all those individual cholesterol values in 11 different colour-symbol combinations would likely result in a confusing plot.

Reviewer #3 (Remarks to the Author):

Review of manuscript – 01 April 2025

Seasonal vertical migration of polar copepods is a winter dispersal rather than translocation to depth

Authors: Katrin Schmidt and co-workers.

Submitted to journal: Communications Earth & Environment

General

This is the second time I see this manuscript. I provided comments to a first version with my review dated 17 January 2025. This is now a revised version of the manuscript. I raised four major points in my first review, and regrettably, I find that only one of them has been adequately addressed.

This point was a poor reflection of previous literature and knowledge. The authors have now included more information from previous studies. In the Discussion, they now refer to the study by Østvedt (1955) on *Calanus hyperboreus* collected at weather ship station M in the Norwegian Sea. **They rightfully point out that this location is at the southern fringe of the habitat for the Arctic *Calanus hyperboreus*.** I don't have the publication by Østvedt at hand, but the species does reproduce in warmer conditions as seen in this study from a deep fjord just outside Bergen: Matthews, J. B. L., Hestad, L. & Bakke, J. L. W. 1978: Ecological studies in Korsfjorden, western Norway. The generations and stocks of *Calanus hyperboreus* and *C. finmarchicus* in 1971-1974. *Oceanologica Acta* 1. 277-284. The authors have also now included results from Soviet ice drift stations in 1950-1956 (Rudiyakov 1983) and the SHEBA ice drift in 1997-98 (Ashjian et al. 2003). Very usefully, results from these previous studies are now shown along with the new results from MOSAiC in Fig. 2b. The new results are broadly similar to the old results, which "suggests that during MOSAiC the winter conditions in the CAO were still similar" (to the conditions during the 1950s) (lines 447-449).

The concept of partial migration

A second point I raised in my first review was on the notion of dispersal rather than translocation. In my view, the previous literature has shown migration (or translocation) to depth AND dispersal. This can be understood in an ecological sense as migrating into the dark deep to avoid visual predators and spread out with dispersion to reduce the likelihood of being found by non-visual invertebrate predators.

The present authors make a contrast between these two aspects as reflected in the title – winter dispersal rather than translocation to depth. I take issue with this. The data from MOSAiC (as well as earlier results) show a translocation to depth as the difference between the vertical distributions in summer and winter, as shown in Fig. 2 b and c. I understand the authors as suggesting that this is merely a dispersal phenomenon and not a migration, basically a concept of migration OR dispersal.

Yes, we agree that the title could cause confusion and have therefore changed it to make it clearer. To conform: we do not suggest that it is 'dispersal' instead of 'migration'. We say that the animals 'disperse' apart from each other and use the extensive water column (here 2000 m sampled) to do so. This is in opposite to all animals migrating to great depth and sitting there together, which is the way seasonal migration of large copepods is often perceived and recently modelled. Thus, most of the animals migrate away from the surface, but to different depths that include great depths, which results in an even dispersal (as seen in our Fig. 8). To clarify this, we have now removed the phrase 'rather than translocation to depth' from the *Title* and instead of just using 'dispersal', we wrote 'dispersal mechanism throughout the water column'.

We don't know (at least not well) what the mechanisms of migration are, nor what the mechanisms for dispersal are. They could both reflect active swimming and/or changes in buoyancy. I have colleagues (Kjell Arne Mork and others) who work with profiling Argo floats equipped with video that records copepods in the Norwegian Sea. They document a rapid descent of *Calanus finmarchicus* from the surface layer to below 500 m depth in a few weeks' time around mid-summer. This suggests active migration, and it is possible that the

visual records may provide information on the behavior of these descending copepods. As far as I know, this remains to be investigated.

We agree, and this is an interesting topic that repays further study. The changes in density are likely to aid active swimming, so speeding up the process of migration, with the further possibility of using 'density corridors' as we described in Results 3.4.

The data on *Calanus hyperboreus* from MOSAIC in the CAO suggests a descent of adult females and stage V copepodites in late summer and autumn at this high latitude, in August-September suggested by Fig. 2b. The winter dispersal is not that easy to see, but the anomaly plots in Fig. 2c suggest a slight increase in concentration of *C. hyperboreus* between late summer/autumn (August-September) and winter (November-March). This would have been easier to see if data were shown with absolute and not relative units. Also, August-September is the time of the seasonal descent from the surface layer (as shown in Fig. 2b), and the contrast between the profiles for late summer/autumn and winter are therefore not ideal to document the seasonal descent and dispersal at depth, not the least because of the very coarse vertical resolution of the sampling. Apparently, Polarstern left the CAO in summer to return to resume sampling in August. Therefore, the dispersal of *Calanus hyperboreus* is not well documented in this study.

The interpretation that a fraction of the population of adult females and CVs remain stationary as a resident component in the surface layer during winter is a dubious claim not well supported by the data. The trendlines in Fig. 2b suggest that there is an increasing proportion in the upper 200 m between autumn and mid-winter. It is therefore possible that the two processes of descent (or translocation or migration to depth) and dispersal are countering each other. As the polar night is coming, the dispersal may include the upper layer under the sea ice, perhaps reflecting the lack of a light cue for the copepods to keep at depth as they would do at lower latitudes where there is light in winter. I have some evidence for *Calanus* species in the central Barents Sea to support this interpretation (published in the grey literature). It is of interest to note that the results from SHEBA suggest a decline in the proportion of both adult females and CVs of *C. hyperboreus* between mid-winter and late winter. SHEBA drifted at lower latitudes in the Amerasian Basin where winter darkness would be of shorter duration compared to MOSAIC.

In my view, it is not warranted to suggest partial migration for the population of *Calanus hyperboreus* in the CAO based on the MOSAIC results. The data is consistent with a concept of dislocation to depth AND dispersion, which in the high-latitude location with heavy sea ice in the CAO, includes dispersion all the way to the surface layer. It is in a way illogical to suggest the copepods are not translocating to depth but simply dispersing (which can only be effective in one direction – downwards) and then claim that a component of the population is not migrating but remains stationary in the upper layer. Is it implied that these copepods are not dispersing and thus have a different behavior from the rest of the population which does disperse? The lengthy discussion of bet-hedging and the ecological advantages of partial migration with reference to latitudinal migrations of birds and mammals, are not particularly relevant in the context of this paper.

We would like to bring three lines of evidence for partial migration:

1. In Fig. S1, we show 26 profiles of *Calanus hyperboreus* AF vertical distribution (2000-0 m) covering Nov 2019-Mar 2020 (winter) and July-Sep 2020 (summer-autumn). This is the highest temporal resolution of deep vertical profiles ever taken in the CAO, esp. when considering that it was done throughout Polar Night. All these 26 profiles

show a positive anomaly in the upper 200m (a green bar in opposite to a black bar), which means that there were always some animals slightly accumulating in the upper water column. Whether these were the same animals that did never migrate to great depth, or if some went up while others went down, we can't say. However, the net effect is the same: some animals remained near the surface, especially adult females (and our unpublished gut DNA data show that those females were feeding in winter).

2. When looking at the other polar species, Fig. 1, we see a continuum --- *Calanoides acutus* is the strongest migrant (having the least proportion of animals near the surface in winter) and *Calanus glacialis* the weakest (having the largest part of the population near the surface in winter). But, they all show partial migration --- some stronger, some weaker – but it is partial migration. Thus, 'partial migration' is a general phenomenon in seasonally migrating copepods.
3. Finally, when looking at the paper by Heath et al. (2000) on *Calanus finmarchicus* overwintering in the NE Atlantic, we see again that some animals remain in shallow waters (even there is no ice cover in the North Atlantic and therefore some light). This means that the ice cover in the CAO is not the main reason that animals remain near the surface, but that it is rather an inherent behaviour in some copepods. This partial migration has likely been developed or retained over evolutionary times as it incurs a benefit for the population, as we described in the 'evolutionary context' part of the *Discussion*.

Habitat and life cycle of *Calanus hyperboreus*

My third main comment in the first review was that there was too much emphasis on generalities and too little on specifics of the CAO. This relates to the previous point on partial migration. The paper could have focused more on the specific results on the distribution, reproduction and biochemistry of *Calanus hyperboreus*, and it could be shortened by removing the more speculative and not well substantiated parts.

There is no mentioning of the multi-year life cycle of *Calanus hyperboreus* in the CAO, which would be relevant in relation to the discussion of trade-offs between finding food and avoiding predators. *C. hyperboreus* may require up to 4 years to mature, and the species appear to be only partially successful in reproducing in the lower latitude portions of the CAO, and to be partly an expatriate in the high latitude central part of the CAO. The CAO is therefore not core habitat for *Calanus hyperboreus* (as claimed on page 11, line 276).

We used the term 'core habitat' in relation to temperature, but realise now that this caused confusion as there is also the aspect of food availability to consider. We now write 'a water column with suitable low temperatures as found in the CAO'.

On the contrary, this is on the fringe of its distribution, and the species is only marginally successful in the CAO. The core areas are the Greenland Sea and Baffin Bay.

The abundance of *Calanus hyperboreus* in the high latitude part of the CAO is typically 500-1000 copepodites m⁻² (Skjoldal, H. R. (Ed.). 2022. Ecosystem assessment of the Central Arctic Ocean: Description of the ecosystem. ICES Cooperative Research Reports Vol. 355. 341 pp. <https://doi.org/10.17895/ices.pub.20191787>). The abundances from the MOSAiC study fall generally in this range (Fig. 2c).

The presence of eggs and nauplii in winter suggests that some of the adult females were reproducing. I believe it is very likely that these were eggs and nauplii of *Calanus hyperboreus*, although it was unfortunate that they were not identified (e.g. from size measurements of the eggs). The abundance of eggs was ~3000 m⁻², compared to ~500

adult females m-2 (Fig. 2c). This gives an apparent egg production of 6 eggs per female, which is relatively low. This could reflect the low proportion of mature females, which appears to be ~10 % from comparison of Fig. 3 a and b. This suggests that only a small fraction of the adult females was actively reproducing at this high latitude location in the CAO. By comparison, Dawson (1978) found that 20-30 % of the females were mature in late winter in his sampling from the T-3 ice drift station.

Adult females generally produce eggs from stored body reserves, without the need to feed. This is what allows them to spawn at depth in winter.

There is some increase in the number of mature females in the upper layer during winter (Fig. 3b), which is implied to reflect winter feeding, supported by a trend in trophic markers in Fig. 3h. Winter feeding by *Calanus hyperboreus* is surprising to me. There is a reference to stomach content analysis based on personal communication from one of the coauthors. It would be useful if this evidence was provided, although I understand it is being prepared for publication in another paper. The evidence from the trophic markers is indirect. Also, without food, there would be a process of maturation driven by internal reserves. Therefore, an increase in proportion of mature females (still at a low level) needs not imply winter feeding.

Interestingly, Hirche 1991 also shows a higher proportion of mature *Calanus hyperboreus* females in the upper water column, compared to the deeper water column. This is despite the temperatures being equal (Hirche 1991) or even lower (our study) in the upper ocean. Thus, either mature females migrate towards the surface, or they are feeding during winter and fuel some extra carbon into the eggs, or both.

A key point related to Fig. 3b is the low proportion of adult females that mature. This suggests unsuccessful reproduction by *Calanus hyperboreus* at this high latitude location, in line with an interpretation that the species is largely an expatriate in this part of the CAO. This is an aspect that could have been included in the discussion.

We can understand Reviewer #3's disappointment about the fact that we do not write more about the general biology of *Calanus hyperboreus* in this manuscript, and withhold interesting data on their winter diet, egg production and life cycle. However, there are two reasons:

1. The MOSAiC zooplankton team received funding from different national funds and each 'nation' needs to show their own lead-publications to the funders. Thus, the feeding, respiration and egg production work is led by the US team; gonad development and species abundances/ vertical distribution by the AWI, and the biomarker studies by the UK ('Sym-pel' project). This means that data belong to different groups and as this is the first paper, the other data can't be incorporated here in detail. It seems that it is especially those data that are not available for this manuscript, that Reviewer #3 is most interested in. However, the reviewer should keep in mind that this present manuscript is already very rich in data, and should be appreciated in its own right.
2. The lead author of this manuscript is not an Arctic expert, but rather interested in animal migration in general --- partly due to previous work on the vertical migration of Antarctic krill. Therefore, general pattern in seasonal migration seems more appealing to the lead author than specific Arctic details. When reading up on seasonal migrants in general, the literature is dominated by specifics on birds, butterflies, large heard animals, whales etc. Thus, we wanted to say, 'Look here, copepods migrate too and even a similar distance' – 'How do they do it despite buoyance effects, we still do not really know'. So, the idea was to open the topic to a wider/ new audience, make it more general, try a journal with a higher impact factor,

bring in a few new ideas like the evolutionary context or the density measurements via pycnometer. This is to inspire new research avenues and encourage some cross-studies. A rather new approach to an old topic ...

Sorry, that Reviewer #3 can't appreciate our concept; but Reviewer #1 definitively did, writing: **'Overall: Very nice points mentioned in the discussion. The case you build on partial migration highlights the individual variability in the migration behaviour, that ultimately leads to a behavioural asynchrony.'**

Thus, we think, there is no major problem with our manuscript, it is just not so much what Reviewer #3 had in mind and is most interested in. However, to move a further step towards Reviewer #3 interests, we have now included the *C. hyperboreus* developmental stage composition for MOSAiC (Fig. S6), which is crucial to distinct between 'residence' and 'expatriates'. The data clearly show that *C. hyperboreus* completed its life cycle in the CAO during MOSAiC, and that our study was not carried out in a 'fringe habitat'.

Density measurements

The fourth point I raised in my first review was on the validity and ecological relevance of the measured tissue density. Density was measured on freeze-dried and crushed copepods (gently broken up with a spatula and partitioned into subsamples) with a pycnometer using helium gas to fill the voids in the freeze-dried and homogenized samples. There is just one general reference to Webb (2001), and no information that this is a valid method for determining density of zooplankton. With freeze-drying, water is removed from the copepods and there will be spaces in the dried material. These spaces are affected by shrinking in the drying process, and by the crushing and homogenization of the samples. I don't see how the original shape of the copepods (or crushed parts of the copepods) is inflated with helium gas back to the original volume that corresponded to the weight of the subsamples used for measurements of density. The results from these measurements of density look strange, characterized by very large variation from 0.6 to 1.7 g cm³ (Fig. 6c).

I am hesitant to accept these results before it is demonstrated that the method is valid for the purpose of investigating buoyancy of copepods and other zooplankton. Am I missing something here?

Validity: The measurements did not aim to establish the 'true' density of *C. hyperboreus*, which is a very difficult undertaking due to the easy uptake or release of water that can happen while the animals are sampled and handled onboard. Instead, we were interested in the expansion and shrinkage of the lipids under different temperatures that reflect the natural temperature gradient encountered by *C. hyperboreus* in the CAO. A crucial role of lipids in density control of copepods is known since a long time (see references in the manuscript), but studies at low temperatures as typical for the CAO have not been carried out yet. There are still methodical difficulties of doing so. Some studies have used 'oil' extracts and a densitometer (e.g. Sakinan et al. 2019), or lipid extracts and heating thermograms (e.g. Pond et al. 2012). These methods require rather large amounts of lipid that we did not have available from our MOSAiC samples, therefore we tried a different approach where we did not extract the lipids but use a highly sensitive method to measure the volume of solid materials (a pycnometer run with helium gas). This highly sensitive method allowed us to see differences in the volume of the lipid rich tissue along a temperature gradient, which normalised by the mass of the tissue results in density measurements. Thus, we are looking at the density of the lipids fixed to their membranes rather than measuring buoyancy of copepods, which is a different topic.

The ecological relevance of our results: The density can increase when animals are rich in cholesterol even when migrating into warmer waters. This is highly relevant in the CAO, because the lipid-packed copepods have to migrate and cross the warm Atlantic layer. If their lipids expand under rising heat, buoyancy increases, and the animal would pop up towards the surface. Interestingly, *C. glacialis* has less cholesterol and seems not to cross the warmer Atlantic layer in the CAO, while most deep-water species are enriched in cholesterol. Clearly, our study does not supply the final answer to density control, but it puts down an important milestone in the overall understanding of vertical migration and dispersal.

Some other issues

I have difficulty understanding Fig. 8, the conceptual model. What are the small individuals in the upper 200 m in July? Are they meant to be young copepodite stages (and were any such observed?), or are they CV and adult females shown with smaller symbols?

All symbols are of the same size, they might just occur smaller as they are white in the illuminated upper ocean and dark at depth.

Does the increase in number of symbols below 500 m between July and August-September illustrate the seasonal downward migration?

Yes.

Does the scheme overall suggest successful reproduction, and is there an implied multi-year life cycle?

This question is now rather answered by Fig. S6. The figure shows the vertical distribution of different developmental stages – inspired by our actual data, but standardised to 50 animals per column. Thus the data are specific for each of the season and developmental stage, comparing from one column to the next is not possible, as here we focus on the vertical distribution, not the seasonal cycle.

In section 3.5 on vertical distribution of potential predators, there is a consideration on predation risk which I find too simplistic. On lines 254-256, it is stated that only the deepest stratum below 1000 m is potentially a zone of lower predation risk for *Calanus*. However, predation risk is not only about the presence of potential predators but also about their density. There is evidence to suggest that predation is low overall, which allows *Calanus hyperboreus* to exist in relatively high abundance as, at least partially, an expatriate with a multiyear life span.

Here we compare the vertical distribution of predators not the horizontal distribution. Thus, the whole issue of *Calanus hyperboreus* lifecycle and resident vs expatriate is not really the tissue of this manuscript, but will be dealt with by our co-authors within a different manuscript.